Resource

# Unified mass imaging maps the lipidome of vertebrate development

Halima Hannah Schede [1,2,6], Leila Haj Abdullah Alieh[1,2,6], Laurel Ann Rohde[1], Antonio Herrera [2], Anjalie Schlaeppi [3], Guillaume Valentin [4], Alireza Gargoori Motlagh [2], Albert Dominguez Mantes[1,2], Chloe Jollivet[1], Jonathan Paz-Montoya [1], Laura Capolupo [1,5], Irina Khven[2], Andrew C. Oates [1], Giovanni D'Angelo [1,7] ✉ & Gioele La Manno [2,7] ✉

Embryo development entails the formation of anatomical structures with distinct biochemical compositions. Compared with the wealth of knowledge on gene regulation, our understanding of metabolic programs operating during embryogenesis is limited. Mass spectrometry imaging (MSI) has the potential to map the distribution of metabolites across embryo development. Here we established uMAIA, an analytical framework for the joint analysis of large MSI datasets, which enables the construction of multidimensional metabolomic atlases. Employing this framework, we mapped the four-dimensional (4D) distribution of over a hundred lipids at micrometric resolution in *Danio rerio* embryos. We discovered metabolic trajectories that unfold in concert with morphogenesis and revealed spatially organized biochemical coordination overlooked by bulk measurements. Interestingly, lipid mapping revealed unexpected distributions of sphingolipid and triglyceride species, suggesting their involvement in pattern establishment and organ development. Our approach empowers a new generation of whole-organism metabolomic atlases and enables the discovery of spatially organized metabolic circuits.

With the rise of single-cell and spatial omic methodologies, our capacity to describe cell type composition and developmental trajectories has advanced substantially. Omic atlases continue to expand our understanding of the different factors influencing cell identity and positioning in both embryos and adult organisms[1–6].

Currently, compositional atlases rely on the ease of detection of nucleic acids and assume that transcriptional profiles are accurate descriptors of cell types and tissue structures[5]. However, cell identity is not solely determined by gene expression, and organs are structured toward a metabolic division of labor[7,8]. Therefore, mapping detailed biochemical compositions of tissues and organs can reveal an important axis of cell state heterogeneity and tissue structure.

To understand the biochemical tapestry of cells and tissues, a focus on lipids promises substantial advances. Fundamental cellular processes rely on lipids[9–12], which also play roles in cell communication, contributing to organism physiology regulation[7,13,14]. Moreover, we recently showed that single-cell lipid configurations drive cell states involved in tissue patterning[15]. During development, lipids display

[1]Institute of Bioengineering, School of Life Sciences, Swiss Federal Institute of Technology (EPFL), Lausanne, Switzerland. [2]Brain Mind Institute, School of Life Sciences, Swiss Federal Institute of Technology (EPFL), Lausanne, Switzerland. [3]Bioimaging and Optics Core Facility, School of Life Sciences, Swiss Federal Institute of Technology (EPFL), Lausanne, Switzerland. [4]Center of PhenoGenomics, School of Life Sciences, Swiss Federal Institute of Technology (EPFL), Lausanne, Switzerland. [5]Friedrich Miescher Institute for Biomedical Research, Basel, Switzerland. [6]These authors contributed equally: Halima Hannah Schede, Leila Haj Abdullah Alieh. [7]These authors jointly supervised this work: Giovanni D'Angelo, Gioele La Manno. ✉e-mail: giovanni.dangelo@epfl.ch; gioele.lamanno@epfl.ch

complex dynamics in vertebrates[16,17]. Yet, despite their relevance, our understanding of how lipid compositional heterogeneity is spatially organized and established at the organism level is limited.

MSI enables the study of the spatial organization of lipids, as it measures the distribution of compounds at micrometric scales in a parallelized and untargeted manner[18,19]. Today, MSI acquisitions are the measurement of choice to obtain unsupervised two-dimensional biochemical descriptions of samples. However, only by combining multiple MSI acquisitions can one create complete maps that survey lipids along organs and organisms (three dimensional; 3D) over time (4D) and upon perturbation (Supplementary Fig. 1).

We performed matrix-assisted laser desorption–ionization (MALDI)-MSI of entire vertebrate embryos at different stages of development to study the establishment of metabolically defined anatomical zones (lipid territories) (Fig. 1a,b). We considered embryos from *D. rerio* (zebrafish) that undergo substantial lipid remodeling during development[20,21].

Studies have attempted to localize different lipid species in zebrafish. However, these analyses had limited spatial resolution or temporal scale, distinguishing just a few regions of the embryo[22,23] or focusing only on small developmental windows[24]. A more detailed description of lipid spatiotemporal dynamics requires analyzing high-spatial resolution MS images of entire organisms over development. However, large MSI datasets are challenging to integrate due to technical variability in sample preparation and instrument performance. This variability causes artifacts, including missing detections at the level of pixels or molecules and nonbiological signal intensity deviations[25–27], that are common to MSI datasets (Supplementary Fig. 2). Ad hoc measures, including the use of internal standards (ISs), have been devised to mitigate these effects, but they require prior knowledge of sample composition, which is incompatible with untargeted approaches[27]. For this reason, in previous work, MSI dataset analysis was limited to the application of supervised machine learning approaches or the study of major tissue compartments[28–30]. Here, we devised a computational framework, the unified Mass Imaging Analyzer (uMAIA), to tackle these problems. uMAIA allows the generation of 4D lipid atlases of whole organisms from large collections of raw MSI acquisitions (Fig. 1c).

With uMAIA, we mapped a sizable fraction of the lipidome of zebrafish embryos during development. We studied lipid spatiotemporal changes in the absence of external confounding factors, as zebrafish embryogenesis entails a closed metabolic system in which macromolecular precursors are provided by yolk storage[8]. We traced the emergence of lipid territories that recapitulate embryo anatomical organization and found unexpected distributions of lipid classes that are relevant for tissue-specific functions. Altogether, our work demonstrates that the vertebrate developmental program involves fine-grained and spatiotemporally regulated remodeling of

metabolism. It also highlights that lipid distribution is an accurate and information-rich descriptor of vertebrate anatomy, reinforcing the link between cell identity and lipid metabolism[31].

## Results

### Count-based adaptive peak calling obtains precise mass images

The uMAIA framework addresses limitations of standard MSI data processing. It achieves this by accurately extracting images from spectra (peak calling), combining them in a shared feature space across acquisitions (featurization) and normalizing intensities to minimize experimental fluctuations (normalization) (Fig. 1c).

Image extraction from raw MSI data involves identifying peaks generated by isobaric molecules across spectra (pixels). Instrument-driven fluctuations around peak $m/z$ values (mass shifts) complicate this process. Post-acquisition alignment methods have been developed to minimize these errors[32,33]. However, calibration only partially corrects the problem[32–34], as we show in different analyses (Fig. 1d–f and Extended Data Fig. 1a–e).

Thus, to extract images from MSI data, $m/z$ intervals that are assumed to encompass signals from isobaric molecules must be considered[35]. Commonly used methods retrieve intervals of fixed sizes or for which extensions scale with $m/z$[36,37] but do not account for variability in peak-specific mass shifts, which often causes artifacts. Adaptive approaches that attempt to encompass mass shifts by retrieving intervals from intensity peaks have been proposed as a solution[38]. However, such approaches bias intervals to consider mainly peaks with high intensities even though lower-intensity peaks contain equally relevant spatial information. Therefore, we reasoned that counting the number of detection events over spectra, disregarding intensity, could better expose peak mass shift distributions (Fig. 1g). Based on this principle, we devised an approach inspired by the watershed algorithm[39] in which $m/z$ intervals are initialized at histogram maxima and expanded until other intervals are encountered or reach a background level of event counts (Fig. 1h and Methods). This procedure produces $m/z$ intervals tailored to the mass shifts of individual peaks.

We benchmarked our approach using data from different experiments and instruments against non-adaptive (Mirion[37]) and adaptive (MALDIquant[38]) methods. Inspection of images extracted by the different methods suggested that uMAIA's peak caller effectively captured individual peaks (Fig. 1i and Extended Data Fig. 1f–i) and better resolved neighboring ones (Fig. 1j). uMAIA was more precise than Mirion in distinguishing quasi-isobaric compounds (2.3% versus 42% of images containing aggregated peaks) (Extended Data Fig. 2a). Conversely, MALDIquant generated multiple images that were similarly distributed and showed a checkerboard-like spatial complementarity, suggesting that the signals originated from single ions (Extended Data Fig. 2b).

**Fig. 1 | Study design and peak-calling performance of uMAIA and other algorithms. a**, Experimental design. Zebrafish embryos at selected developmental stages were sectioned, and MALDI-MSI was performed for alternating sections. **b**, Four-dimensional metabolic atlas analysis of regional and developmental trends. **c**, Illustration of uMAIA modules: (1) adaptive peak calling: image extraction from MSI acquisitions, (2) featurization: identifying ions across acquisitions and (3) normalization: reducing experimental variability in intensity distributions. **d**, Peak positions (Da) and intensities across three spectra (indicated with colors) for three molecules of comparable $m/z$ (estimated values shown with dashed lines). Px., pixel. **e**, Scatterplot between mass shifts of two molecules of similar mass. Top right: Pearson's $R$ indicated. **f**, Heatmaps of extent and direction of mass shifts in space for molecules in **d** and two other molecules of similar mass. **g**, Illustration of the peak-calling approach. Spectra are reduced to a frequency (across pixels) histogram (left). Plots of 1,000 spectra stacked above corresponding histograms for 3 representative cases (right). **h**, Overview of uMAIA's adaptive peak-calling algorithm (left). Intervals are iteratively expanded in the frequency histogram, decreasing a threshold

$(t_0, t_1, t_2)$ as new intervals are considered. Peak intensity is integrated for each interval across spectra, generating images (right). **i**, Representative example of bins called using MALDIquant, Mirion and uMAIA over the frequency histogram (top) and corresponding images (bottom). Binning intervals shown by lines (color coded by method). **j**, Frequency histograms (top) and images (bottom) featuring a case of different peak-calling behavior among MALDIquant, Mirion and uMAIA. **k**, Sensitivity analysis on real data comparing uMAIA, MALDIquant and binning for Orbitrap and Fourier-transform ion cyclotron resonance (FTICR) MS acquisitions. Venn diagrams depict set sizes of molecules passing three assessment criteria to estimate true positive (TP) counts. Numbers within Venn diagrams indicate quantity of high-confidence TPs satisfying all criteria. FP estimate indicated (gray number within outer circle). **l**, Left: line plot tracking performance (average mutual information score) of uMAIA, binning and MALDIquant on simulated spectra ($n = 50$; shaded area depicts s.d.). Right: heatmaps displaying average mutual information score over simulations for different SNRs and peak densities ($n = 50$).

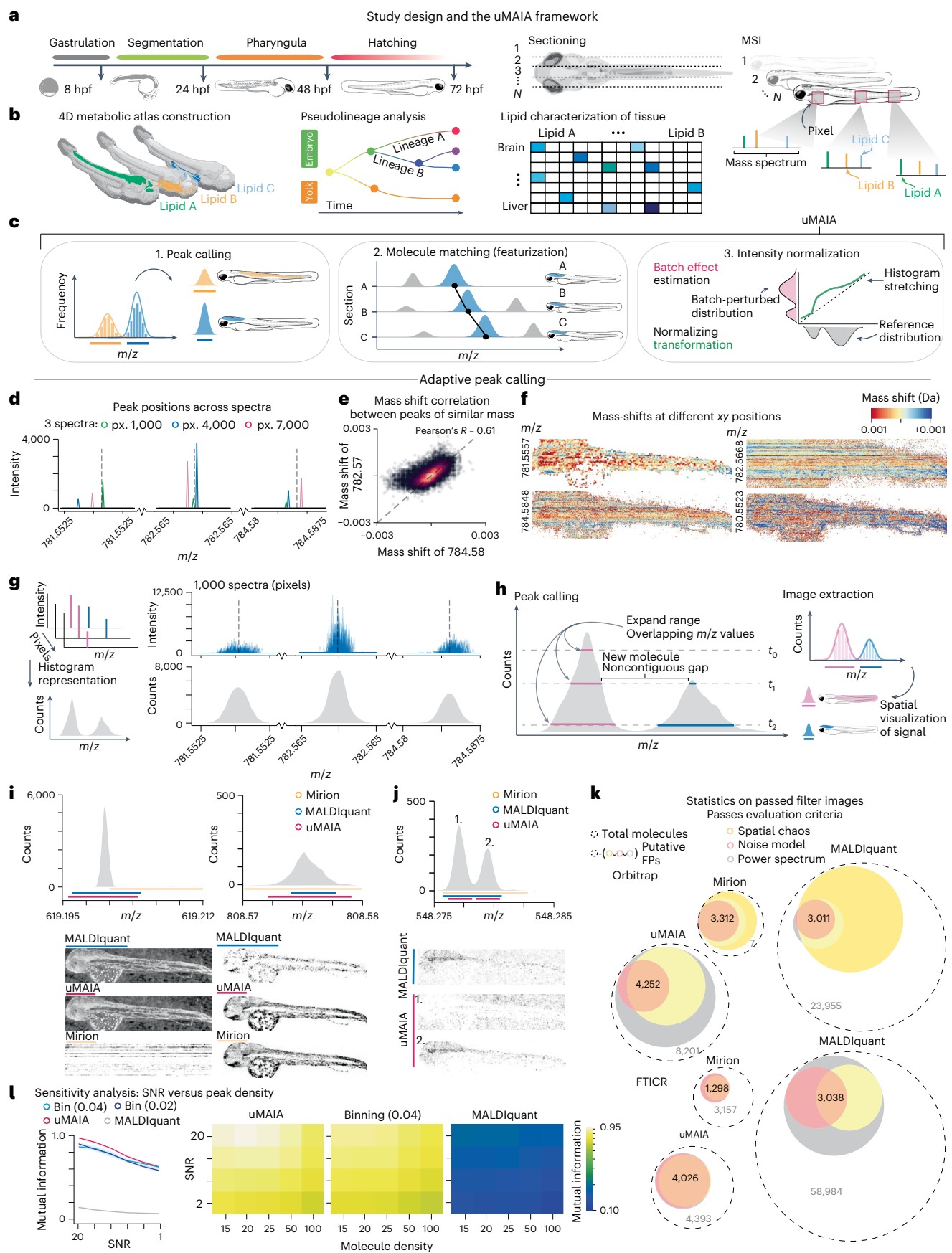

Notably, peaks detected by each method typically preserved signal spatial distributions (Extended Data Fig. 2c). Overall, uMAIA retrieved up to 55% more high-quality images (Methods) than the runner-up method, depending on the MSI technology (Fig. 1k and Extended Data Fig. 2d–f). These findings were confirmed with simulated data where we controlled for different signal-to-noise ratios (SNRs) and peak density (Extended Data Fig. 2g). uMAIA broadly outperformed existing methods, with a significantly greater signal recovery per peak than the next-best method (mutual information score uMAIA = 0.98, binning = 0.8, $P$ value = $2.03^{-16}$) (Fig. 1l). Notably, when data were preprocessed with standard mass alignment[33], uMAIA retrieved 33% more high-quality images (Extended Data Fig. 2h and Methods), while processing times remained within the same order of magnitude (Extended Data Fig. 2i).

### Network flow-based matching creates a unified feature space

To analyze datasets comprising multiple acquisitions, peaks representing isobaric molecules must be identified (or 'matched') across different samples to featurize the data. Typically, this task is accomplished by aligning spectra across samples using reference peaks, followed by binning (Fig. 2a). While straightforward for a few samples and molecules, this approach is difficult to scale to tens or hundreds of acquisitions, which increases spurious matches. To obtain coherent matchings in large datasets, we conceived an unsupervised and alignment-independent approach that automatically links peaks and tested it on our dataset, for which we profiled the mass accuracy (Fig. 2a and Extended Data Fig. 3a–c). We consider the possible links between peaks and their mutual positioning to pose a network flow problem that can be efficiently optimized (Methods). This formulation incorporates constraints to avoid solutions representing inconsistent scenarios, including cases in which a molecule has multiple matches originating from one acquisition.

We tested our approach on simulated and real data and benchmarked it against binning procedures (Methods). First, experiments on simulated data highlighted that our method performed best for accuracy, recall and precision and was robust to the number of sections provided (Extended Data Fig. 3c). Next, we evaluated our approach on our zebrafish dataset (Fig. 2b,c). Due to the absence of a ground truth, we considered two cases with reliable expectations. In the first case, we used signals from [M + 0] and [M + 1] naturally occurring isotopologs, which should be detected in the same acquisitions. Selecting the 20 most intense lipid peaks and their isotopologs across the dataset indicated that uMAIA more consistently identified isotopologs in the same set of acquisitions than binning methods (Fig. 2d and Extended Data Fig. 3d). For the second case, we considered the 50 most intense MALDI matrix peaks (α-cyano-4-hydroxycinnamic acid; CHCA) that are present across acquisitions as matrix is applied to all samples. To evaluate the matching here, we introduced an ambiguity score to identify spurious matches by counting multiple detections from the same section that cannot originate from isobaric molecules (Fig. 2e). We quantified the occurrence of missing or multiple (ambiguous) matches using different binning sizes or uMAIA. We found that uMAIA maximizes the number of matched peaks while excluding ambiguous detections (Fig. 2f,g).

We next asked to what extent peaks could be matched across the entire dataset. When binning approaches were applied, the number of matched peaks and ambiguity scores scaled with bin sizes: smaller bins prioritized unambiguous detections at the expense of numbers of matched molecules, while increasing bin sizes retrieved larger groups but had higher ambiguity (Fig. 2h). Our approach restricts groups to be unambiguous and retrieves a greater number of peaks across all acquisitions, corresponding to a 20% increase in detected signals (uMAIA = 1,200, binning = 1,000), than the largest bin size tested (0.01) (Fig. 2h–j). This improvement was maintained even after spectral alignment (Extended Data Fig. 3e–g and Methods).

Thus, uMAIA extracts peaks from multiple MSI acquisitions and featurizes the dataset reliably. Importantly, uMAIA is annotation free, thereby enabling further analyses on larger portions of the data, and efficient: over 50 sections and 20,000 molecules can be featurized in <15 min (Extended Data Fig. 3h).

### uMAIA transforms intensities to reduce technical variability

Even when peaks are called and matched, integrating MSI acquisitions remains challenging due to batch effects resulting from experimental variability that deform intensity distributions[25]. While, in the genomic field, many batch correction methods have been proposed[40–45], MSI data substantially differ in signal distributions and batch effect properties.

We observed that logged intensity values of MALDI-MSI data exhibited bimodal distributions with low (background) and high (foreground) modes (Extended Data Fig. 4a). Variable magnitudes of intensity distortions were observed across foreground modes in peaks from consecutive sections. This was seen even for matrix peaks that should have similar dynamic ranges across sections (Fig. 3a). We hypothesized that the deformations have factorizable components: one molecule specific and another acquisition specific. To investigate this aspect, we derived an ad hoc approximated empirical estimator of batch effect shifts. We exploited a property of our dataset, which is that true biological intensity should vary smoothly between consecutive sections (Methods). Next, we visualized the matrix structure of batch effect estimates (Fig. 3b). Singular-value decomposition revealed that the matrix of empirically estimated batch effects was low rank: approximately 31% of its total variability could be explained by its rank-one approximation, confirming the hypothesis of a strong factorization of the batch effect (Extended Data Fig. 4b,c). Based on this verified relationship, we devised a probabilistic model to perform a regularized estimate of these factors and remove nonbiological distortion from intensity distributions (Fig. 3c, Extended Data Fig. 4d,e and Methods).

First, we tested our approach on two simulated datasets, applying distortions that mimicked batch effect characteristics from real datasets (Extended Data Fig. 4f and Methods). The first was constructed to facilitate interpretation: overlapping ellipsoids were generated with different average signal intensities. To challenge our model with more complex and realistic spatial distributions, we considered a second simulation based on expression patterns from the Allen Brain Atlas (ABA)[46]. We simulated batch effect intensity distortions for each section in both datasets, corrected the data with uMAIA, $z$ normalization and ComBat, a popular method for batch effect normalization[42], and evaluated how well each algorithm approximated ground truth intensities (Extended Data Fig. 4g–p and Supplementary Figs. 3 and 4).

uMAIA performed well in all cases and was both qualitatively and quantitatively superior to other methods in instances in which low-intensity values risked to be stretched to signal, implying appropriate regularization of the estimated parameters. This was indicated in the first dataset in which uMAIA recovered the ordering of the modes from raw data and reduced their variances, while ComBat and $z$ normalization intermixed the intensities of distinct modes by assuming normally distributed data (Extended Data Fig. 4h–j). The result was also reflected in the ABA-based simulation: root mean square error scores between ground truth and normalized pixel values were lowest for uMAIA-corrected data (mean 89% improvement), followed by ComBat (mean 23% improvement) and worst for $z$ normalization (mean 17% decline) (Extended Data Fig. 4m). Additionally, multivariate analysis qualitatively indicated that uMAIA-corrected data better recapitulated the ground truth (Supplementary Fig. 4a,b).

We next asked to what extent each method enabled biologically meaningful comparative analysis between regions. We selected five major ABA regions that spanned multiple sections and applied a two-sided $t$-test between pairs of regions using average values within sections (Methods). Comparing these results to ground truth images identified false negative (FN) and false positive (FP) detections

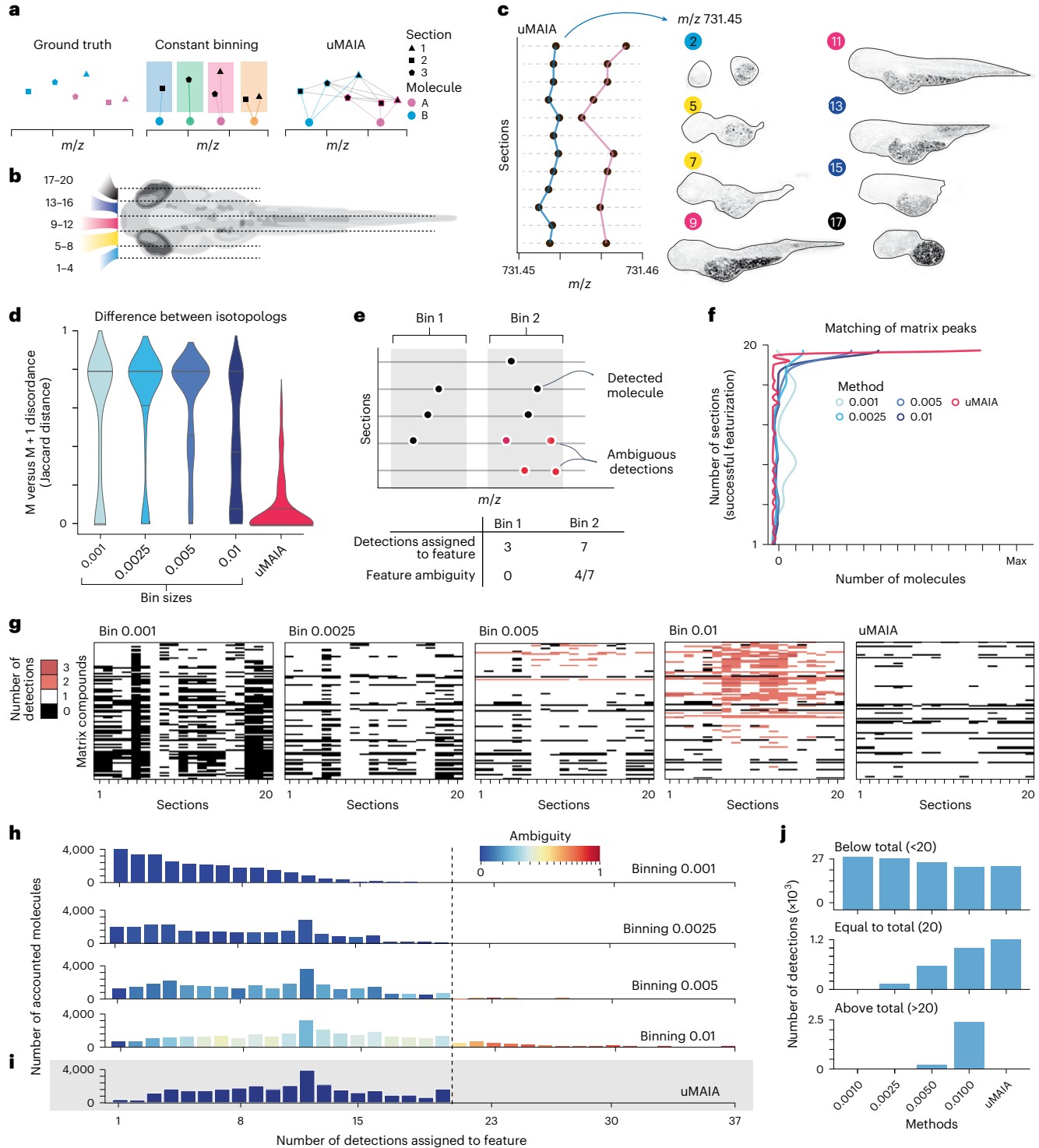

**Fig. 2 | Assessment of unified feature space by uMAIA's peak matching and comparison with binning approaches. a**, Illustration of the peak-matching problem to place acquisitions into coherent feature spaces. Ground truth, solutions of the standard binning approach and uMAIA. **b**, Illustration of sagittal sectioning of zebrafish at 72 hpf for MSI. Numbers indicate section position. **c**, Example of molecules matched across sections with detected *m/z* values shown (left) and visualization of the corresponding molecule, *m/z* = 731.45 (right). The color and number of the section refer to the scheme in **b**. **d**, Violin plots reporting the distribution of Jaccard distances between isotopolog M + 0 and M + 1 presence across acquisitions after featurization. Distributions are computed across all pairs of sections for 20 molecules with the highest signal intensity. Horizontal lines indicate means. **e**, Illustration clarifying the definition of feature ambiguity and score calculation. **f**, Density plots displaying frequency

of different peak-matching outcomes for 50 MALDI matrix ions. *y* axis quantifies the number of sections from the experiment in **b** where featurization was successful. Max, maximum. **g**, Heatmaps of MALDI matrix ions (rows) identified over sections (column) for different binning sizes and uMAIA. Color coded according to whether zero, one, two or three peaks were identified within the bin for a given section. Matrix compounds are displayed as in **f**. **h**, Bar plots of the distribution of different peak-matching outcomes. *x* axis quantifies the number of sections in which a compound was identified across the dataset. Bars are colored according to the average ambiguity within the range. Vertical dashed line indicates total number of sections in the dataset. **i**, As in **h**, corresponding to uMAIA-retrieved sets. **j**, Bar plots of different peak-matching outcomes stratified by features: not detected in all sections (less than 20 sections), all sections (exactly 20 sections) or more than 20 sections for each method.

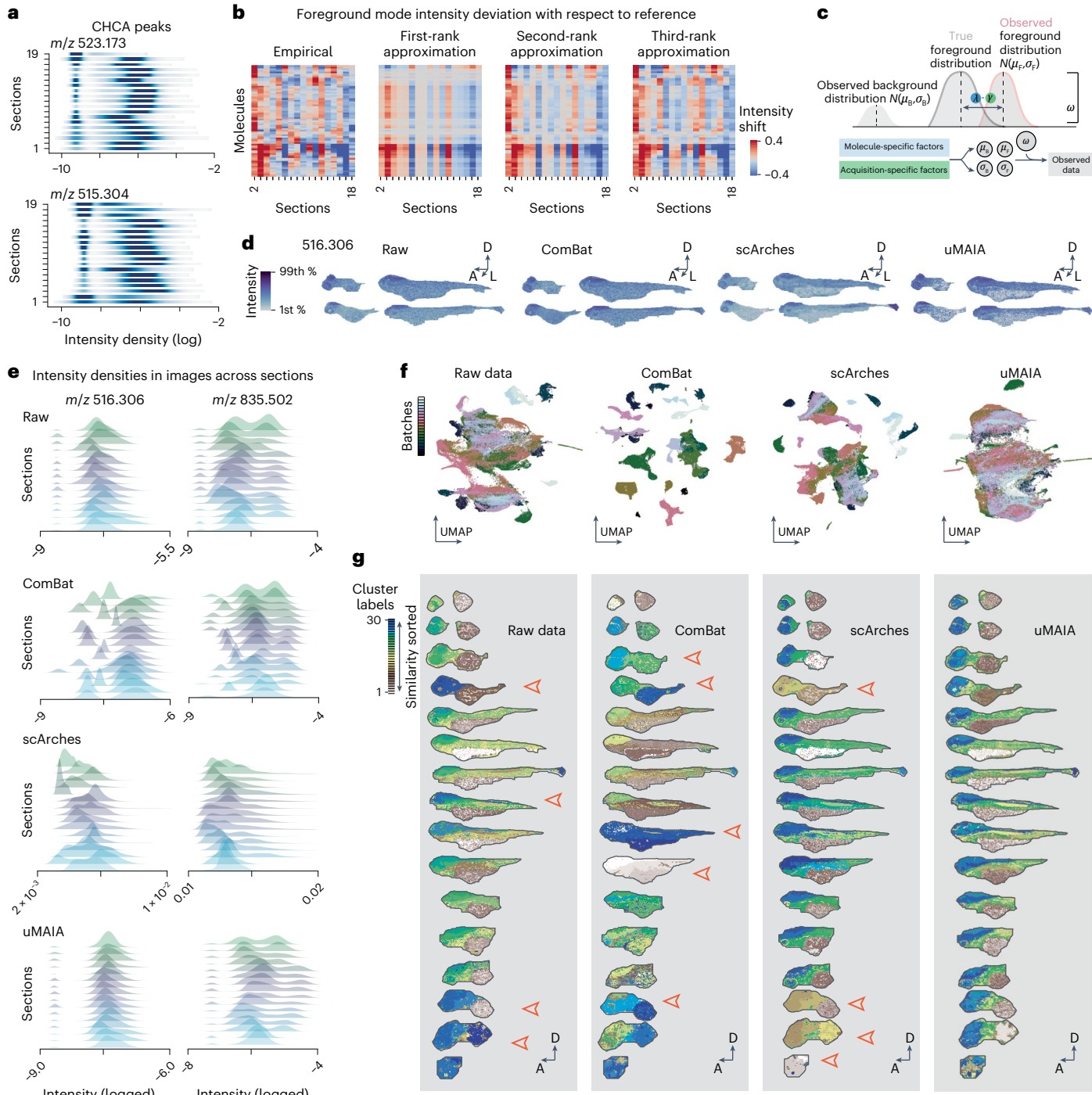

**Fig. 3 | Characterization of technical variability in MSI data: intensity distortions and uMAIA normalization. a**, Intensity distribution densities of two CHCA (matrix) peaks across sections of the 72-hpf zebrafish dataset. **b**, Far left: heatmap displaying empirical estimates of foreground mode intensity shifts for 50 molecules ('Batch effect characterization' in Methods). Heatmaps on the right represent the first-, second- and third-rank approximation of the empirical matrix. Approximations were obtained by the outer product of the group of one, two or three singular vectors, respectively. **c**, Overview of the key modeling idea of distribution shifts. Signal distribution is approximated by a bimodal distribution (that is, a mixture of two Gaussians) with foreground and background mode. Observed foreground distribution parameters: center and standard deviation are subjected to displacement with an offset, which we consider factorizable in two sources (molecule and acquisition specific:

$\lambda$ and $\gamma$, respectively). **d**, MSI raw and normalized images with different methods (ComBat, scArches, uMAIA) for representative molecules across landmark sections of the 72-hpf dataset. Color bars span between the 1st and 99th quantiles (1st % and 99th %) of intensities across all sections. D, dorsal; A, anterior; L, lateral. **e**, Intensity distributions of molecules before and after normalization using different methods across sections. **f**, Low-dimensional representation (uniform manifold approximation and projection, UMAP) of pixels for raw data and outputs after ComBat, scArches and uMAIA normalization. Pixels are color coded by the section from which they originate. **g**, Spatial visualization of discrete clusters after application of the $k$-means algorithm on raw data and data processed with ComBat, scArches and uMAIA. Arrows indicate clear residual batch effects after clustering. The 72-hpf zebrafish data were used.

(Supplementary Fig. 4c). uMAIA followed ground truth trends more often than other methods, resulting in lower FP and FN rates (FP rate (FPR) = 0.07, FN rate (FNR) = 0.12) than $z$ normalization (FPR = 0.15, FNR = 0.24) and ComBat (FPR = 0.18, FNR = 0.22), approaching the lower bound (FPR = 0.05, FNR = 0.08; Methods and Extended Data Fig. 4o,p).

To assess the ability of the methods to correct intensity distortions in real data, we applied them to our featurized zebrafish dataset. In addition to ComBat, we tested a deep learning method, scArches, that recombines features to integrate data[47]. uMAIA normalization reduced high-frequency intensity differences that were likely attributable to experimental noise while retaining signal mean levels (Fig. 3d,e and Extended Data Fig. 5a–d). Without a ground truth for this dataset, we evaluated performance using multivariate and clustering analysis jointly applied to the pixels across sections, expecting similar clusters in consecutive sections with successful normalization. We used ComBat, scArches and uMAIA to normalize the dataset and applied principal-component analysis (PCA) followed by $k$-means clustering[42,47] (Extended Data Fig. 5e,f). Pixel clusters after uMAIA normalization were the most consistent across sections (Fig. 3f,g and Extended Data Fig. 5f). In addition, alternative models of the batch effect were less effective at correcting intensity distortions (Extended Data Fig. 6 and Methods). Analysis of a proteomic and metabolite MSI dataset suggested that our method was also effective at reducing intensity distortions captured by different modalities (Supplementary Fig. 5).

Overall, uMAIA enables MSI data integration for whole-organism and organ atlases, helping reveal patterns in lipid distributions.

## Lipid territories emerge and diversify over development

Previous studies indicated that lipids undergo substantial remodeling throughout zebrafish development[20,21]. To clarify the extent of such changes between developmental milestones, we performed bulk lipidomics by maximum-coverage, high-throughput targeted hydrophilic interaction liquid chromatography coupled to electrospray ionization tandem MS (HILIC–ESI-MS/MS or LC–MS/MS)[48,49]. We selected zebrafish embryos at four developmental stages: gastrula (8 h after fertilization (hpf)), late somitogenesis (24 hpf), pharyngula (48 hpf) and hatching (72 hpf)[50] (Fig. 4a). We quantitatively profiled 850 individual lipid species in zebrafish embryos, revealing substantial lipid remodeling during development (Fig. 4b,c, Supplementary Fig. 6 and Supplementary Table 2). Specifically, hexosylceramide levels increased the most, rising more than tenfold from 8 to 72 hpf. Levels of ceramides, sphingomyelins (SMs), phosphatidylserines (PSs) and phosphatidylglycines (PGs) also increased, while those of phosphatidylcholines (PCs) and lyso-PCs (LPCs) significantly decreased (Fig. 4c). Triacylglycerols (TGs), primarily stored in the yolk and serving as energy and building blocks for embryogenesis, showed no collective concentration change (Fig. 4c and Supplementary Fig. 6a–c) but surprisingly increased for species with long fatty acid (FA) chains (≥54 carbon atoms) (Supplementary Fig. 6d,e).

However, bulk measurements cannot reveal tissue-specific compositional changes, thereby overlooking the importance of lipids during tissue patterning. To properly investigate lipid spatial distributions, we built a 4D atlas by collecting MALDI-MSI acquisitions of sagittal sections encompassing whole embryos at each developmental stage and applied uMAIA. We annotated 176 lipids by $m/z$ matching using bulk lipidomics data and reconstructed their 3D distributions (Fig. 4d, Extended Data Fig. 7 and Supplementary Table 1). To increase our confidence in the annotation, we performed a global renormalization of MALDI and LC–MS/MS signals and calculated the correlation of lipids detected between methods (Methods and Extended Data Fig. 7g–i). Lipids that did not correlate were discarded as potential FPs. The images indicated a substantial spatial reorganization over time, with several lipids localizing to specific structures by 48 or 72 hpf (Fig. 4d).

We systematically assessed these changes and found that the number of spatially patterned lipids more than doubled from 26 at 8 hpf to 58 by 24 hpf, increasing to 77 at 48 hpf and 100 at 72 hpf (Fig. 4e,f, Extended Data Fig. 8a and Methods). The initial increase observed at 24 hpf aligns with the end of segmentation when primary organs and somites are established[50]. Using lipid levels as features, we performed PCA and pixel clustering at each time point. Resultant clusters were spatially coherent within embryos, defining regions that we named 'lipid territories'. Analysis of lipid territories throughout development revealed an overall increase in lipid distribution complexity (Fig. 4g,h).

Using lipid composition similarity to connect lipid territories across time points, we observed that metabolically related territories occupied equivalent regions at each stage (Fig. 4h). As complexity increases during development, early lipid territories diversified, defining metabolic pseudolineages (Fig. 4h and Methods). Notably, unlike posterior territories, anterior metabolic regions did not appear to diversify, suggesting that their lipid composition is already defined at early stages (Fig. 4h). Supporting this finding, anterior territories at earlier stages harbored more diverse metabolic content than posterior ones (Extended Data Fig. 8b and Methods).

PC lipids underwent substantial spatial reorganization during development. Over time, a two-component structure emerged in PC lipid covariance, intriguingly revealing a partitioning according to unsaturation degree (Fig. 4i). PC species with fewer double bonds localized more anteriorly, mainly in the head, while those with more unsaturations were found posteriorly (Fig. 4j and Extended Data Fig. 8c–e).

Overall, this analysis indicates that lipids finely diversify spatiotemporally during development. Biochemical remodeling and lipid territory emergence occur over time, with increasing complexity present as development proceeds. Importantly, these results, revealed by spatial analysis, were not deducible from bulk lipidomic experiments.

## Lipid-defined anatomy of a zebrafish embryo at 72 hpf

The compelling finding that lipid metabolism is spatially organized motivated us to directly investigate the correspondence between anatomical structures and biochemical organization. We focused on the zebrafish at 72 hpf, the most mature time point in our dataset, as it harbored the greatest complexity in tissue lipid composition.

Annotation using LC–MS/MS data identified 142 lipids (122 membrane lipids) in our MSI dataset. This enabled the 3D reconstruction of approximately 27.6% of membrane lipid species, accounting for 73.3% of the overall membrane lipid molar fraction (excluding cholesterol). The reconstructed lipids spanned various classes (in number of species; mol%), including PC (44; 66.64% of imaged lipids), TG (20; 26.71%), PS (16; 0.80%), phosphatidylethanolamine (PE) (15; 1.88%), LPC (12; 1.43%), PG (9; 0.05%), phosphatidylinositol (8; 0.37%), SMs (6; 1.29%), diacylglycerol (5; 0.74%), lysophosphatidylinositol (3; 0.0004%), lysophosphatidylethanolamine (2; 0.003%), ceramides (1; 0.05%) and hexosylceramide (Hex1Cer) (1; 0.04%) (Extended Data Figs. 7d–f and 9).

Remarkably, specific lipids delineated regions including the yolk and yolk extension (TG 52:4), the nervous system (PC 32:0), the hindbrain (PE 38:6) and musculature and internal organs (PC 32:2) (Fig. 5a,b and Supplementary Fig. 7a). This correspondence between lipid distributions and anatomy generalized across the lipidome, as revealed by low-dimensional data representations (PCA and nonnegative matrix factorization)[51]. The spatial distribution of pixel PCA coordinates indicated strong anatomical covariance of lipid abundance (Fig. 5c). Nonnegative matrix factorization factors further resolved this covariance by decomposing the data into interpretable lipid contributions, representing axes of lipid program modulation (Supplementary Fig. 7b). Additionally, diffusion map embeddings of pixels uncovered finer covariations, highlighting subtle distribution differences between related lipids including LPC 18:1 and LPC 22:6 (Supplementary Fig. 7c,d). A sorted distance matrix of voxel similarity in lipid space revealed a block-like pattern, suggesting a finite number

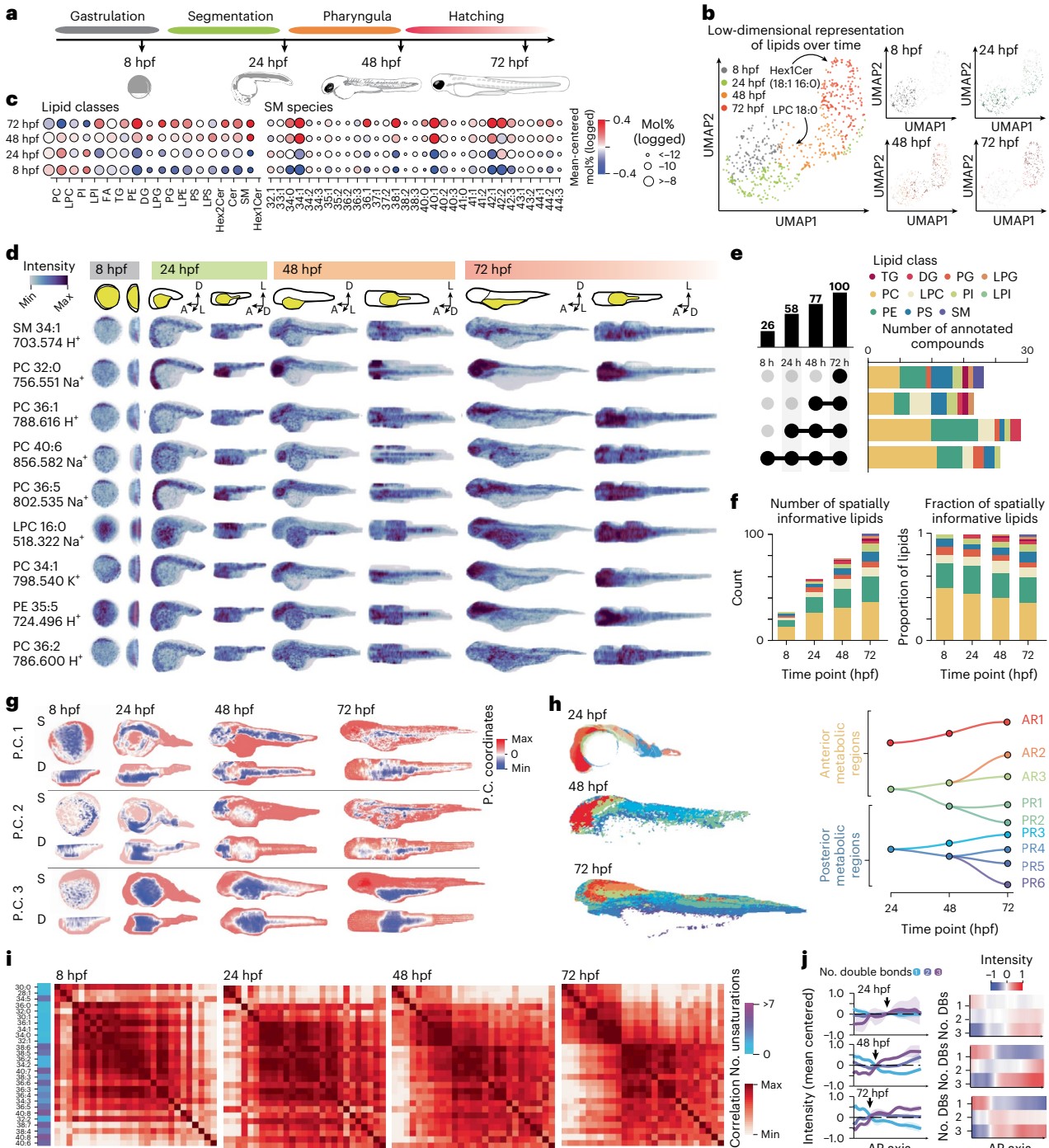

**Fig. 4 | Four-dimensional metabolic atlas of zebrafish development.**
**a**, Overview of developmental stages used for bulk LC–MS/MS and MALDI-MSI acquisitions with selected developmental stages indicated. DG, diacylglycerol; LPE, lysophosphatidyletanolamine; LPI, lysophosphatidylinositol; PI, phosphatidylinositol. **b**, Low-dimensional representation (UMAP) of bulk LC–MS/MS data. Points represent lipids and are color coded by stage at which concentration peaks (left) or relative changes in concentration over time (right). **c**, Dot plot indicating overall lipid concentration changes and absolute quantities averaged over classes (left) and specific SM species (right) for each developmental stage. Cers, ceramides; HexCers, hexosylceramides. **d**, Four-dimensional distributions for selected lipids for analyzed time points (sagittal and dorsal projections, schematic of orientation shown in top row with yolk outlined in yellow). Note that the precise orientation of the embryo at 8 hpf is unknown. Min, minimum. **e**, Quantification of spatially localized lipids across the sampled time points. Upset plot depicting total molecule counts for each time point (left, *n* = 1). The time points that are considered within each set are

indicated by solid black circles. Stacked bar plot depicting lipid class breakdown for different sets shown in the upset plot (right). **f**, Stacked bar plots representing the total number (left) and proportions (right) of spatially informative molecules detected at specified time points (*n* = 1) stratified by lipid class. **g**, Visualization of the first three principal components (P.C.) over each developmental stage, sagittal (S) and dorsal views. **h**, Tissue clusters (left) and pseudolineages (right) across sampled time points split generally into anterior and posterior metabolic regions. AR, anterior region; PR, posterior region. **i**, Correlation matrices between PC species for each developmental stage sorted according to optimal sorting at 72 hpf with SPIN[69]. Color bar indicates the number of unsaturations for each lipid species. **j**, Lipid intensity trends for PC species along the anteroposterior (AP) axis for each developmental stage (left). Shaded area indicates standard deviation between biological replicates (*n* = 2). Varying degrees of unsaturation are indicated by colors. Heatmap of mean-centered average lipid intensity for three species over developmental time (right).

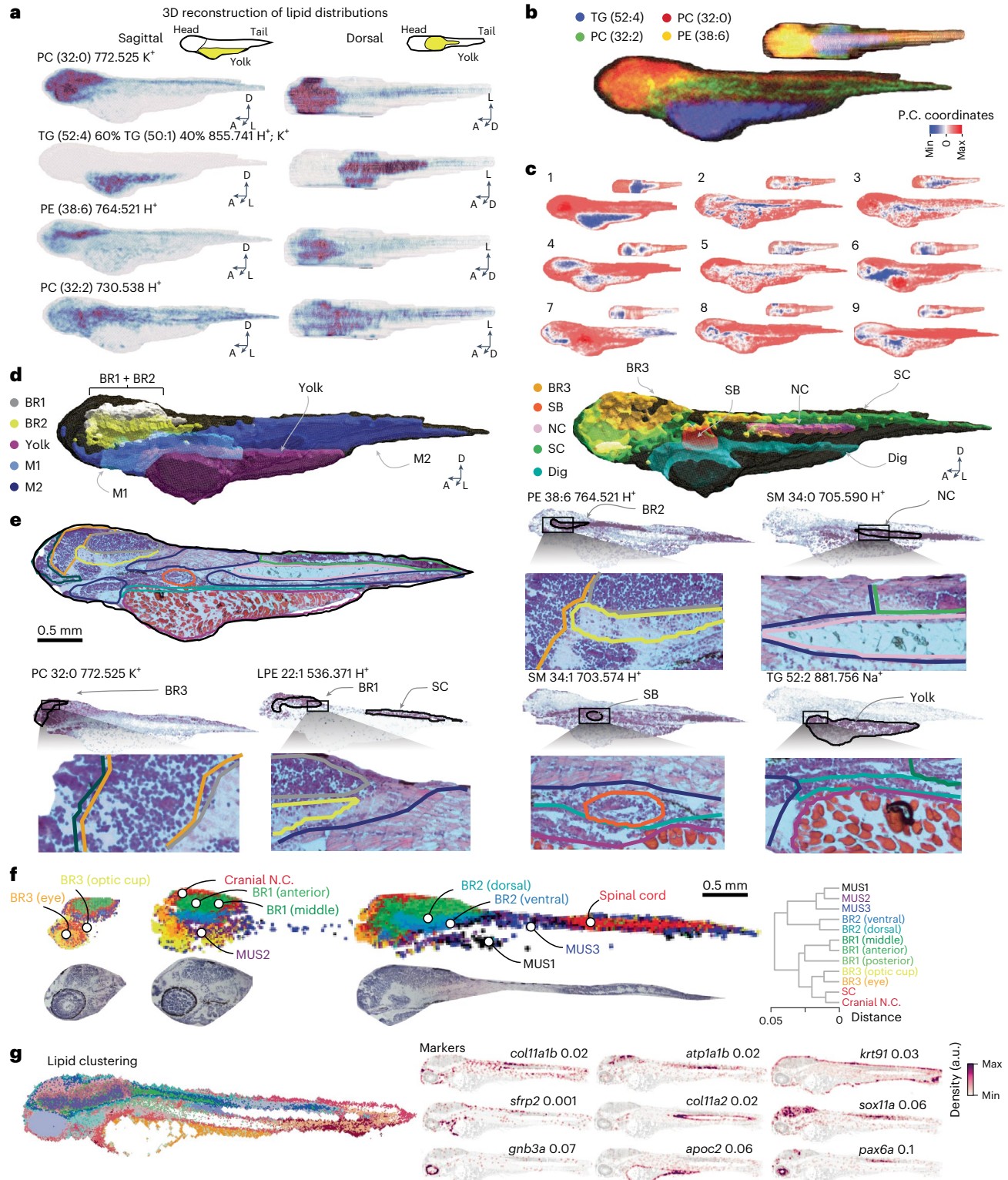

**Fig. 5 | Characterization of metabolically defined tissues in the embryonic zebrafish at 72 hpf. a**, Three-dimensional reconstructions for four lipids with *m/z* value and lipid indicated (sagittal and dorsal views). Schematic indicating yolk, head and tail displayed above. **b**, Overlay of the four lipids visualized in **a**. **c**, Sagittal and dorsal views of the first nine principal components retrieved after selecting for annotated lipids. BR1, BR2, BR3, brain regions; dig, digestive system; NC, notochord; SB, swim bladder; SC, spinal cord. **d**, Visualization of metabolically defined tissue clusters after application of the unsupervised clustering algorithm (sagittal view). M1, M2, musculature. **e**, Top row: medial section (sagittal view) of H&E staining overlaid with contours of metabolic

regions shown in **d**. Lipids that spatially localize to specific regions are indicated along with an inset of the H&E image. Both images are overlaid with contours representing delineated clusters. **f**, Left: subclustering of neural regions (top) with corresponding H&E sections (bottom). Right: dendrogram displaying relative distances between clusters. N.C., neural crest; MUS, musculature. **g**, Lipid clustering of a sagittal section at 72 hpf (left) and marker gene distributions from a consecutive section from HybISS (right). The importance of a gene in predicting lipid territories is indicated beside the gene name as a score. Scores are computed from the impurity decrease within each tree of a decision tree classifier and range from 0 to a maximum of 0.1.

of embryonic lipidic states (Supplementary Fig. 7e). Clustering voxels into discrete categories (Methods) resulted in lipid territories, which, after comparison with hematoxylin and eosin (H&E) staining, identified several organs, including the notochord, the swim bladder, the spinal cord and different brain regions (Fig. 5d,e). The correspondence between lipid territories and clusters was consistent across different clustering methods (Supplementary Fig. 7f–h). To explore finer subdivisions within lipid territories, we focused on the nervous system, rich in lipid diversity (Extended Data Fig. 8b), and reclustered. Interestingly, this revealed subregions corresponding to anatomical tissues including the eye and the optic cup that differed primarily in LPC, PC and PG concentrations (Fig. 5f and Supplementary Fig. 7i).

We further tested lipid territory correspondence with anatomical structures by performing hybridization-based in situ sequencing (HybISS)[52,53] on a sagittal section of the zebrafish at 72 hpf and MALDI-MSI on a consecutive section. A panel of tissue markers was used to highlight anatomy (Methods), while lipid-based pixel clustering was performed on MALDI-MSI data to retrieve lipid territories[54–56]. Several tissue markers faithfully localized to lipid territories, indicating that lipid distribution follows anatomy and that anatomy, in turn, can be inferred from lipid patterns (Fig. 5g).

Prompted by the tight correspondence with tissue–marker gene expression, we investigated whether lipid territories could be derived from metabolic gene expression. Analyzing publicly available single-cell expression profiles, we found that metabolism-related genes matched poorly with annotated cell identity due to their overall low expression. Specifically, lipid metabolic genes (for example, enzymes and transporters) were significantly less capable of predicting cell identity ($R^2 = 0.65$; bootstrap estimate) than the most variable genes ($R^2 = 0.78$) and performed similarly to random gene sets ($R^2 = 0.60$)[57] (Extended Data Fig. 10a,b and Methods). Therefore, gene expression data may not capture key predictors of metabolic activity.

To assess whether metabolic genes, although weakly expressed, could still localize to anatomical features, we expanded our HybISS approach to include highly variable metabolic genes (Methods). Except for a few cases, metabolic genes were typically noisier than marker genes and less capable of identifying lipid territories than MALDI-MSI data (Extended Data Fig. 10c–e). Only three of the ten most region-predicting genes were metabolic: *apobb.1*, *apoeb* and *fabp1b.1* (Fig. 5g, Extended Data Fig. 10f and Methods).

Next, we biochemically characterized anatomical regions by performing an enrichment analysis of lipids across the delineated lipid territories and identified the one where each species was maximally abundant (Fig. 6a). Stratifying these assignments by tissue revealed that several regions, including the swim bladder, the digestive system and brain, were predominantly characterized by a few lipid classes. This indicates that lipid species from the same class tend to localize within identical anatomical structures (Extended Data Fig. 10g).

By combining spatial and bulk lipidomics data, we obtained relative concentrations between lipid classes. As expected, PCs constitute a substantial proportion of lipids across clusters, while other classes showed more variability. For example, a divergent trend between PE and PS concentrations was evident (Fig. 6b) and may be ascribed to expression levels of *pisd*, which encodes a decarboxylase that converts PS to PE[58]. Accordingly, in the nervous systems, where *pisd* localizes[57],

PE concentrations were higher and PS concentrations were lower than in other regions (Fig. 6b and Extended Data Fig. 10h).

Similarly, a biochemical organization by lipid class emerged when lipid–lipid relationships were visualized in a connectivity graph based on spatial covariation (Fig. 6c). Diacylglycerols and lysolipids populated distinct nervous system regions, TGs were distributed between two clusters, SMs colocalized to the swim bladder, spinal cord and pericardial region, and PCs localized to two major lipid territories in the head and posterior tissues.

Interestingly, TG localization was generally dictated by FA carbon chain lengths and unsaturation levels. Shorter, less unsaturated TGs (≤54 C; less than four double bonds) accumulated in the yolk as energy reserves[20]. Surprisingly, long-chain, highly unsaturated TGs populated anterior anatomical regions distinct from the nervous system (Fig. 6c,d). In zebrafish embryos, neutral lipids are restricted to the yolk until 24 hpf, appearing in the head only at later stages[20]. Our bulk lipidomics data and spatial atlas consistently suggest that extra-yolk TGs are synthesized during development (Supplementary Fig. 6d,e) and contain longer (>20 C), more unsaturated FAs (Fig. 6d). Our atlas shows that, in the head, TGs localized to a mesenchymal compartment related to cartilage and bone primordia (Fig. 6e), aligning with evidence of extensive fat depots within bones[59].

SM localization to the swim bladder was also unexpected. This organ, homolog of the tetrapod lung[60–62], begins forming at 48 hpf and inflates around 96 hpf to regulate buoyancy. As in lungs, inflation requires lipid-laden surfactant secretion in the organ cavity to lower surface tension[61,62]. SMs began accumulating in the prospective swim bladder at 48 hpf, forming a distinguishable cluster by 72 hpf (ref. 63; Fig. 5d). Inspecting SM species distribution in coronal sections from embryos at 72 hpf, we found that they populate concentric layers of the swim bladder, possibly reflecting the organ's known tissue structure[63] (Fig. 6f).

Intrigued by their peculiar distribution, we assessed sphingolipid function in development by perturbing its biosynthesis. To this aim, we injected morpholinos into one-cell-stage embryos (Methods) targeting *sptlc1*, which encodes an enzyme that catalyzes the rate-limiting step in sphingolipid biosynthesis[64]. At 120 hpf, *sptlc1* morphants displayed developmental defects including tail curling, pericardial edema (Extended Data Fig. 10i) and significantly impaired swim bladder inflation (Extended Data Fig. 10j). Notably, *sptlc1*-knockout lines were previously reported to exhibit faulty swim bladder inflation[21]. Finally, Oil Red O staining[65] highlighted lower surfactant levels in *sptlc1* morphants, without histological alteration (Fig. 6g and Extended Data Fig. 10k). These data indicate that sphingolipids accumulate in the forming swim bladder and are involved in surfactant production and inflation during zebrafish embryogenesis.

Altogether, this evidence reveals that, during embryogenesis, specific metabolic programs are established within different structures, shaping their formation and function.

## Discussion

During embryogenesis, signals and spatially defined differentiation trajectories inform identical precursor cells to generate anatomical structures with distinct biochemical compositions. While morphogenic programs have been described well through the lens of transcriptomics,

**Fig. 6 | Biochemical organization of metabolically defined tissues in the embryonic zebrafish at 72 hpf. a**, Enrichment heatmap of lipids within clusters shown in Fig. 5d. **b**, Mol% composition of metabolite-defined clusters stratified by lipid class for 3D fish (*n* = 1). Pie plots from left to right show three lipids of increasing concentrations (lowest, median and maximum) and display localization according to mol%. **c**, Force-directed layout network representation of lipid colocalization. Lipid class is indicated with a color code. Images beside node clusters display representative lipid distribution of node clusters. **d**, Subset of TG species images (left) with FA compositions (right). Blue and red boxes

indicate FAs with low and high carbon content and unsaturations, respectively. **e**, H&E (top left), brightfield (bottom left), genes from HybISS (top right) and lipids from MALDI-MSI images overlaid (bottom right) from a coronal section through the head of a zebrafish at 72 hpf (*n* = 1). Scale bars, 100 μm. **f**, H&E (left), brightfield (middle) and lipids from MALDI-MSI images overlaid (right) from a coronal section through the swim bladder of a zebrafish at 72 hpf (*n* = 1). Scale bars, 100 μm. **g**, Representative brightfield (BF) and Oil Red O (ORO) staining of control and *sptlc1*-mutant fish with a magnified view of the swim bladder in the right column. Scale bars, 1 mm. MO, morpholino.

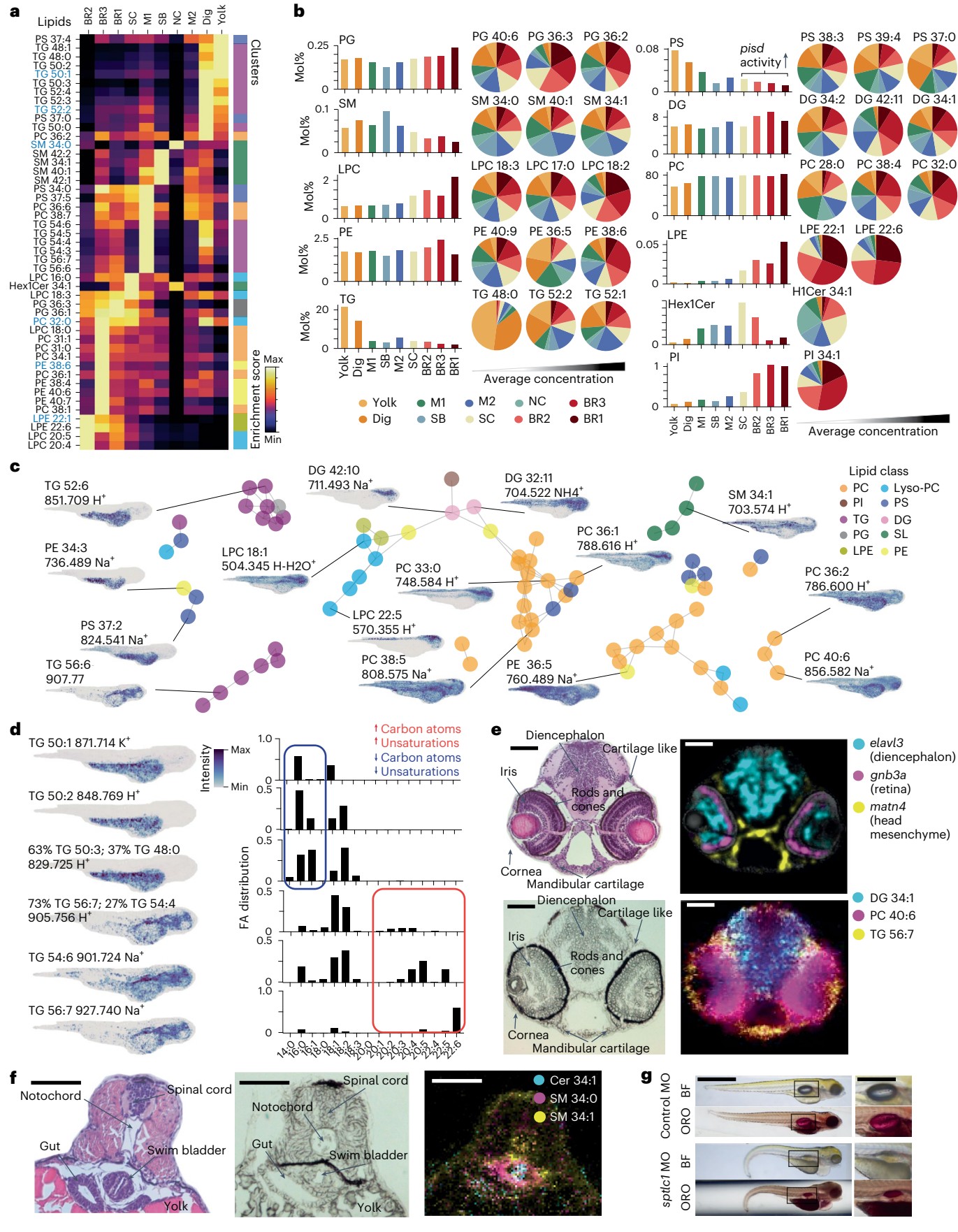

understanding how metabolism evolves spatiotemporally in the context of embryogenesis is essential to comprehend molecular processes that guide tissue patterning.

Here, we generated a 4D lipid metabolic atlas of vertebrate development to characterize the emergence of biochemical territories. To achieve this goal, we acquired a comprehensive dataset of high-resolution MS images from zebrafish embryos at different developmental stages. We also developed a general and accessible computational framework, uMAIA, which rethinks metabolomic MSI data processing toward the integration of multiple acquisitions. This advancement unlocks new possibilities for the construction of whole-organism metabolomic atlases while facilitating unbiased analyses and ensuring reliable cross-acquisition comparisons.

Analysis of metabolomic MSI data presents technical challenges that become relevant when analyzing multiple datasets. While analyses of large datasets have been pursued, for example, in relation to the study of the proteome and organ variability[28,29], batch effects and intensity variations across acquisitions have limited the scope of spatial metabolomic and lipidomic investigations. Previous studies adapted to these constraints by applying supervised machine learning approaches[28,30] or by restricting analyses to major tissue compartments[29], as finer-scale analysis might have been confounded by technical variations. uMAIA addresses technical challenges by providing an integrated pipeline that enables both reliable cross-acquisition comparisons and unbiased detection of biochemical territories, expanding the analytical capabilities of large-scale metabolomic and lipidomic MSI studies. Furthermore, our initial tests also suggested applicability of uMAIA to other MSI datasets, including proteomics. Different acquisitions are made interoperable with conservative procedures and minimal assumptions: uMAIA creates normalized and coherent feature sets, has no black box components and does not impute values or create dependencies between features. Unlike deep learning integration methods, uMAIA is a principled and conservative normalization method that preserves biological variation, while it may retain extreme batch effects. This advancement in data handling facilitates the study of spatial metabolomics with other modalities to reveal new interactions and regulatory axes within tissues and cells, as was recently shown for lipotypes[15].

In this work, we present a complete multivariate analysis of the MSI atlas we generated. Our results reveal a remarkable correspondence between lipid distribution and anatomy, reflecting how metabolic programs are organized based on cell type compositions of different tissues. Importantly, transcriptomic methods showed suboptimal sensitivity to capture the spatial distributions of key metabolic genes. This indicates a necessity to quantify lipids directly to address the lipidome's spatial organization.

When examining spatiotemporal lipid level changes over development, we encountered unexpected findings. This included PC species spatially segregating based on unsaturation degrees. As key membrane lipids with distinct biophysical properties[66–68], their varying tissue distribution likely affects membrane behavior. Determining how this pattern emerges mechanistically and understanding its physiological relevance are promising goals for future studies.

More generally, the colocalization of lipid classes to lipid territories suggested that territories are defined by metabolic pathway activity and that they are possibly involved in specific tissue functions. Examples include sphingolipid accumulation in the swim bladder, where they might function as surfactants, as well as long-chain TG species in bone primordia, where they may play roles in osteogenesis in the zebrafish head. We expect discoveries of this kind to help link diseases with potential therapeutic targets, as exemplified by the disruption in swim bladder function observed upon perturbation of sphingolipid biosynthesis.

Global assessment of lipid distributions during embryo development revealed a close parallel between the unfolding of lipid territories

and organogenesis. Embryonic lipid territories did not specify simultaneously over time; the anterior regions, particularly the nervous system, exhibited the highest variation in metabolic composition and were the earliest to undergo fine metabolic patterning. Investigating whether this rostral lipidomic specification is conserved or more pronounced in species with finely regionalized central nervous systems, such as mammals, holds promise for future research. More generally, multispecies atlases will enhance our understanding of whether lipid patterning coevolved with brain expansion and increased complexity.

A limitation of our study is the inability to discern lipid transport from in situ de novo synthesis during embryogenesis. Additional experiments at higher temporal resolution and including metabolic labeling of newly synthesized lipids in combination with spatial transcriptomics would be necessary to address metabolic fluxes in space. Furthermore, due to ion fragmentation and sample quality differences, MSI is limited to capturing a fraction of molecular species that may alter abundance estimates.

In conclusion, by developing uMAIA, we have made the unbiased analyses of large, organismal-level MSI datasets tractable. Exploiting these capabilities, we conducted a whole-embryo exploration of an important compositional aspect of living systems: their lipidome. Our results reveal substantial heterogeneity along this underexplored axis, highlighting it as a compelling ground for discoveries in the regulation of development.

## Online content

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

## Methods

### Data generation

**Animal work and cryosectioning.** Zebrafish embryos of the Tübingen long fin strain were kept at 28 °C until collection at 8, 24, 48 and 72 hpf. Following manual dechorionation, embryos were euthanized by fixation in 4% PFA at 4 °C overnight. After rinsing with 1× PBS, embryos were included in a block of 10% pork skin gelatin (type A, Sigma), which was allowed to solidify at 4 °C for 1 h, and then immediately frozen in isopentane prechilled to −65 °C. Specimens were kept at −80 °C until cryosectioning over the sagittal axis at −28 °C for a sectional thickness of 12 μm. Sections were stored at −80 °C until MALDI-MSI acquisition. All zebrafish husbandry procedures have been approved and accredited by the Federal Food Safety and Veterinary Office of the Canton of Vaud (VD-H23).

**Matrix-assisted laser desorption–ionization mass spectrometry imaging datasets.** Publicly available MALDI-MSI datasets were retrieved from METASPACE. Human kidney samples were generated by the NIH Kidney Precision Medicine Project (ID 2024-06-06_19h26m13s, 2024-06-06_19h22m09s, 2024-06-06_19h16m21s, 2024-06-06_19h08m24s, 2024-06-06_19h03m38s, 2024-06-06_18h57m58s, 2024-06-06_18h45m39s, 2024-06-06_18h40m58s), and baboon lung samples were generated by the Pacific Northwest National Laboratory (ID 2024-06-11_18h30m14s, 2024-06-11_18h28m14s, 2024-06-11_18h35m41s, 2024-06-11_18h26m59s, 2024-06-11_18h43m25s). The datasets were chosen such that the acquisitions represented similar samples, allowing us to assume that molecules should be present with similar intensities. From each section, images were retrieved on the METASPACE interface, and intensity ranges were set equal for all sections. Compounds that were not detected were marked as 'missing', and those with aberrations were highlighted.

We collected MALDI-MSI datasets for zebrafish embryos. Sections from alternating sagittal sections were dried at room temperature before coating with CHCA matrix prepared at 15 mg μl⁻¹ in a solution of acetone and water (1:1) and 0.1% trifluoroacetic acid for positive ionization. Matrix was deposited with a sprayer (Sprangler) at 350 rpm and a flow rate of 5 μl min⁻¹ for 40 min. MALDI-MSI acquisitions were acquired using an AP-SMALDI[5] AF system coupled to a Q Exactive orbital-trapping mass spectrometer in full MS scan and positive ion mode in the 400–1,200-$m/z$ range. The MALDI laser focus was optimized manually using source cameras aiming at a diameter of the focused beam of <5 μm. For each pixel, the spectrum was accumulated from 50 laser shots at 100 Hz. MS parameters in Tune software (Thermo Fisher Scientific) were set to a spray voltage of 4 kV, S-Lens at 100 eV and capillary temperature to 250 °C. Calibration was performed regularly with ISs (that is, known matrix ions), and mass error was kept within ±2 ppm. A pixelated scan mode was used at a speed of 1.6 pixels per second. The spatial resolution for acquisitions of embryos at 24, 48 and 72 hpf was 7 μm. For the embryo at 8 hpf, a resolution of 5 μm was used to account for its smaller size. In addition to these sections used for atlas construction, we acquired sections from a biological replicate for each time point to check the consistency of results. A total of 96 MS images were acquired.

**Hybridization-based in situ sequencing.** The protocol was followed according to ref. [52]. Zebrafish larvae were euthanized by fixation in 4% PFA at 4 °C overnight, embedded in 2% carboxymethyl cellulose and cryosectioned at 10 μm. Sections were stored at −80 °C and processed. For transcript detection, sections were incubated with padlock probes (concentration of 10–50 nM per probe) for marker and metabolic genes in the same mix. Samples were imaged on a Leica DMi8 epifluorescence microscope equipped with an LED light source (Lumencor, SPECTRA X, nIR, 90-10172), an sCMOS camera (Leica, DFC9000 GTC, 11547007) and a ×20 objective (HC PC APO, NA 0.8, air), with 10% overlap between tiles and 8–12 z-stack planes with 1-μm spacing.

For image processing, z projection was carried out using a variant of wavelet-based extended depth of field[70]. Tiles were stitched with ASHLAR[71] using the DAPI channel of the acquisition. Intercycle registrations were obtained using the wsireg Python package[72], which wraps elastix[73]. Spot detection was performed with Spotiflow[74] using the pretrained HybISS model with default settings. Detections were decoded using a variant of starfish's[53] nearest-neighbor decoder.

**Bulk lipidomic experiments.** Zebrafish embryos were collected at the 8-, 24-, 48- and 72-hpf stages and anesthetized with tricaine. After manual dechorionation, 15 individuals per replicate were pooled together and immediately frozen in liquid nitrogen. For each developmental stage, four replicates were analyzed by the high-throughput targeted HILIC–MS/MS lipidomic workflow at the Université de Lausanne. Complex lipids were quantified using a high-coverage, stable isotope dilution LC–MS/MS approach. Zebrafish tissue (15 embryos) was extracted with isopropanol[48] prespiked with the IS mixture containing 75 isotopically labeled lipid species (with multiple representatives per lipid class, with varying FA composition). The resulting extract was analyzed by the HILIC–ESI-MS/MS approach in positive ([M + H]⁺ and [M + NH₄]⁺ adducts) and in negative ([M + AcO]⁻, [M − 2H]⁻ and [M − H]⁻ adducts) ionization mode, using a TSQ Altis LC–MS/MS system (Thermo Scientific), as previously described by Medina et al.[48]. In two separate runs (using a dual-column setup), 1,166 lipid species belonging to five major classes of complex lipids (glycerolipids, glycerophospholipids, cholesterol esters, sphingolipids and free FAs) were quantified with high precision and specificity. Using HILIC separation, endogenous lipids coelute with their corresponding IS, thus allowing for the appropriate correction of matrix effects (in the same solvent composition)[49,75]. Optimized lipid class-dependent parameters were used for data acquisition in timed selected reaction monitoring mode. Raw LC–MS/MS data were processed using TraceFinder Software (version 4.1, Thermo Fisher Scientific). Lipid abundance was reported as an estimated concentration using a mixture of ISs spiked at known concentrations and designed to correct for the differences in ionization and fragmentation efficiency depending on FA chain length and degree of unsaturation[76].

**Morpholino injection and Oil Red O staining.** A translation-blocking morpholino (sptlc1-ATG MO, 5′-ACCCACTGTTGCCCCGACGCCATTT[21]) (Gene Tools) was used to knock down *sptlc1* expression. Morpholinos were diluted to 0.1 mM. A bolus of 100 μm was injected in the cell of one-cell-stage embryos of the AB zebrafish strain. As a control, a mixture of random sequences was injected. Injected embryos were raised in an incubator at 28 °C with the addition of 1-phenyl-2-thiourea to inhibit melanin production to better visualize the swim bladder. The phenotype was assessed at 120 hpf. Non-injected siblings of the same clutch were also grown in the same conditions to control for any additional defects not related to the injections. At 120 hpf, about eight representative larvae of each condition were placed in depression wells and photographed. Larvae were immediately fixed in 4% PFA at 4 °C for 5 h. Fixed zebrafish were then rinsed with PBS, incubated in methanol at −20 °C for 24 h, briefly equilibrated in 60% isopropanol and stained with a solution of 0.3% Oil Red O (Sigma-Aldrich, O0625) in 60% isopropanol for 3 h as previously described[20]. Samples were then rinsed with 60% isopropanol. After a final wash with PBS, organisms were placed in depression wells and photographed.

### Models and computational approaches

The uMAIA framework comprises three main algorithms designed to achieve the following tasks:

- Peak calling for image generation from raw mass spectra
- Peak matching for the creation of a unified feature space
- Intensity normalization for batch effect removal.

**Peak calling: processing of single mass spectrometry imaging acquisitions.** The first algorithm processes raw mass spectra within single MSI acquisitions to call peaks and construct images.

*Peak calling of mass spectrometry imaging spectra.* Raw MSI spectra are stored as collections of tuples: detected $m/z$ values with corresponding signal intensity and coordinates ($m_z$, intensity, $x$, $y$). We round $m/z$ values to the nominal mass resolution of the instrument to obtain $N$ values (bins of size $10 \times 10^{-5}$ were used). The data are loaded as a sparse matrix $S_{p,m}$ containing the signal intensity values recorded at each pixel $p \in (0, P)$ and position in the mass spectrum indices with $m \in (0, N)$. Next, a histogram representation of the data, $\mathbf{f}_m$, is obtained by counting nonzero entries across pixels, that is, $\mathbf{f}_m = \sum_p (S_{p,m} > 0)$. A Gaussian kernel of small bandwidth ($10 \times 10^{-4}$, approximately reflecting the mass accuracy of the instrument) is applied to $\mathbf{f}_m$ to reduce the impact of noise.

Next, the following two-step (algorithms 1 and 2) peak-calling procedure, inspired by the watershed algorithm, is applied to the histogram representation of the data to retrieve the set of peaks $\mathcal{M}$ and their boundaries $\mathcal{B}$. Algorithm 1 identifies seeding points based on histogram representation peak maxima, whereas algorithm 2 retrieves the bin sizes for imaging.

**Image retrieval and further processing.** To extract images from raw data, we aggregate $S_{p,m}$ according to the set of boundaries $\mathcal{B}$ resulting from algorithms 1 and 2. For each pixel $p$ and each compound $c$, we have $X_{p,c} = \sum_{m \in \mathcal{B}_c} S_{pm}$. Images are normalized for total ion content by dividing each pixel by the total signal detected: $X_{pc} \leftarrow X_{pc}/\sum_c X_{pc}$.

Outputs of the peak-calling method are (1) images saved in a .h5ad format and (2) a .csv file specifying the intervals that were selected among other metadata.

**Algorithm 1** Retrieval of $\mathcal{M}$

| | |
|---|---|
| 1. **Procedure** InitializeBins ($f_m$) | ▷ Retrieval of unique $m/z$ representing individual molecules |
| 2. $m \leftarrow \arg\max_m f_m$ | ▷ Initialize the first $m/z$ detection |
| 3. $\mathcal{M} \leftarrow \mathcal{M} \cup \{m\}$ | ▷ Add $m$ to set |
| 4. $t \leftarrow \max_m f_m$ | ▷ Initialize threshold to begin iterations |
| 5. **While** $t \neq 0$ **do** | |
| 6. $\quad t \leftarrow t - 1$ | ▷ Continue reducing the threshold |
| 7. $\quad \mathcal{D} \leftarrow \{d | \forall i \in \mathcal{M} : f_i > t\}$ | ▷ Retrieve contiguous intervals above threshold |
| 8. $\quad$ **For all** $d \in \mathcal{D}$ **do** | ▷ Iterate over contiguous sequences |
| 9. $\quad\quad$ **If** $d \cap \mathcal{M} = \varnothing$, **then** | |
| 10. $\quad\quad\quad m \leftarrow c_{\text{center}}$ | ▷ Calculate the center of $d$ |
| 11. $\quad\quad\quad \mathcal{M} \leftarrow \mathcal{M} \cup \{m\}$ | ▷ New molecule identified |
| 12. **Return** $\mathcal{M}$ | |

**Algorithm 2** Determine the intervals $\mathcal{B}$

| | |
|---|---|
| 1. **Procedure** ExpandBins ($\mathcal{M}$, $\mathbf{f}$) | ▷ Expand bins beginning with $\mathcal{M}$ |
| 2. **For all** $m \in \mathcal{M}$ **do** | ▷ Iterate over $m/z$ of unique molecules |
| 3. $\quad l \leftarrow m - 1$ | |
| 4. $\quad r \leftarrow m + 1$ | |
| 5. $\quad$ **While** $\mathbf{f}[l] \geq \mathbf{f}[l-1]$ **do** | ▷ Expand the left boundary of the bin |
| 6. $\quad\quad l \leftarrow l - 1$ | |
| 7. $\quad$ **While** $\mathbf{f}[r] \geq \mathbf{f}[r+1]$ **do** | ▷ Expand the right boundary of the bin |
| 8. $\quad\quad r \leftarrow r + 1$ | |
| 9. $\quad \mathcal{B} \leftarrow \mathcal{B} \cup \{[l, r]\}$ | ▷ Store bin range |
| 10. **Return** $\mathcal{B}$ | |

**Network flow-based peak matching: optimized matching of peaks into a unified feature set.** MSI datasets are not directly produced in the same feature space, differing in the number of identified peaks and their precise $m/z$ positions. uMAIA automatically creates a unified feature space (featurization): a peak from each acquisition is connected with one from another acquisition if they are considered the same molecule, without requiring external references.

Connected peaks should satisfy a set of properties and constraints: closer peaks should be matched, it is better to match a peak than to leave it unconnected, and each peak can be connected only to another from a different acquisition (as they are the same molecule). These properties are not trivially achievable with a heuristic or procedural approach (for example, nearest-neighbor graph). We aim to identify the choice of connections that respects all properties and constraints.

To formalize the problem, we consider the following quantities:

- Each of the $n$ acquisitions provides a set of $m/z$ peaks $\mathcal{A}_1, \mathcal{A}_2, \dots, \mathcal{A}_n$. For simplicity, we refer to $\mathcal{A}$ as the union of all sets, with $|\mathcal{A}| = L$.
- Considering each element in $\mathcal{A}$ as a node in a graph $\mathcal{G}$, we define $G_{ij}$, the adjacency matrix indicating the possible connections that are allowed. This is defined using a distance threshold $t$ so that $G_{i,j} = 1$ if dist$(i,j) < t$, forcing $G_{ij} = 0$ if $i$ and $j$ are from the same acquisition.
- The desired matching uMAIA achieves is represented by a graph $\mathcal{X}$ with an adjacency matrix $X_{ij}$. $X_{ij} = 1$ when peaks from two acquisitions are considered to have originated from the same ion or compound. Each connected component of this graph is then considered a feature and used to construct the final feature space.
- $C_{ij}$ indicates the associated cost of a particular matching $X_{ij} = 1$. The matrix is set by the distance in $m/z$ between two peaks.

We aim to find the optimal $\hat{X}_{ij} = \arg\min \sum_{i=1}^m \sum_{j=1}^m X_{ij} C_{ij}$ while obeying a set of constraints. In its general form, the problem can be categorized as an integer linear problem (ILP), which is only tractable for small $L$ values (while we have a large graph). However, two important simplifications we make here render it tractable in polynomial time. First, we divide the mass spectra into $m/z$ intervals spanning one unit, allowing us to solve mz_range smaller subproblems. Second, we do not

consider all the possible links $G_{ij}$ between any acquisitions but allow only peaks between some pairs of acquisitions. Specifically, we considered the acquisitions ordered and allowed edges between the peaks of an acquisition and the $k$ consecutive ones (for all analyses, $k = 2$). Also, we introduce a start and end nodes in the graphs that can be connected with high-cost log ($L$) but are required by the constraint to complete a path. Because the solution of each subproblem can be found extremely fast using LP solvers, we solve it $M$ times for $M$ orders of the acquisitions obtained by random permutation and choose the best solution for each interval. The problem we solve can be formulated as follows:

$$\min_{X} \quad \sum_{i=1}^{L}\sum_{j=1}^{L} X_{ij}C_{ij}$$
$$\text{subject to} \quad X_{ij} \in \{0,1\}$$
$$X_{ij} \leq G_{ij} \tag{1}$$
$$\sum_j X_{ij} = \sum_i X_{ij} = 1 \quad \forall i,j \notin \{\text{start,end}\}.$$

After solving the problem, features are retrieved by traversing all paths beginning from the start node until the end node is encountered.

**Normalization: intensity correction for mass spectrometry imaging datasets.** MSI experiments are prone to fluctuations in the detected signal intensity due to experimental factors, potentially distorting the intensity histograms of different acquisitions.

Pixel intensity histograms from MSI data are typically bimodal, consisting of background and foreground signal components. Not all images contain both components. Our framework models batch effects in the foreground signals, assuming background signals are relatively stable.

Our approach to correct batch effects follows these steps:

- Estimation of the batch effect using a hierarchical Bayesian model
- Use of the estimation to determine signal intensity transformations for each image
- Application of functions for data normalization.

*Theory, model and implementation of normalization.* We consider the complete set of pixels $p$ measured across all acquisitions $a$ so that $p \in \bigcup_a s_a$, and denote $y_{cp}$ the ground truth value associated with each compound $c$ measured. Our goal is to correct batch effects and recover ground truth values despite distortions $T_{ca}$ induced by the measurement process for each acquisition $a$ and compound $c$ where we measure

$$x_{cp} = T_{ca}(y_{cp}) \tag{2}$$

with $p \in s_a$. To compare images from different acquisitions, we aim to estimate $T_{ca}^{-1}$ so that we can perform the transformation to correct the data.

$$\hat{y}_{cp} = \hat{T}_{ca}^{-1}(x_{cp}) \tag{3}$$

We propose to estimate $T_{ca}^{-1}$ from the intensity value histograms observed across acquisitions and compounds. We start by defining the 'reference' intensity distribution across pixels with a probability density function $f_c(y_{cp})$ or, equivalently, $f_c(y)$. The transformation $x_{cp} = T_{ca}(y_{cp})$ induces a new probability distribution $g_{ca}(x)$. The relation between $g$, $f$ and $T$ is simplified if we consider the cumulative density functions $G_{ca}$ and $F_c$:

$$G_{ca} = F_c(T_{ca}^{-1}(x)), \tag{4}$$

which provides a way to estimate the desired normalizing transform $\hat{T}_{ca}^{-1} = \hat{F}_c^{-1}(\hat{G}_{ca}(x))$. Because $\hat{F}_c$ does not depend on the acquisition, it can be chosen with some freedom, as all the normalization is performed by applying $\hat{G}_{ca}$ and $F$. We focus thus on estimating $\hat{G}_{ca}$ or, equivalently, $\hat{g}_{ca}$.

Generally after parameterizing $g(x; \hat{\theta}_{ca})$, one would estimate the parameters $\hat{\theta}_{ca}$ by maximum likelihood:

$$\hat{\theta}_{ca} = \arg\max_{\theta_{ca}} \prod_{p \in s_a} g(x_{pc} | \theta_{ca}). \tag{5}$$

What complicates this endeavor is that the measured $x_{pc}$ coming from a single acquisition cannot be expected to be representative of the complete set of pixels $\bigcup_a s_a$: foreground or background components may or may not be present within a given acquisition and biological variability would likely not be retained. Histogram matching therefore is not suitable for the problem: we need to combine information across acquisitions and compounds to obtain a robust $\hat{T}_{ca}^{-1}$.

To achieve this, we first consider the parameterization of $g$ as a mixture of two Gaussians:

$$g(x_{pc}|\theta_{ca}) = g(x_{pc}|\mu_{ca}^0, \mu_{ca}^1, \sigma_{ca}^0, \sigma_{ca}^1, \rho_{ca})$$
$$= \rho_{ca} \frac{1}{\sqrt{2\pi(\sigma_{ca}^0)^2}} \exp\left(-\frac{(x_{pc}-\mu_{ca}^0)^2}{2(\sigma_{ca}^0)^2}\right) \tag{6}$$
$$+(1-\rho_{ca}) \frac{1}{\sqrt{2\pi(\sigma_{ca}^1)^2}} \exp\left(-\frac{(x_{pc}-\mu_{ca}^1)^2}{2(\sigma_{ca}^1)^2}\right).$$

We assume that parameters of the mixture can be factorized as to consider batch effect derived from acquisition- and compound-specific factors, given by $\gamma_a$ and $\lambda_c$, respectively. Specifically, we choose $\mu_{ac}^1 = \mu_{ac}^0 + \gamma_a\lambda_c + \delta_c$, where $\gamma_a \in (-2, 2)$ $\lambda_c \in (-2, 2)$ and a prior for $\delta_c$ normal(3, 1) represents the expected difference between modes. The prior for the background mode is $\mu_{ac}^0 \sim \text{normal}(m_0, 1)$, with $m_0$ chosen empirically by considering the distribution of $\sigma^1$ of a set of naive two-component GMM fits. Similarly, $\sigma_{ac}^1 = \Sigma_a + \Sigma_c$. The following priors $\Sigma_a, \Sigma_c$ - exponential($s$), where $s$ was chosen empirically by considering the distribution of $\sigma^1$ of the full set of naive two-component GMM fits.

This corresponds to the following model:

$$\delta_c \sim \text{normal}(3, 1)$$
$$\lambda_c \sim \text{uniform}(-2, 2)$$
$$\gamma_a \sim \text{uniform}(-2, 2)$$
$$\Sigma_a \sim \text{exponential}(s)$$
$$\Sigma_c \sim \text{exponential}(s)$$
$$\sigma_c^0 \sim \text{uniform}(0.05, 0.5)$$
$$\mu_{ac}^0 \sim \text{normal}(m_0, 1)$$
$$\mu_{ac}^1 = \mu_{ac}^0 + \gamma_a\lambda_c + \delta_c$$
$$\sigma_{ac}^1 = \Sigma_a + \Sigma_c$$
$$\pi_{ca} \sim \text{Dirichlet}(0.5, 0.5)$$
$$z_{cp} \sim \text{Bernoulli}(\pi_{ca})$$
$$x_{pc} \sim \delta(z_{cp})\text{normal}(\mu_{ac}^1, \sigma_{ac}^1) + \delta(z_{cp}-1)\text{normal}(\mu_{ac}^0, \sigma_c^0) \quad for\ p \in s_a.$$

The model was implemented and fit using the NumPyro probabilistic programming language. The MAP estimates of the parameters $(x_{pc}, \mu_{ca}^0, \mu_{ca}^1, \sigma_{ca}^0, \sigma_{ca}^1)$ are determined and used to compute the transformation. $f$ was chosen as the bimodal Gaussian mixture model with the following parameters:

$$\mu_c^0 = \text{mean}_a(\mu_{ca}^0); \quad \sigma_c^0 = \text{mean}_a(\sigma_{ca}^0)$$
$$\mu_c^1 = \text{mean}_a(\mu_{ca}^1); \quad \sigma_c^1 = \text{mean}_a(\sigma_{ca}^1). \tag{7}$$

The CDF, $G_{ca}$ and the inverse CDF $F_{ca}^{-1}$ are evaluated numerically and used to compute the transformation $\hat{T}_{ca}^{-1}$.

## Simulations and method evaluation

**Analysis of single acquisitions and peak calling.** *Comparison with other methods.* Freely available software was used for benchmarking:

1. MALDIquant: R package for MSI processing and calibration (adaptive binning) (Gibb & Strimmer[38])
2. METASPACE: online platform for visualization and annotation (static binning and references *m/z* values)[77]
3. Mirion: software visualizing molecules in MSI data (static binning) (Paschke et al.[37])
4. MSiReader: software visualizing molecules in MSI data (static binning and reference *m/z* values).

Imaging molecules requires identifying a range of *m/z* values that encompasses mass shifts for peaks. Ranges identified by each method were retrieved and compared with those from the uMAIA method (algorithms 1 and 2).

*Evaluation of mass shifts within images.* The degree to which mass shifts are correlated between molecules that are almost isobaric was evaluated by manually establishing ranges for each molecule in one MSI acquisition, and a reference *m/z* (the most frequent *m/z* detection) was identified. Next, for each molecule (that is, *m/z* range), a mass shift for each spectrum was calculated by subtracting the observed *m/z* from the reference *m/z*. Mass shifts for different pairs of molecules were then scattered together for the subset of pixels that contained both molecules. Pearson's *R* was calculated for the same subset.

*Simulations for tested approaches.* We performed simulations that mimicked mass shifts from real data. First, a set of numbers representing theoretical *m/z* values was generated in a unit range. To create datasets of variable molecular crowding, we varied the size of the set of numbers. To model mass shifts, random values sampled from normal distributions ($\sigma = 0.01$) and $\gamma$ distributions ($k = 0.1$ and $\theta = 0.04$) (as approximated from real data) were added to the theoretical *m/z* values for each of the 1,000 simulated spectra in a way such that the order of the molecules after addition of noise still followed the original ordering of the molecules, preventing unreal inversions of *m/z* positions between spectra.

Next, noise 'spike ins' were added at random across the range with variable intensities to reflect different SNRs. An SNR of 10 in a spectrum with 100 detections would mean that there are ten times as many real signals as noise, resulting in 90 true signals and ten noise signals.

For each dataset that varied in its molecular crowding and SNR, ten instances were simulated. Bins were retrieved by static binning methods (sizes of 0.02 and 0.04), MALDIquant and uMAIA. Mutual information scores were calculated for each peak with its prediction, and a weighted average was reported.

*Image quality assessment of mass spectrometry imaging data.* For each interval returned by the methods (uMAIA, Mirion and MALDIquant), the fraction of generated images likely to contain multiple different molecules was estimated by counting images for which the signal was contributed by more than a peak per spectrum or pixel.

We aligned bins from different methods by considering the extent of overlap between two ranges. With this information, we calculated the following:

- The average intensity between two images referring to the same molecule from different methods
- The spatial correlation between the two images

- The number of uMAIA detections identified in individual MALDIquant ranges
- The number of MALDIquant detections identified in individual uMAIA ranges.

We considered the number of MALDIquant detections that were within a given uMAIA range and calculated the average mutual information for each molecule set.

Three metrics were used to quantify various characteristics of image quality.

1. Spatial chaos considers intensities in an image and partitions them into levels. By assuming images arising from noise have equal distributions of intensity at each level, a score is assigned based on how different the sizes of the intensity partitions are. Codes for the spatial chaos metric were taken from https://github.com/alexandrovteam/pySM (ref. 78).
2. The noise model metric was devised to distinguish images with spatial structure. A basis function consisting of small Gaussian-distributed points spanning the image was used, and a generalized linear model was fit to the image using a Poisson distribution to account for noise. A likelihood ratio test against a null hypothesis was used to assess fit.
3. The power spectrum metric was devised to quantify high-frequency aberrations across images. As MSI data are acquired by scanning, it is not uncommon to see aberrations that consist of stripes in the image. Images were converted into its Fourier space, and the area representing low- and high-frequency signals was summed, and their ratio was calculated.

A threshold was set for each metric based on manual inspection and kept the same for all analyzed datasets. These datasets included a DESI, FTICR and four samples from an AP-SMALDI instrument. The number of images that surpassed all thresholds for their respective metrics was calculated for each dataset and method.

**Peak matching.** *Simulation to assess matching.* To mimic mass shifts in real data, the same simulation was done as described above, varying the number of sections over ten realizations. The ILP optimization was run and benchmarked to a binning approach in which the size of the bin was equal to 2 s.d. of the specified noise distribution. Metrics including precision, accuracy and recall were calculated.

*Evaluation of peak matcher on real data.* The peak-calling algorithm was applied to 20 embryonic zebrafish sections. Theoretical *m/z* values were retrieved for molecules in each section, which were subjected to either our network flow-matching method or traditional binning methods. Four different bin sizes (0.001, 0.0025, 0.005, 0.01) were used to group together compounds across the acquisitions.

Detection accuracy assessment was performed in two ways. First, 50 abundant matrix compounds across acquisitions were considered. To assign an interval or peak to one of these compounds, we used the overlap with the theoretical *m/z* value as the criterion. For each individual acquisition, we considered a score of ambiguity computed by counting the number of observations falling inside the interval over the single one expected. Second, to score how consistently isotopolog pairs (M, M + 1) were present after featurizing with our approach versus standard binning, we considered the two binary vectors representing the N and N + 1 isotope presence across acquisitions, respectively, and computed both Jaccard and Euclidean distances between the vector pairs. This was performed only for the 50 most abundant compounds to avoid stochastic dropouts occurring closer to the detection threshold, confounding the analysis.

Finally, we considered all molecules present across the acquisitions. Four different bin sizes (0.001, 0.0025, 0.005, 0.01), including our ILP approach, were used to group together compounds across the sections.

**Batch effect characterization.** *Empirical estimation of the error matrix to study batch effects.* To verify that the normalization model is justified by the error structure of the data (that is, that fluctuations in the data are a combination of sample-dependent and feature-dependent factors), we consider a simplified empirical estimation framework to estimate the error. This empirical estimation exploits the fact that our acquisitions are sequential sections of the same structure.

The estimation is based on assumptions:

1. The observed background mode is equal to the true background mode $m_{ca}^0 = \mu_{ca}^0$.
2. As sections are consecutive, the high-frequency deviations in foreground mode shifts between consecutive sections are attributable to the batch effect, which we desire to remove. In other words, biological signals should vary smoothly across sections.

Following the two equations from the model described above:

$$\mu_{ac}^1 = \mu_{ac}^0 + \gamma_a \lambda_c + \delta_c, \tag{8}$$

$$x_{pc} \sim \delta(z_{cp})\text{normal}(\mu_{ac}^1, \sigma_{ac}^1) + \delta(z_{cp} - 1)\text{normal}(\mu_{ac}^0, \sigma_c^0), \quad \text{for } p \in \mathcal{S}_a. \tag{9}$$

Assuming a simple factorization of the noise, we write a reparameterization:

$$\mu_{ca}^1 = \mu_{ca}^0 + \lambda_c \gamma_a + \delta_c + \epsilon_c \lambda_c + \epsilon_a \gamma_a + \epsilon_c \epsilon_a$$
$$\epsilon_a \sim \text{normal}(0, \Sigma) \tag{10}$$
$$\epsilon_c \sim \text{normal}(0, \Sigma).$$

We factor out from this expression the real biological signal $y_{ca}$:

$$m_{ca}^1 = m_{ca}^0 + \lambda_c \gamma_a + \delta_c + \epsilon_c \lambda_c + \epsilon_a \gamma_a + \epsilon_c \epsilon_a + y_{ca}. \tag{11}$$

Considering this equation and the assumption above, empirically estimate the variables $\hat{\delta}_v$ and $\hat{y}_{i,v}$ :

$$\hat{\delta}_c = \text{mean}_a(m_{ca}^1 - m_{ca}^0)$$
$$\hat{y}_{ac} = \text{mean}_{a \in [i-2, i+2]}(m_{ca}^1) - \text{mean}_{a \in [0, i-2] \cup [i+2, A]}(m_{ca}^1).$$

We can then approximate the matrix $M_{ca}$, which quantifies batch effect shifts for each molecule and section.

$$M_{ca} = m_{ca}^1 - m_{ca}^0 - \hat{\delta}_c - \hat{y}_{ca} \tag{12}$$

$$M_{ca} = (\gamma_a + \epsilon_a)(\lambda_c + \epsilon_c) \tag{13}$$

Singular-value decomposition was applied to the matrix to factorize it into two components representing acquisition- and molecule-dependent effects, and the first eigenvalue from singular-value decomposition is reported. A bootstrapped estimate of the first eigenvalue of this matrix was calculated to retrieve its variance by sampling 80% of molecules for 50 repetitions.

*Simulation datasets for normalization method and evaluation.* Two simulated datasets were created to assess the performance of various normalization algorithms. The first dataset evaluated methods on a phantom object, while the second used distributions taken from ISH experiments (ABA).

The first dataset is an oversimplified structure constructed from three ellipsoids in a 3D space, with two of the three ellipsoids partially overlapped to facilitate result interpretation. Molecules were simulated to be present in one or more than one of the ellipsoids. When a molecule was assigned to an ellipsoid, pixel values in the region were drawn from normal distributions with a mean sampled from a normal distribution. In cases in which a molecule was present in ellipsoids that overlapped, the intensities were summed at the overlap, resulting in a third mode within intensity distributions. We simulated sectioning of the data by retrieving 2D images.

Each section was then perturbed with batch effects to mimic the effect seen from real data. Specifically, factors representing section- and molecule-specific noise were drawn from a normal distribution. Next, these parameters were used to derive and transform the intensity distributions from ground truth data to batch effect affected.

Another simulation was performed using murine brain ISH images from the ABA. Images were normalized by minimum–maximum rescaling to reflect the dynamic range present in MALDI-MSI data after logging the data ([−7, −1]). Next, batch effect noise was applied as above. Noise-perturbed data were corrected using $z$ normalization, ComBat and uMAIA, and quality of the correction was calculated via the root mean square error between corrected and ground truth data. Because treatment by the different normalization methods places the output data in different scales, a line was fit to the q–q plot between ground truth and corrected data. The slope of the line was used to place the two distributions in similar ranges so that intensity distributions within the image could be compared between methods.

*Ability to perform regional differential intensity testing.* ABA ISH data were used (refer to simulations in the preceding section). For each gene, a differential expression test was made for all pairs of five selected regions by considering the set of pixels belonging to each region. For each tested pair, a matrix representing TP and TN values was constructed (TP identified when $P < 0.05$). The process was repeated for perturbed data, and data were corrected using different methods ($z$ normalization, ComBat and uMAIA). FP, FN, TP and TN instances could then be ascertained when compared to ground truth. The ground truth FPR and FNR were estimated by bootstrapping sections belonging to the ground truth.

*Evaluation of algorithm on real data via principal-component analysis and clustering approaches.* We evaluated pixel clustering before and after normalization and compared the result with an L/S algorithm and an imputation–integration algorithm (ComBat, scArches). All molecules present in at least 15% of pixels were used as features for clustering. Clustering was performed with the $k$-means algorithm, and the number of clusters was set to 30.

Alternative models for batch effect correction were tested and evaluated. Specifically, these models included a batch effect that was given by $\lambda + \gamma$, $\gamma$ and $\lambda$. The outputs of these models were evaluated and compared to raw data and uMAIA results in PCA and UMAP space (by transforming all results by the same principal-component bases) and clustering with $k$ means as above. Results for each model were summarized by calculating Wasserstein distances between intensity distributions of corrected data for neighboring pairs of sections. The mean over all section-wise comparisons for each molecule was calculated and reported.

*Evaluation on proteomic and metabolic mass spectrometry imaging datasets.* Proteomic MSI datasets from ten consecutive kidney sections were retrieved from ref. 29 and processed with uMAIA using default parameters. Raw and corrected data distributions were summarized by calculating Wasserstein distances between intensity distributions of neighboring pairs of sections as above. The mean over sections for each molecule is calculated and reported. PCA coordinates across the sections without feature selection were calculated and displayed. The analysis was repeated with four sections representing similar tissues from the NIH Kidney Precision Medicine Project (Data availability).

**Evaluation on aligned data.** Spectra were aligned using MSIWarp, an open-source Python library (https://github.com/horvatovichlab/MSIWarp). For alignment, one reference spectrum was used to calibrate all acquisitions from the zebrafish at 72 hpf with the following parameters: bandwidth = 15, $\sigma = 3.0 \times 10^{-7}$, $\varepsilon = 1.55$.

We assessed the performance of the peak-calling modules from MALDIquant and uMAIA on the 72-hpf dataset after spectral alignment. For all images reconstructed between methods, we evaluated quality metrics as described in 'Analysis of single acquisitions and peak calling'. Images are counted as TPs if they are extracted by a method and they surpass two image quality metrics from either method, as FPs if they are extracted by the method but do not pass the quality metrics or as FNs if they were not called by a method while passing TP criteria for the other method.

The assessment of peak matching after spectral alignment involved featurizing the dataset using various bin sizes (0.001 Da, 0.0025 Da, 0.005 Da, 0.01 Da) and uMAIA. Next, for each method, we quantified the percent of features (that is, set of corresponding peaks) with missing detections across sections. For each feature, we estimated the maximal number of expected detections $D_f$ across sections by counting the sections that contained any detection within 0.01 Da from the feature centroid. A feature was considered 'incomplete' if the number of sections where it was detected was lower than $D_f$. Variance of the estimator was computed with a jackknife procedure.

Benchmarking tests to compare processing speeds of uMAIA's peak caller and alignment were performed using real data. Spectra were stacked to test larger dataset sizes.

## Zebrafish analyses

**Lipid annotation.** *Annotation of lipids.* Peak annotation was achieved by considering the $m/z$ values retrieved from bulk quantitative lipidomic experiments for only sum compositions of lipids. All lipids identified by bulk LC–MS/MS in positive and negative ion mode were considered, and $H^+$, $Na^+$, $K^+$, $H–H_2O^+$ and $NH_4^+$ adducts were selected for matching to MALDI-MSI data. MALDI compounds were matched to the nearest neighbor (within 0.01 Da) in $m/z$ from possible annotations.

In cases of multiple lipids matching a single $m/z$ value, including isobars and compounds with similar $m/z$ values, the signal was attributed to the most abundant lipid as measured by bulk lipidomics. To disentangle isobaric PC and PE species as well as PC-P/O and PE-P/O their relative abundance was specifically assessed (Supplementary Fig. 11b,c), and the $m/z$ value was assigned to the most abundant lipid. When comparable amounts of isobaric species were found, multiple annotations for the same $m/z$ value were reported. Lipid identity was assessed by querying the SwissLipids database via the METASPACE online platform[78]. Lipid annotations unconfirmed by SwissLipids (denoted by 'no' or blank entries in the 'Confirm_lipid_identity' column of Supplementary Table 1) should be interpreted with caution.

FP annotations were further removed by considering bulk LC–MS/MS data and MALDI-MSI data jointly. Global ad hoc renormalization was implemented by identifying four coefficients to rescale all lipid concentrations from MALDI-MSI data that would maximize the average similarity between datasets. Lipids with quantities that did not result in a positive correlation coefficient were discarded.

**Developmental analyses.** *Quantitative bulk liquid chromatography coupled to tandem mass spectrometry analysis.* Data were converted to mol% (measured concentration/total lipid concentration). Averages across replicates for each lipid and time point were calculated. To produce volcano plots, fold change in concentration was calculated between fish at 8 and 72 hpf with $P$ values (Student's $t$-test; $n = 4$). The UMAP representation was colored according to the time at which lipids were maximally abundant after dividing lipid concentration by the average.

*Identification of spatially informative molecules across time points.* To identify spatially informative lipids, we implemented a statistical test based on Moran's $I$ spatial autocorrelation index. For each lipid image, we calculated the observed Moran's $I$ and compared it to a null distribution established through pixel value reshuffling. To account for local baselines, we stratified pixels based on their associated total ion count measurements into eight bins. Next, independently for each lipid, the randomization procedure shuffled pixel values within these strata, maintaining a global notion of underlying tissue structure while disrupting specific lipid patterns. For each lipid, we computed the mean and standard deviation of the null distribution of Moran's $I$ from the realizations ($N = 100$) and calculated the $z$ score for the observed Moran's $I$, which was converted to $a$ $P$ value. $P$ values were adjusted (Bonferroni correction) to control for multiple comparisons. Lipid images with adjusted $P$ values below 0.01 were classified as showing significant spatial structure.

*Four-dimensional metabolic atlas and multivariate analysis.* Sagittal sections of zebrafish embryos at 8, 24, 48 and 72 hpf were processed with uMAIA modules. All molecules that had nonzero intensity values in at least 15% of pixels in 80% of sections were kept for downstream analysis.

To construct the volumetric data, consecutive slides were aligned using affine transforms. A 3D array was constructed, concatenating all the images, and smoothed (Gaussian filter; $\sigma = 0.4$). Arrays were visualized in napari.

Next, PCA was computed, and the first ten components (explaining over 90% of the dataset's variance) were considered. The $k$-means algorithm was applied for unbiased clustering of pixels.

*Computationally tracing metabolic pseudolineages.* To link lipid territories together across developmental stages, a pairwise squared Euclidean distance matrix between clusters was constructed considering the sequence of time points. Specifically, we start by initializing the distance matrix $D_{ij}$ to a large value (2.5). Next, we consider consecutive time points $t$ and $t + 1$, and we subset the lipids that were detected at both time points to obtain two matrices of cluster averages $X_{km}^t$ and $X_{lm}^{t+1}$ $\forall m : m \in \text{feat}_t$ and $m \in \text{feat}_{t+1}$, and squared Euclidean distances between these pairs were computed and set as values of $D_{ij}$ so that $D_{kl} \leftarrow \| X_{lm}^{t+1} - X_{km}^t \|_2$. Next, $D_{ij}$ was sorted by optimal leaf ordering. We show clusters in that order and display them to be spaced proportionally to the distance between consecutive clusters. The Hungarian algorithm was applied to match the clusters of each pair of time points $t$ and $t + 1$.

**Atlas at 72 hpf and multivariate analysis.** Twenty sagittal zebrafish sections at 72 hpf were processed with uMAIA. Before peak matching, the lowest two percentiles of intensity peaks were removed. After normalization, all molecules that had nonzero intensity values in at least 15% of pixels in 80% of sections were kept for downstream analysis.

Low-dimensional embeddings (nonnegative matrix factorization and diffusion map) were computed using as feature all the peaks annotated as lipids ('Lipid annotation'). A signal enhancement procedure inspired by that described by Pagoda 2 (ref. 79) was applied, in which each feature is rescaled by a factor proportional to its variance. The adaptation consists of using Moran's $I$ as a scaling factor instead of the observed–expected variance ratio because typical Poisson mean variance scaling is not expected in MSI data. Specifically, after minimum–maximum normalizing each lipid feature to 0–1, each lipid was multiplied by the median Moran's $I$ score achieved across sections, and the embedding routines were called on the rescaled matrix. Lipids modules were associated with top lipids sorting the positive loadings of each factor.

*Lipid–lipid correlation, enrichment and graph representation.* A pairwise correlation distance matrix was calculated across lipids and visualized after column–row sorting using the SPIN algorithm[69].

To construct the cluster enrichment heatmap, we applied *k* means and, for each cluster, averaged the intensity for each lipid, resulting in a matrix with rows (dimension number of regions) and columns (dimension number of lipids). Each column was minimum–maximum normalized to highlight the region in which lipids were most abundant.

To obtain the graph representation of the lipid–lipid similarity at each time point, a cluster enrichment score was calculated for each lipid. Pairwise Euclidean distances between enrichment scores were computed. The matrix of distances was used to compute a force-directed layout using the 500 edges with the smallest distances considered.

**Selection of metabolic genes, scRNA-seq and HybISS analysis.**
To identify zebrafish genes involved in metabolic processes, we used the KEGG database, including genes associated with enzymatic reactions and metabolite transport[80]. Single-cell transcriptomics data of zebrafish embryos at 72 hpf were obtained from the Zebrahub database[57]. We selected three subsets of genes: the 400 most variable genes, the 400 most variable metabolic genes and 400 random genes. For each subset, a training set of total cells was used to fit a linear discriminant analysis model to predict cell types. Predictive accuracy was assessed over 100 bootstrap iterations. Among these genes, a set of 50 marker genes and 50 metabolic genes was selected for the HyBISS experiment.

HybISS data analysis: for each section, individual transcripts were binned into 13.6-µm (40 pixels squared) areas, serving as proxies for cells. Genes with fewer than five counts and meta-cells with fewer than five genes were excluded. We used the SCANPY Python package[54] to cluster cells with similar transcriptional profiles and Squidpy[55] to generate spatial plots of these clusters. Cluster annotation was performed by correlating the mean expression of the HybISS cluster with that of annotated clusters from the scRNA-seq dataset, selecting clusters with the highest correlation scores. The analysis was repeated using only marker genes or metabolic genes.

To compare the information content between marker genes, metabolic genes and lipids, HybISS data were aligned with lipid data using Fiji. We considered all marker gene expression distributions, performed PCA across the feature space and applied Leiden clustering to the top ten principal components (Traag et al.[56]). Next, we retrieved bootstrap (*n* = 35) estimates of each set's (marker genes, metabolic genes, lipid distributions) ability to predict cluster identity: decision trees (scikit-learn implementation) with shallow depths (four) were used as classifiers. To identify the most informative genes for each lipid cluster, a decision tree was fit per cluster using gene expression as independent variables and binarized cluster labels as response variables (1 when belonging to the cluster in question, 0 otherwise). Feature importance was computed from the impurity decrease within each tree of the classifier for each lipid cluster.

**uMAIA package**
uMAIA is coded in Python, and its source is available at https://github.com/lamanno-epfl/uMAIA.

| Relevant packages | |
| --- | --- |
| **Figure** | **Required packages** |
| Fig. 1 | MALDIquant v1.22.1 |
| Fig. 2 | gurobipy v9.1.2 |
| Fig. 3 | ComBat v0.3.0, scArches v0.5.4, umap-learn v0.4.6, scikit-learn v0.24.2 |
| Fig. 4 | napari v0.4.17, umap-learn v0.4.6, scikit-learn v0.24.2, SciPy v1.10.1 |
| Fig. 5 | napari v0.4.17, scikit-learn v0.24.2, SciPy v1.10.1, Spotiflow v0.5.0 |
| Fig. 6 | NetworkX v2.5 |

| Relevant packages | |
| --- | --- |
| **Figure** | **Required packages** |
| Extended Data Fig. 1 | METASPACE (online interface) |
| Extended Data Fig. 2 | MALDIquant v1.22.1, MSiReader v1.0 |
| Extended Data Fig. 3 | MSIWarp v0.1, gurobipy v9.1.2 |
| Extended Data Fig. 4 | pyro-ppl v1.8.0, pyro-api v0.1.2, ComBat v0.3.0, brainmap v0.2.3 |
| Extended Data Fig. 5 | ComBat v0.3.0, scArches v0.5.4, scikit-learn v0.24.2 |
| Extended Data Fig. 6 | umap-learn v0.4.6 |
| Extended Data Figs. 7–9 | napari v0.4.17 |
| Extended Data Fig. 10 | Spotiflow v0.5.0, umap-learn v0.4.6 |
| Extended Data Figs. 3 and 4 | brainmap v0.2.3 |
| Extended Data Figs. 6 and 7 | napari v0.4.17, brainmap v0.2.3 |

v, version.

## Statistics and reproducibility
No statistical method was used to predetermine sample sizes. No data were excluded from the analyses. MSI experiments were not randomized and did not have covariates. Morpholino experiments were not randomized, and covariates were controlled by merging different clutches, and random embryo groups were allocated to morpholino or control experiments. A pool of non-injected siblings was kept under the same conditions to control for phenotypic defects unrelated to injections. The investigators were not blinded to allocation during experiments and outcome assessment.

## Reporting summary
Further information on research design is available in the Nature Portfolio Reporting Summary linked to this article.

## Data availability
All links to data and tutorials can be found on the website https://ZEBRA-L.epfl.ch. The website also includes a tool to visualize lipids in 3D across the sampled developmental time points. The raw data can be retrieved on METASPACE in .IBD and .imzML format at https://metaspace2020.eu/project/uMAIA. HybISS data are available at the following links: https://zenodo.org/records/14170238 (native coordinates)[81] and https://zenodo.org/records/14514399 (matching MSI)[82]. Publicly available MALDI-MSI datasets were retrieved from METASPACE. Human kidney samples were from the NIH Kidney Precision Medicine Project (ID 2024-06-06_19h26m13s, 2024-06-06_19h22m09s, 2024-06-06_19h16m21s, 2024-06-06_19h08m24s, 2024-06-06_19h03m38s, 2024-06-06_18h57m58s, 2024-06-06_18h45m39s, 2024-06-06_18h40m58s). Baboon lung samples were from the Pacific Northwest National Laboratory (ID 2024-06-11_18h30m14s, 2024-06-11_18h28m14s, 2024-06-11_18h35m41s, 2024-06-11_18h26m59s, 2024-06-11_18h43m25s).

## Code availability
uMAIA is implemented in Python and available as an open-source package on GitHub at https://github.com/lamanno-epfl/uMAIA. Source code, installation instructions and tutorials are also available on GitHub and our portal at https://ZEBRA-L.epfl.ch.

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

## Acknowledgements

This project has been made possible thanks to SNF grant numbers TMSGI3_218393, IZSEZ0_213427, 316030_183503, 310030_184926 and 310030_219382 as well as support from the Kristian Gerhard Jebsen Foundation, the Swiss Data Science Center and a SERI-funded ERC Consolidator Grant for Lipo-Trace. We thank members of the La Manno, D'Angelo, Oates and Gisou van der Goot laboratories for their generous feedback and discussions on the project, particularly A. Crotta Asis, J. Maillat and D. Korotkova. We also thank the entire EPFL Histology Facility for their assistance with experiments.

## Author contributions

H.H.S. developed the idea, designed and implemented the model framework, ran simulation experiments and performed the corresponding analysis, analyzed the MSI and LC–MS/MS data, made the figures, and wrote the paper. L.H.A.A. developed the idea, collected and acquired all MSI zebrafish data, collected zebrafish embryos for LC–MS/MS experiments, analyzed scRNA-seq data, helped analyze MSI and LC–MS/MS data, performed and analyzed HybISS experiments, and wrote the paper. L.A.R., G.V. and C.J. collected zebrafish embryos and helped with biological interpretation of the data. G.V. performed morpholino injections. A.G.M. refined the normalization component of the framework for faster implementation. A.H. performed HybISS experiments and analysis. A.S. adapted the HybISS pipeline for zebrafish, designed the probes and performed HybISS experiments. A.D.M. performed image analysis on raw HybISS acquisitions. J.P.-M. performed initial LC–MS/MS experiments. L.C. provided initial MSI data used for developing the framework, helped in their interpretation and advised on technical aspects. I.K. contributed to initial development of the pipeline for MSI data analysis. G.D'A. and G.L.M. developed the idea, supervised the project and wrote the paper. All coauthors read and approved the paper.

## Competing interests

The authors declare no competing interests.

## Additional information

**Extended data** is available for this paper at https://doi.org/10.1038/s41592-025-02771-7.

**Correspondence and requests for materials** should be addressed to Giovanni D'Angelo or Gioele La Manno.

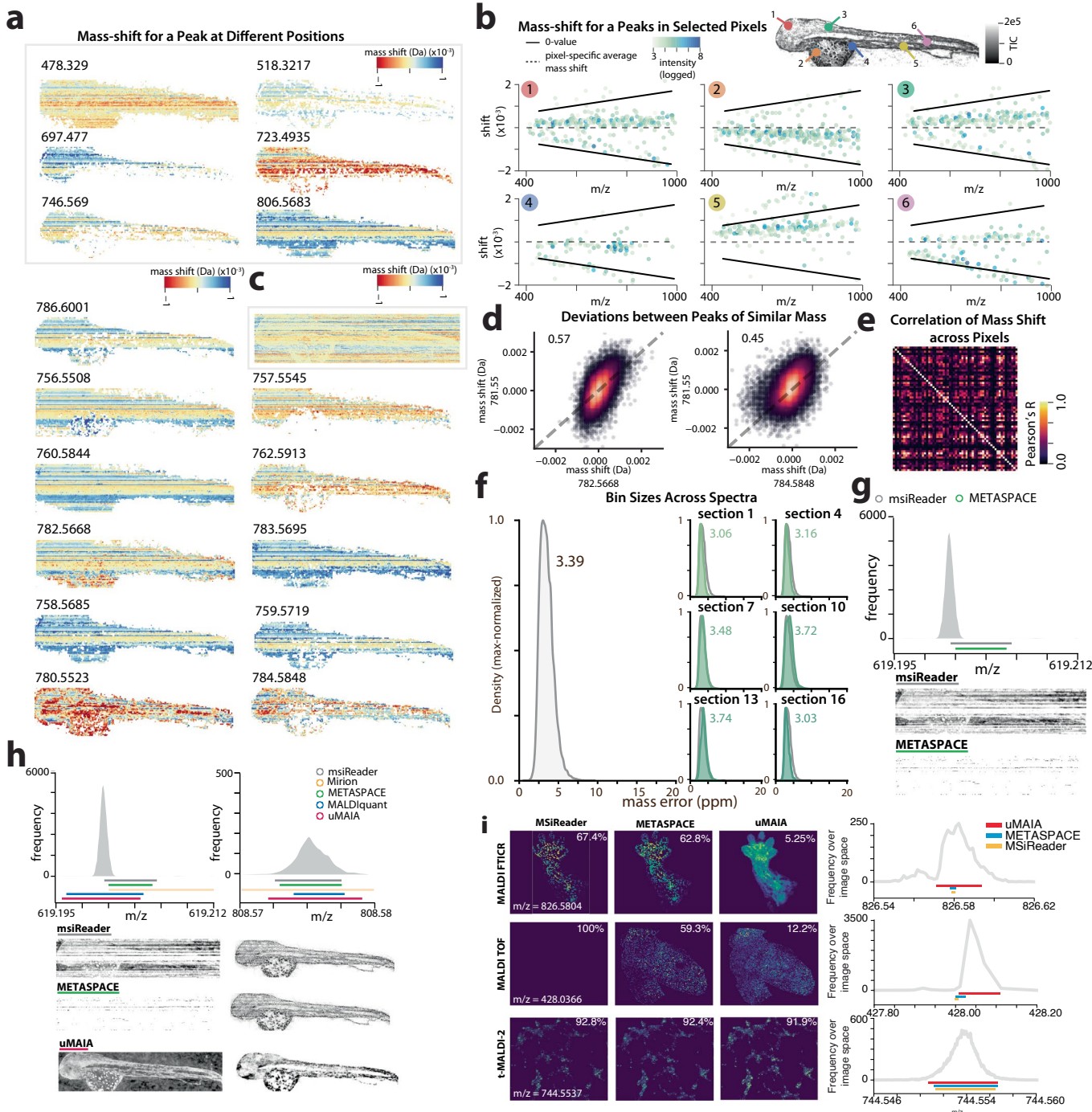

**Extended Data Fig. 1 | Characteristics of mass shifts and imaging of peaks by various software. a**. Mass shifts represented as images for a variety of peaks across spectra (pixels) **b**. Mass shifts for all peaks within selected spectra indicated by the color of dots (refer to image on top right for positioning) shaded by logged intensity. Solid and dotted lines represent −/+ 1.5ppm mass shifts and 0 within the spectrum, respectively. **c**. Overall average mass shift for all peaks. **d**. Scatterplots of mass shifts between quasi-isobaric molecules. Pearson's R is indicated in the top left corner. **e**. Correlation matrix of mass shifts between molecules with m/z values between 700-750. **f**. Left: overall bin sizes after peak calling with uMAIA across all sections in the 72 hpf zebrafish. The number indicates the mean value of the distribution. Right: bin sizes as in the left panel for individual sections (green) and overall distribution across sections as on the left (gray). **g**. Histogram representation of spectra for specific peaks using non-adaptive binning methods (msiReader and METASPACE) with the called bin shown as solid lines (above) and resultant image (below). **h**. Histogram representations of spectra for two peaks, with identified bin from various methods shown as solid colored line (above) and correspondent image (below). **i**. Images for various MSI datasets using msiReader, METASPACE and uMAIA (left). Imaged peak m/z indicated on bottom left. The percentage of missing pixels is indicated (top-right corner). The correspondent histogram representation is shown, with identified bins from various methods shown as a solid colored line (right).

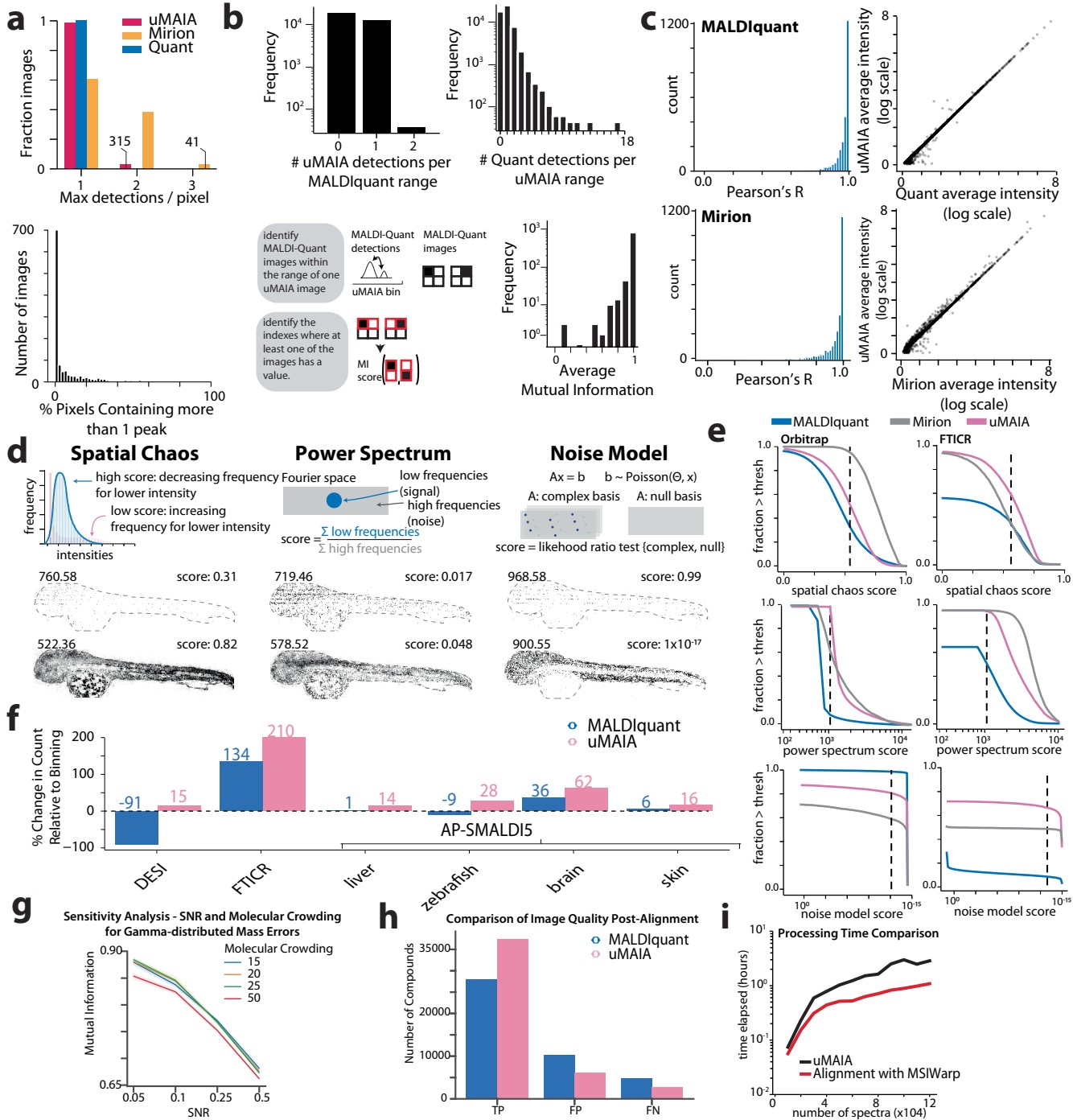

**Extended Data Fig. 2 | uMAIA peak calling performance on MSI datasets.**
**a**. Fraction of images containing multiple peaks for uMAIA, Mirion and MALDIquant (upper). Histogram across images reporting proportion of pixels containing more than one peak from uMAIA-retrieved images for one acquisition (lower). **b**. uMAIA-retrieved bins were paired with overlapping MALDIquant-retrieved bins. Number of uMAIA bins for each MALDIquant bin (upper left). Number of MALDIquant bins for each uMAIA bin (upper right). In instances where multiple MALDIquant bins were detected in a uMAIA bin, average mutual information between images is shown (bottom row). **c**. Similarity between images extracted for different compounds (MALDI vs uMAIA comparison above, and Mirion vs uMAIA below) and Pearson's R is between images (left column). Scatter plots show average intensity between images (right column). **d**. Metrics quantifying image quality: Spatial chaos, Power spectrum, and Noise model. Schematic depiction of metric (upper row) and example image with score displayed (lower row). **e**. Fraction of images (y-axis) surpassing metrics in (d) (x-axis) for uMAIA, MALDIquant and Mirion shown as a function of the metric.

Left column: Orbitrap MSI data; Right column: FTICR MSI data. **f**. Percent change in count of images retrieved by uMAIA and MALDIquant surpassing metrics in (d) with respect to the number retrieved by binning for various MSI acquisitions. **g**. Line plot tracking performance (mutual information) of uMAIA on simulated spectra (n=50) with mass errors drawn from Gamma distributions and various extents of molecular crowding (colors). Average over simulations (solid line) and standard deviation over realizations (shaded area) indicated. **h**. Barplots stratifying peak extraction outputs from MALDI-MSI experiment (72 hpf section) using uMAIA and MALDIquant after alignment with MSIWarp (Eriksson et al. 2020), and evaluated using metrics in (d). Images are counted as True-positives (TP) when extracted by a method and surpass 2 image quality metrics from either method; False-positives (FP) when extracted by the method but do not pass quality metrics; False-negatives (FN) if were not called by a method while passing TP criteria for the other. **i**. Processing time for dataset as a function of a number of spectra for uMAIA's peak calling module (black) and alignment with MSIWarp (red).

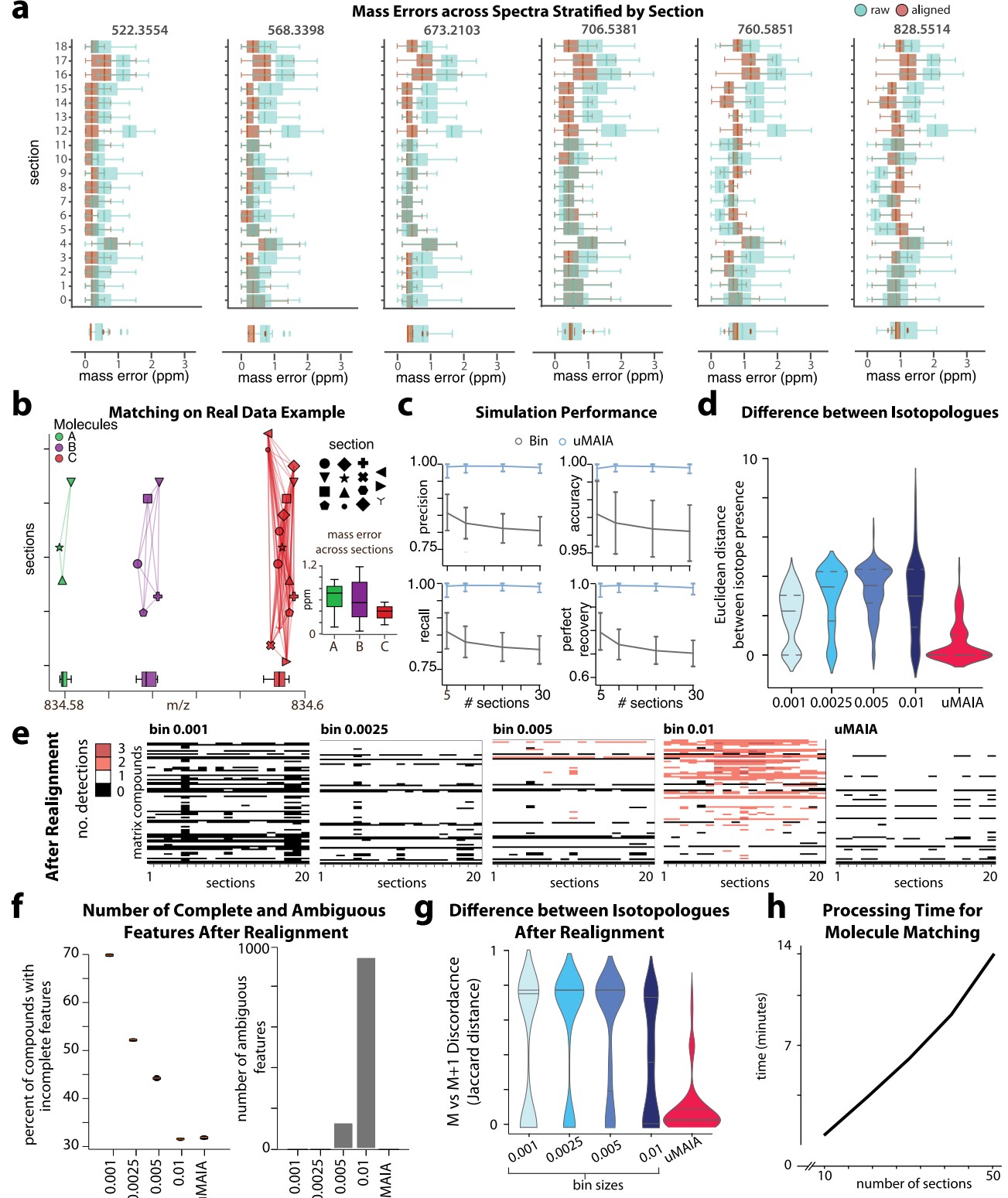

**Extended Data Fig. 3 | See next page for caption.**

**Extended Data Fig. 3 | Characterizations of uMAIA's molecule matching outputs. a**. Box and whisker plot representing interquartile ranges and medians of mass errors distributions (in ppm) associated with 6 different ions spanning 500-830 m/z range between within 19 sections from 72 hpf zebrafish dataset before (blue) and after (red) alignment with MSIWarp. Box and whisker plot in the bottom row indicate median and inter-quartile mass errors over sections. **b**. Molecule matching outputs for representative molecule set. Markers indicate peaks detected in different sections across datasets. Detections for each molecule matched by uMAIA are colored and connected with edges of the same color. **c**. Sensitivity analysis of molecule matching using different numbers of MSI acquisitions. Lineplots track different performance metrics (precision, accuracy, recall and perfect recovery) for simulated data (n=50). Error bars indicate standard deviation. **d**. Violin plots reporting distribution of Euclidean distances between isotopologue M+0 and M+1 presence across acquisitions after featurization. Distributions are computed across all pairs of sections for 20 molecules with highest signal intensity. **e**. Heatmaps visualizing MALDI matrix ions (rows) identified for sections (column) and different binning sizes and uMAIA after realignment with MSIWarp. Color-coded according to whether 0, 1, 2, or 3 peaks were identified within the bin for a given section. **f**. Boxplots representing percentage of compounds with incomplete features (left) and number of ambiguous detections (right) after realignment with MSIWarp for tested binning sizes and uMAIA after removing at random a section at a time (n=20). Boxplots show interquartile range and median. **g**. Violin plots reporting distribution of Jaccard distances between isotopologue M+0 and M+1 presence across acquisitions after featurization and alignment with MSIWarp. Distributions are computed across all the pairs of sections for 20 molecules with the highest signal intensity. **h**. Processing time for uMAIA's matching module as a function of the number of sections provided to algorithm.

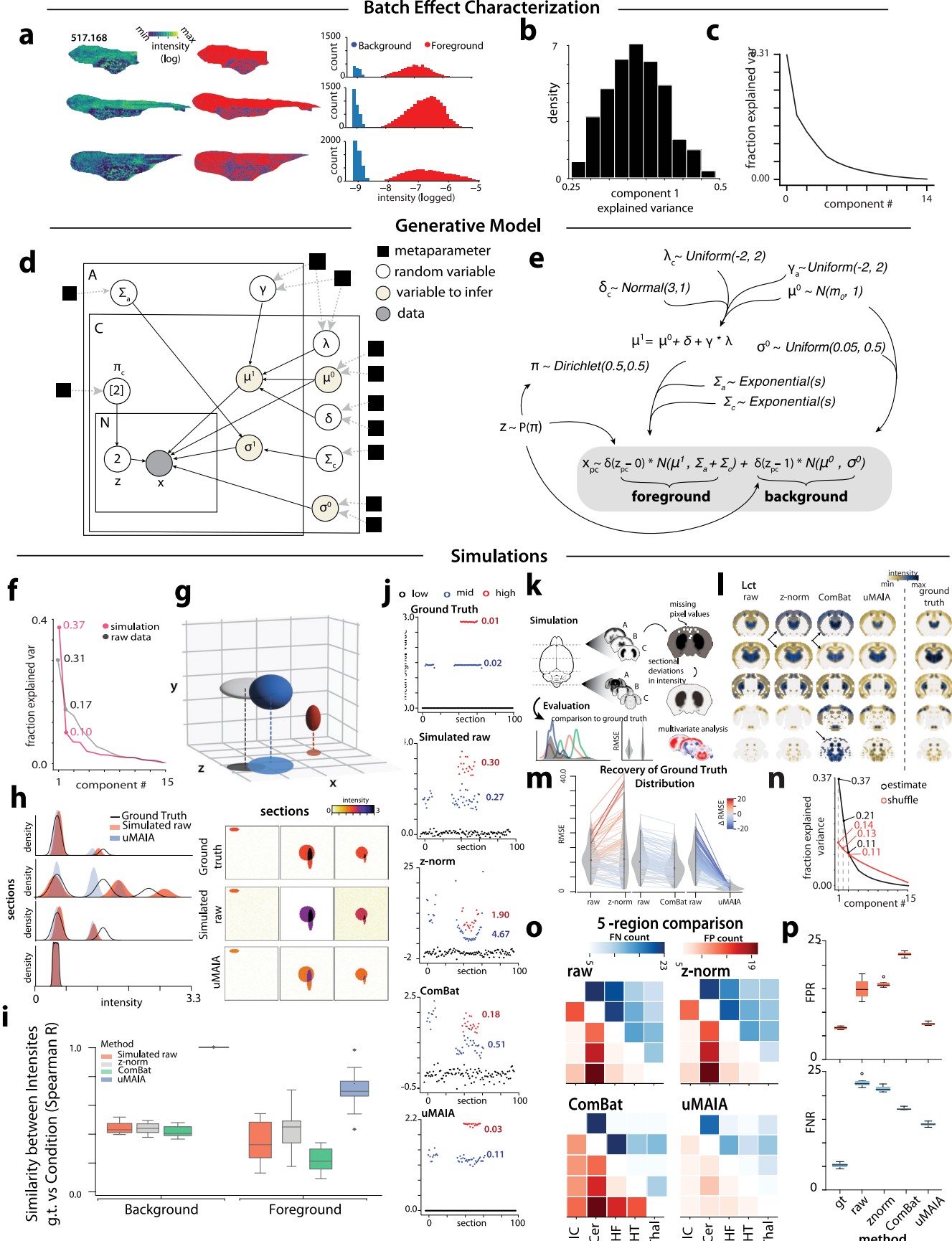

**Batch Effect Characterization**

**Generative Model**

**Simulations**

**Extended Data Fig. 4 | See next page for caption.**

**Extended Data Fig. 4 | Batch effect characterization and evaluation of ground truth retrieval by uMAIA normalization. a**. MSI images for m/z 517.17 at different mediolateral positions (left) and corresponding background/foreground assignments (center) and histograms of intensity values (right), histogram bars colored by assignment (for example, foreground/background). **b**. Histogram displaying distribution of bootstrap estimates of first eigenvalue of empirical batch-effect matrix. **c**. Line plot showing fraction of explained variance contributed by each principal component of batch effects estimator. **d**. Hierarchical Bayesian model in plate notation, including metaparameters, latent variables, and data for intensity-distorted signal estimation. **e**. Generative model representation of model in (d). **f**. Fraction of explained variance contributed by each component from SVD as in (c) (black line) overlaid with those from simulation data (red line). **g**. Visualization of ellipsoid simulation. **h**. Intensity distributions in simulated data, comparing ground truth, simulated raw and uMAIA normalized data for 4 sections (left) with corresponding images shown (right). **i**. Box and whiskers plots quantifying similarity of corrected images with ground truth results are computed across 4 sections and variables of simulation. Similarity calculated with Spearman's R between intensities. **j**. Scatterplots for ground truth, simulated raw, z-normalized, ComBat- and uMAIA-corrected data, each displaying mean molecule signal across sections. Low- (black), mid- (blue) and high-intensity (red) modes are highlighted. Variance for the three categories indicated by colored numbers. **k**. Schematic ABA ISH data simulation and evaluation. **l**. Visualization of images corresponding to gene Lct of ground truth (right) and simulated raw and corrected data with z-normalization, ComBat, and uMAIA (left). **m**. Root Mean Square Error (RMSE) distribution with respect to ground truth images for simulated raw data and data corrected with z-normalization, ComBat, and uMAIA. Lines drawn between genes and colored by relative increase or decrease of RMSE compared to simulated raw data. **n**. Fraction of explained variance contributed by each component from SVD applied to batch effect estimates for ISH simulations (black line) overlaid with those from shuffled data (red line). **o**. Heatmaps quantifying total number of false-positives and false-negatives after differential expression tests across all genes for 5 major anatomical regions for simulated data and data corrected with different methods. **p**. False-positive and false-negative rates after differential expression tests using bootstrapped sections (n=20). Boxplots indicate interquartile range and median. Distributions shown for ground truth (gt), simulated raw and corrected data with z-normalization, ComBat and uMAIA.

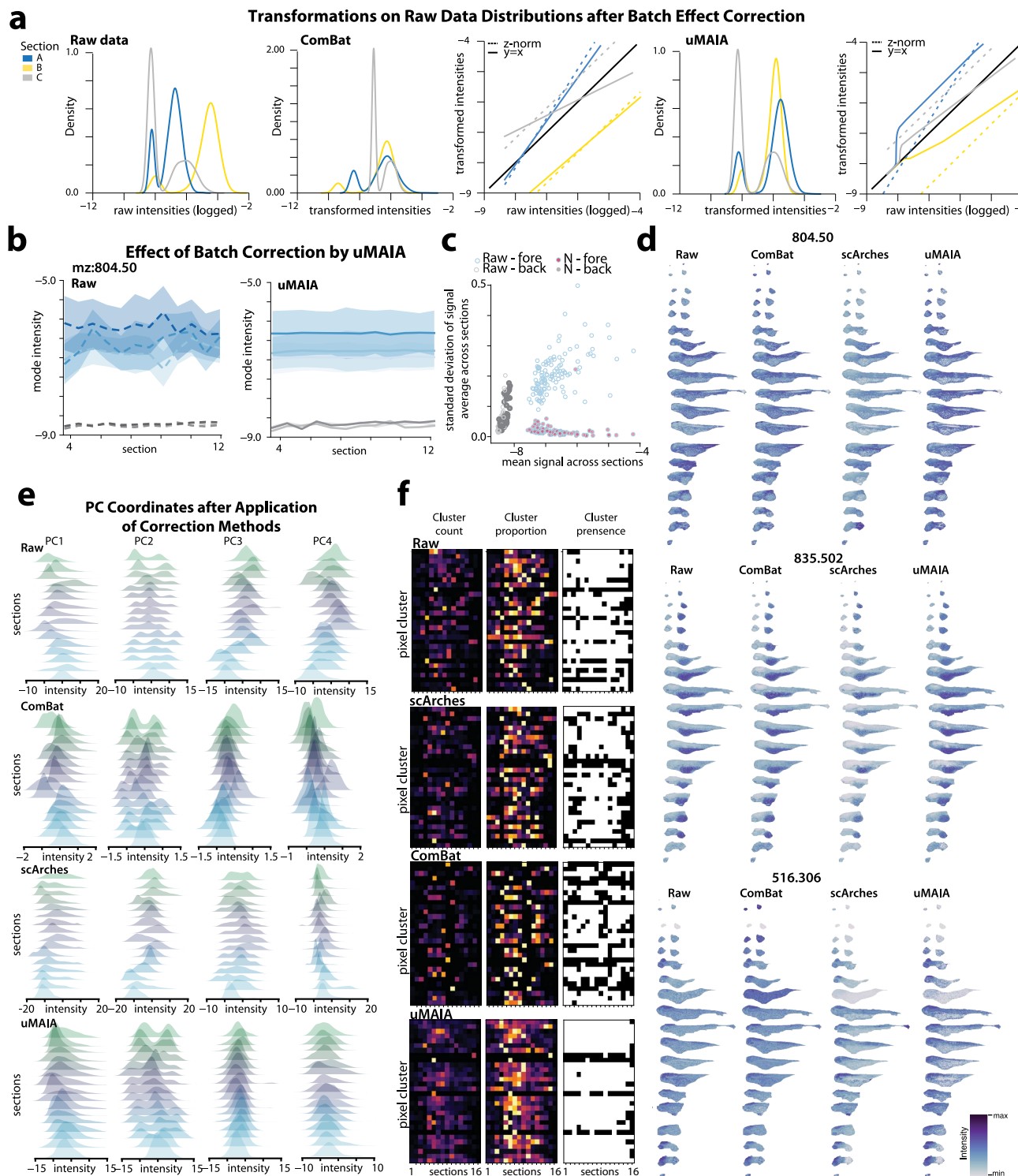

**Extended Data Fig. 5 | Comparison of batch effect correction methods on data.**
**a.** Signal intensity distributions across three acquisitions, color corresponding
to the acquisition. On the left is the raw MALDI MSI data. To the right, the same
data corrected with ComBat and uMAIA with related quantile-quantile plots
representing the transformation between raw and normalized intensities **b.** Line
plot tracking the mean molecule (mz=804.50) intensity and variance (shaded
area) in background and foreground modes across consecutive samples for raw
and uMAIA-corrected data. **c.** Scatter plot of mean signal vs standard deviation of
signal average across sections for background and foreground modes for raw and

uMAIA-corrected data (right) **d.** Images of 3 peaks (mz=804.50, 516.30, 835.50)
with spatially variable distributions for raw data and data corrected with ComBat,
scArches and uMAIA. **e.** Distribution plots showing principal component
coordinates of pixels across different acquisitions for raw data and normalized
data. The principal components appear to shift as a result of the distortion of
many individual features, the normalization rescues their consistency between
acquisitions. **f.** Heatmaps representing the number, proportion and presence
of pixel clusters across sections ('cluster count', 'cluster proportion' and 'cluster
presence') for raw and corrected data.

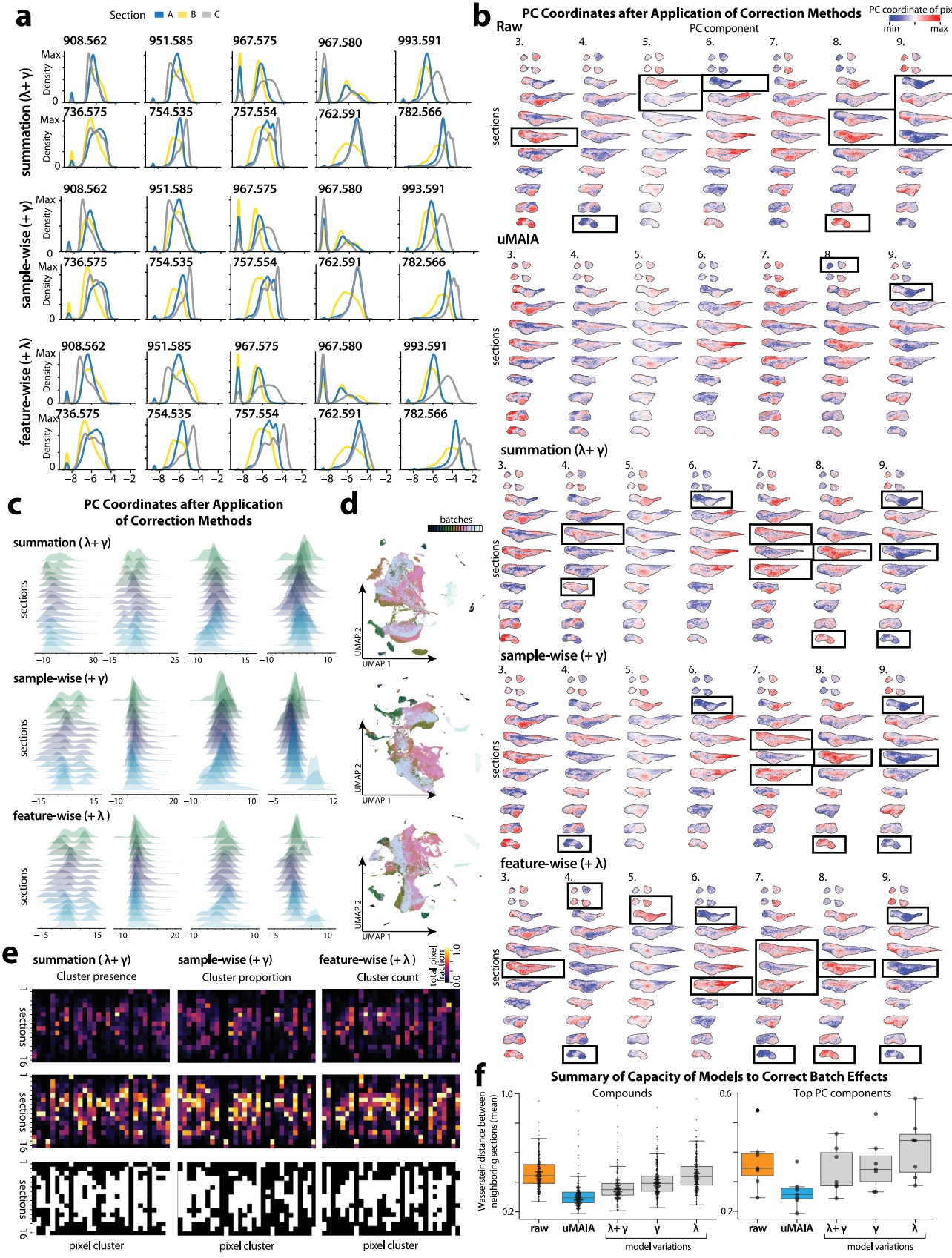

**Extended Data Fig. 6 | See next page for caption.**

**Extended Data Fig. 6 | Performance of Alternative Factorizations of the Batch-Effect Parameters in the Normalization Framework. a**. Intensity distributions of 10 different compounds in the 72 hpf zebrafish after normalization using 3 alternative models for batch-effect factors. Colors correspond to different sections. **b**. Images of principal component coordinates of Raw, uMAIA and the 3 alternative models (see Methods: 'Batch effect characterization'). Black boxes indicate residual batch-effects present in the images. **c**. PC coordinates of dataset after correction by various models across sections after normalization using the alternative models. **d**. Low dimensional representation (UMAP embedding) of pixels for dataset after normalization by alternative models. Pixels are colored by the section from which they originate. See Fig. 4F. **e**. Heatmaps representing matrices counting the number of pixels belonging to a given pixel cluster across sections ('cluster count'), distribution of cluster proportion across sections ('row-normalized') and presence of cluster within section ('>10 pixels in cluster') within each section for raw and corrected data. Clustering was performed for each dataset independently. See Extended Data Fig. 5 for a comparison with other tested methods. **f**. Average Wasserstein distances between intensity distributions of neighboring sections over molecules (left) and top 10 principal components (right) for raw data, data corrected with uMAIA and alternative models of batch-effect factorization. Boxplots indicate interquartile range and median.

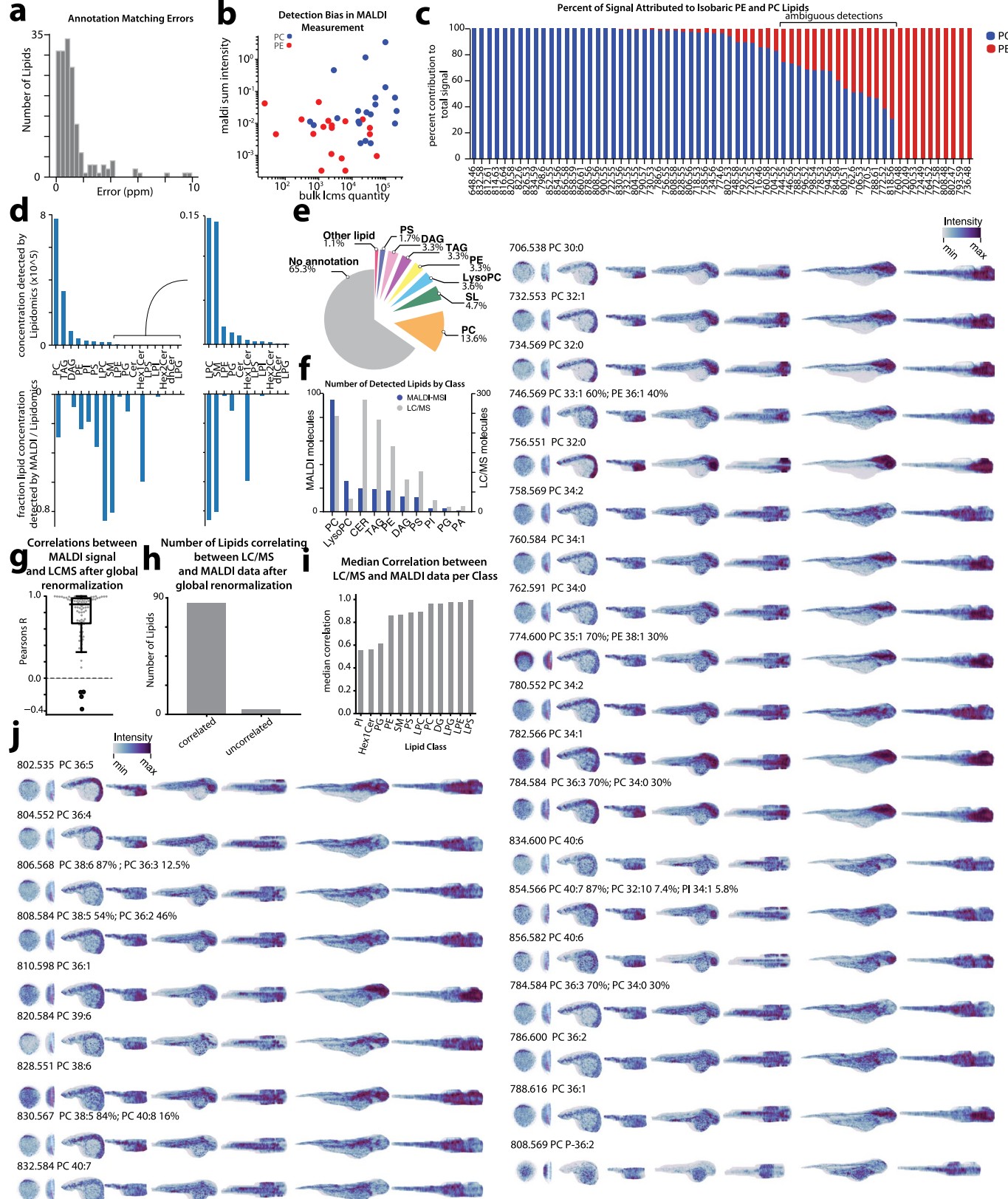

**Extended Data Fig. 7 | See next page for caption.**

**Extended Data Fig. 7 | 3D visualization of lipids during zebrafish development.**
**a**. Histogram of mass errors for all annotated compounds with respect to measured m/z in the 72 hpf zebrafish embryo dataset. **b**. Scatter plot depicting measured concentration from bulk LC-MS/MS against MALDI-MSI quantification after TIC normalization. **c**. Stacked barplot of the percent contribution of PC and PE isobars (based on LC-MS/MS quantification) for specific m/z peaks identified in MALDI-MSI. **d**. Barplots show the concentration of different lipid classes as estimated by a quantitative LC-MS/MS experiment (top row). Relative intensity values of MALDI-MSI are shown below for comparison (bottom row). **e**. Proportion of lipid species detected by MALDI-MSI stratified by class. **f**. Barplot reporting the number of lipids detected by MALDI-MSI compared to LC-MS/MS stratified by lipid class (n=1). **g**. Boxplot and swarmplot of

Pearson's R correlation scores between MALDI-MSI intensity sums and quantitative LC-MS/MS measurements per lipid after global renormalization. Boxplot indicates interquartile range and median. Horizontal dashed line at 0. **h**. Number of lipids with positive correlation coefficients in (g) (n=1). **i**. Median correlations in (g) after grouping by lipid class (n=1). **j**. Sagittal and dorsal views of 3D reconstructed images (max projection) of lipids detected in MALDI-MSI data at 8, 24, 48 and 72 hpf sorted by lipid class. Note orientation of 8 hpf embryo is of uncertain orientation. Species name and m/z displayed on top left. PC: phosphatidylcholine; LPC: lysophosphatidylcholine; SL: sphingolipid; TG: triacylglycerol; PS: phosphatidylserine; PE: phosphatidylethanolamine; PI: phosphatidylinositol; LPI: lysophosphatidylinositol; PG: phosphatidylglycerol; LPE: lysophosphatidylethanolamine; DG: diacylglycerol.

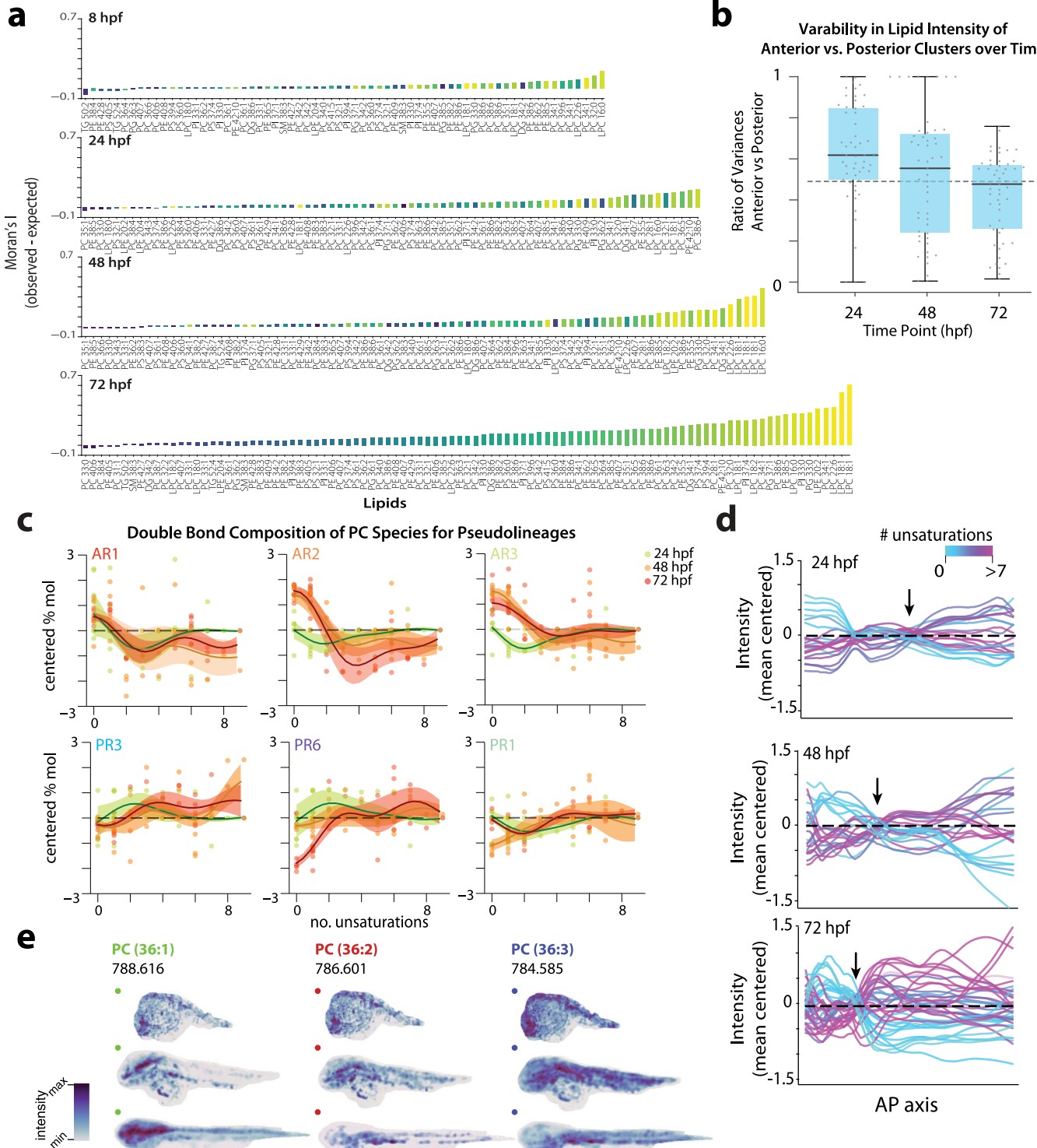

**Extended Data Fig. 8 | 4D MSI atlas of zebrafish development. a**. Barplots indicating Moran's I score per lipid after subtraction of the expected score per timepoint. Colors are according to the sorting of lipids by the observed - expected score at 72 hpf. **b**. Box and whiskers plot displaying the distribution variability ratio between different metabolic regions, plot is stratified by time point, each dot represents a lipid. Box plot indicates interquartile range and median. **c**. Mean-centered mol% of PC lipids for different pseudolineages as a function of their unsaturation content. Shaded area indicates variance. Green: 24 hpf; Orange: 48 hpf; Red: 72 hpf. **d**. Lipid intensity trends for PC species along anterior-posterior (AP) axis for each developmental stage. Varying degrees of unsaturation are indicated by colors. Heatmap of mean-centered average lipid intensity for the 3 species over developmental time (lower row). **e**. Visualization of sagittal view of PC 36:1, PC 36:2 and PC 36:3 shown in Fig. 4J.

**Extended Data Fig. 9 | 3D visualization of lipid distributions across the 72 hpf zebrafish embryo. a.** Lateral and dorsal views of 3D distributions (max projection) of lipids at 72 hpf. Intensities across the whole volume are presented using maximum intensity projection visualization. **b.** Lateral views of 3D distributions of lipids detected in MALDI-MSI data at 72 hpf sorted by lipid class. Species name and m/z displayed on the top right. PC: phosphatidylcholine; LPC: lysophosphatidylcholine; SL: sphingolipid; TG: triacylglycerol; PS: phosphatidylserine; PE: phosphatidylethanolamine; PI: phosphatidylinositol; LPI: lysophosphatidylinositol; PG: phosphatidylglycerol; LPE: lysophosphatidylethanolamine; DG: diacylglyceride.

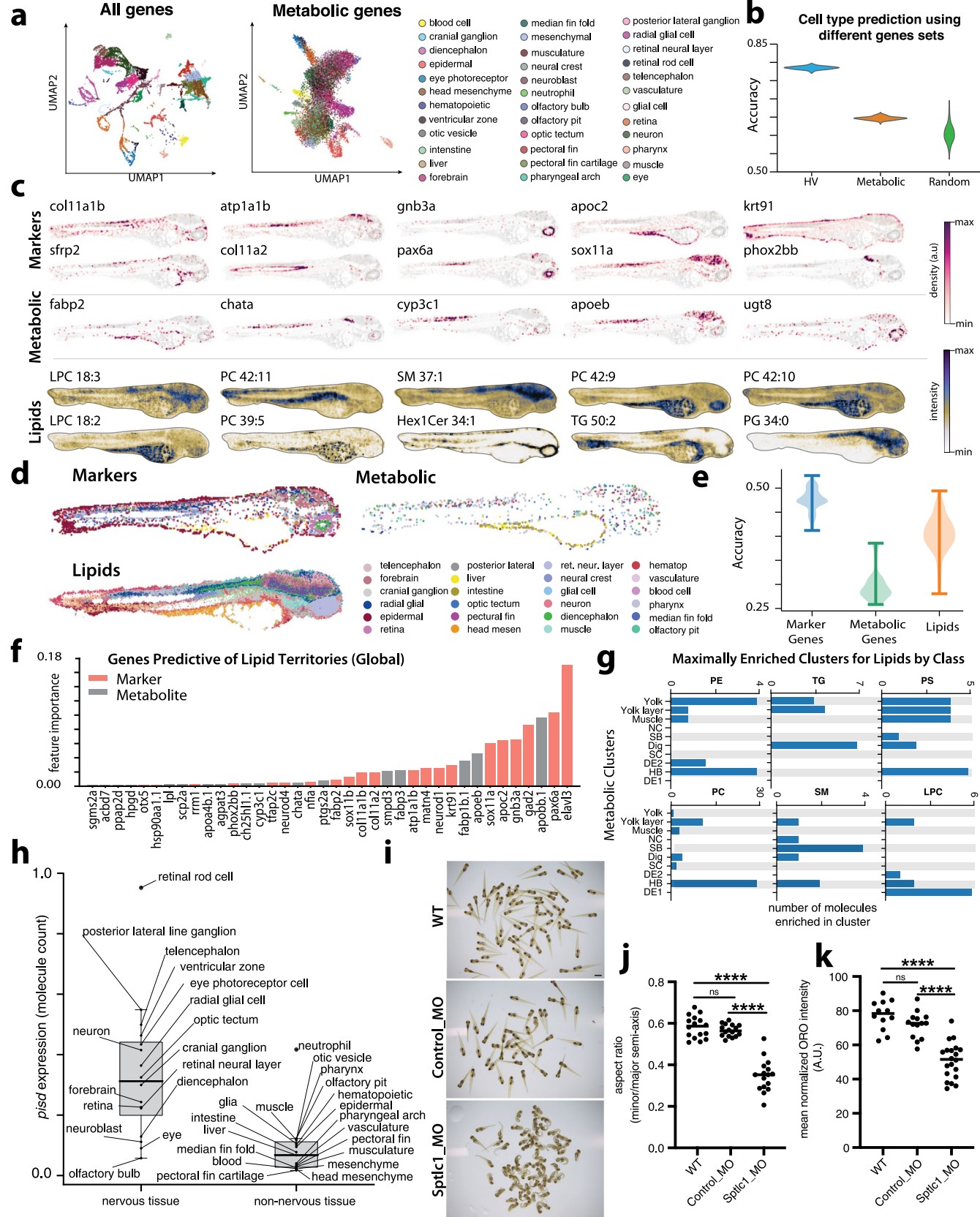

**Extended Data Fig. 10 | See next page for caption.**

**Extended Data Fig. 10 | Comparison between spatial lipidomics with transcriptomics and metabolic organization of lipids in the 72 hpf zebrafish embryo. a**. Low dimensional representation embeddings (UMAP) of cells' gene expression from scRNAseq data (Lange et al., 2023). Two embeddings are shown using either all measured genes or the subset associated with lipid metabolic processes. Cells are color-coded according to annotated cluster identity. **b**. Accuracy of models using different gene sets (Highly Variable: HV, Metabolic, and a random set of genes) in predicting annotated cell identity. Violin plots represent bootstrap distributions (n=100). **c**. Examples of spatial gene expression profiles of cell-type marker (upper row), and metabolic (bottom row) genes quantified by HybISS. Representative lipid distributions from MALDI-MSI data are shown (bottom row). **d**. Cell clusters according to gene expression in different gene sets. Clusters were obtained independently for each gene set, and assigned to the annotated cell type with the most similar transcriptional profile. Only cells with at least 5 counts were considered for clustering, thus generating sparser clusters in the metabolic set. Lipid clusters were obtained independently and color-matched to anatomically similar transcriptional clusters. **e**. Bootstrap estimates for weighted average accuracy for classifiers using marker genes, metabolic genes, or lipids. Error bar indicates maximum and minimum value in distribution. **f**. Barplot reporting sorted feature (gene) importance scores computed from impurity decrease within each tree of a decision tree classifier of lipid territories (n=1). Bars indicate genes from the marker (red) and metabolic (gray) subsets. **g**. Barplots for each lipid class detailing lipid enrichment of different clusters (that is, metabolic region). Lipids are assigned to metabolic region where they are maximally expressed (n=1). **h**. Barplot displaying molecule counts of Pisd transcript in cell types defined from 72 hpf zebrafish scRNAseq dataset (Lange et al., 2023), data is stratified in nervous and non-nervous tissue. **i**. Representative images of 120 hpf zebrafish morphants for Sptlc1 (Sptlc1 MO), control injected (Control MO) and not-injected siblings (WT). Scale bar=1mm. **j**. Aspect ratio (minor/major semiaxis) of swim bladder in larvae injected with control (Control MO) or anti-Sptlc1 morpholinos (Sptlc1 MO) and not-injected siblings (WT); unpaired t-test, ****= p<0.0001(*Sptlc1*) zebrafish. **k**. Mean ORO intensity in swim bladder in larvae injected with control (Control MO) or anti-Sptlc1 morpholinos (Sptlc1 MO) and not-injected siblings (WT); unpaired t-test, ****= p<0.0001(*Sptlc1*) zebrafish.

# Reporting Summary

Please do not complete any field with "not applicable" or n/a.  Refer to the help text for what text to use if an item is not relevant to your study.
For final submission: please carefully check your responses for accuracy; you will not be able to make changes later.

## Statistics

For all statistical analyses, confirm that the following items are present in the figure legend, table legend, main text, or Methods section.

| n/a | Confirmed | |
|-----|-----------|---|
| ☐ | ☑ | The exact sample size (*n*) for each experimental group/condition, given as a discrete number and unit of measurement |
| ☐ | ☑ | A statement on whether measurements were taken from distinct samples or whether the same sample was measured repeatedly |
| ☐ | ☑ | The statistical test(s) used AND whether they are one- or two-sided *Only common tests should be described solely by name; describe more complex techniques in the Methods section.* |
| ☐ | ☑ | A description of all covariates tested |
| ☐ | ☑ | A description of any assumptions or corrections, such as tests of normality and adjustment for multiple comparisons |
| ☐ | ☑ | A full description of the statistical parameters including central tendency (e.g. means) or other basic estimates (e.g. regression coefficient) AND variation (e.g. standard deviation) or associated estimates of uncertainty (e.g. confidence intervals) |
| ☐ | ☑ | For null hypothesis testing, the test statistic (e.g. *F*, *t*, *r*) with confidence intervals, effect sizes, degrees of freedom and *P* value noted *Give P values as exact values whenever suitable.* |
| ☐ | ☑ | For Bayesian analysis, information on the choice of priors and Markov chain Monte Carlo settings |
| ☐ | ☑ | For hierarchical and complex designs, identification of the appropriate level for tests and full reporting of outcomes |
| ☐ | ☑ | Estimates of effect sizes (e.g. Cohen's *d*, Pearson's *r*), indicating how they were calculated |

*Our web collection on statistics for biologists contains articles on many of the points above.*

## Software and code

Policy information about availability of computer code

| Data collection | MALDI MSI data were collected withTune from Thermofisher scientific. Described in the methods section. |
|-----------------|---|
| Data analysis | Python 3.11, Fiji 2.9.0 and Ashlar .Source code at https://github.com/lamanno-epfl/uMAIA/tree/main. Described in methods section. |

For manuscripts utilizing custom algorithms or software that are central to the research but not yet described in published literature, software must be made available to editors and reviewers. We strongly encourage code deposition in a community repository (e.g. GitHub). See the Nature Portfolio guidelines for submitting code & software for further information.

## Data

Policy information about availability of data

All manuscripts must include a data availability statement. This statement should provide the following information, where applicable:
- Accession codes, unique identifiers, or web links for publicly available datasets
- A description of any restrictions on data availability
- For clinical datasets or third party data, please ensure that the statement adheres to our policy

The raw data can be retrieved on METASPACE in .IBD and .imzML format at https://metaspace2020.eu/project/uMAIA. HybISS data are available at the following link: https://zenodo.org/records/14170238 (native coordinates) and https://zenodo.org/records/14514399 (matching MSI).

Publicly available MALDI-MSI datasets were retrieved from METASPACE. Human kidney samples from the NIH Kidney Precision Medicine Project (ID: 2024-06-06_19h26m13s, 2024-06-06_19h22m09s, 2024-06-06_19h16m21s, 2024-06-06_19h08m24s, 2024-06-06_19h03m38s, 2024-06-06_18h57m58s, 2024-06-06_18h45m39s, 2024-06-06_18h40m58s)

Baboon lung samples from the Pacific Northwest National Laboratory (ID: 2024-06-11_18h30m14s, 2024-06-11_18h28m14s, 2024-06-11_18h35m41s, 2024-06-11_18h26m59s, 2024-06-11_18h43m25s)

Supplementary Table 1: Contains the set of all annotable m/z detections retrieved from MALDI-MSI data (in rows) with metadata in columns including, but not limited to, lipid quantities, potential isobars, sum compositions, specific lipid species identified from LC-MS/MS.

Supplementary Table 2: Contains all information retrieved from bulk LC-MS/MS quantification. These include 4 sheets: (1) lipid quantifications before normalization, (2) total protein content for each sample, (3) lipid quantifications after normalization, (4) a lipid transition table that lists lipid names with their precursor mz, retention time, ionisation and adduct.

## Research involving human participants, their data, or biological material

Policy information about studies with human participants or human data. See also policy information about sex, gender (identity/presentation), and sexual orientation and race, ethnicity and racism.

| | |
|---|---|
| Reporting on sex and gender | NA |
| Reporting on race, ethnicity, or other socially relevant groupings | NA |
| Population characteristics | NA |
| Recruitment | NA |
| Ethics oversight | NA |

Note that full information on the approval of the study protocol must also be provided in the manuscript.

# Field-specific reporting

Please select the one below that is the best fit for your research. If you are not sure, read the appropriate sections before making your selection.

☑ Life sciences    ☐ Behavioural & social sciences    ☐ Ecological, evolutionary & environmental sciences

For a reference copy of the document with all sections, see nature.com/documents/nr-reporting-summary-flat.pdf

# Life sciences study design

All studies must disclose on these points even when the disclosure is negative.

| | |
|---|---|
| Sample size | MSI dataset: two individuals per developmental stage were sampled; sample size was chosen to provide representative data while accounting for the labor-intensive nature of MSI technology; LC-MS/MS: four biological replicates (each consisting of 15 pooled individual embryos) were used per developmental stage; this number was chosen based on previously published protocol indicating that this sample size provides robust signal detection and reproducibility in LC-MS/MS. Morpholino experiments: at least six embryos were analyzed for each condition and experiment. This sample size is consistent with standards in developmental biology for detecting phenotypic effects.HybISS:only 2 embryos were used; for a qualitative assessement this sample size was sufficient |
| Data exclusions | Only normally developing wild-type embryos were used. Low-quality MSI data were excluded from the final dataset. Embryos that did not survive the morpholino injection were excluded. For HybISS, genes with few counts were not considered. |
| Replication | MSI atlases are based on one embryo densely sampled (every 20 um), with additional representative sections from a second embryo, for each developmental stage. Morpholino injections were repeated 3 times. HybISS was performed once on a sagittal section and once on a coronal section.All attempts at replication were succesful. |
| Randomization | No randomization was performed for the MSI data. For morpholinos experiments, different clutches were merged and random embryos groups were allocated to morpholinos or controls; a pool of non-injected siblings were kept at the same conditions to control for phenotypic defects not-related to the injections; no abnormalities were observed. |
| Blinding | For the morpholino experiment the phenotype was so strong that blinding was not deemed necessary. |

# Behavioural & social sciences study design

All studies must disclose on these points even when the disclosure is negative.

| | |
|---|---|
| Study description | NA |
| Research sample | NA |
| Sampling strategy | NA |
| Data collection | NA |
| Timing | NA |
| Data exclusions | NA |
| Non-participation | NA |
| Randomization | NA |

# Ecological, evolutionary & environmental sciences study design

All studies must disclose on these points even when the disclosure is negative.

| | |
|---|---|
| Study description | NA |
| Research sample | NA |
| Sampling strategy | NA |
| Data collection | NA |
| Timing and spatial scale | NA |
| Data exclusions | NA |
| Reproducibility | NA |
| Randomization | NA |
| Blinding | NA |

Did the study involve field work?    ☐ Yes    ☐ No

## Field work, collection and transport

| | |
|---|---|
| Field conditions | NA |
| Location | NA |
| Access & import/export | NA |
| Disturbance | NA |

# Reporting for specific materials, systems and methods

We require information from authors about some types of materials, experimental systems and methods used in many studies. Here, indicate whether each material, system or method listed is relevant to your study. If you are not sure if a list item applies to your research, read the appropriate section before selecting a response.

## Materials & experimental systems

| n/a | Involved in the study |
|---|---|
| ☑ | ☐ Antibodies |
| ☑ | ☐ Eukaryotic cell lines |
| ☑ | ☐ Palaeontology and archaeology |
| ☐ | ☑ Animals and other organisms |
| ☑ | ☐ Clinical data |
| ☑ | ☐ Dual use research of concern |
| ☑ | ☐ Plants |

## Methods

| n/a | Involved in the study |
|---|---|
| ☑ | ☐ ChIP-seq |
| ☑ | ☐ Flow cytometry |
| ☑ | ☐ MRI-based neuroimaging |

## Antibodies

| | |
|---|---|
| Antibodies used | NA |
| Validation | NA |

# Eukaryotic cell lines

Policy information about cell lines and Sex and Gender in Research

| Cell line source(s) | NA |
| --- | --- |
| Authentication | NA |
| Mycoplasma contamination | NA |
| Commonly misidentified lines (See ICLAC register) | NA |

# Palaeontology and Archaeology

| Specimen provenance | NA |
| --- | --- |
| Specimen deposition | NA |
| Dating methods | NA |

☐ Tick this box to confirm that the raw and calibrated dates are available in the paper or in Supplementary Information.

| Ethics oversight | |
| --- | --- |

Note that full information on the approval of the study protocol must also be provided in the manuscript.

# Animals and other research organisms

Policy information about studies involving animals; ARRIVE guidelines recommended for reporting animal research, and Sex and Gender in Research

| Laboratory animals | Danio rerio embryos up to 120 hours post-fertilization |
| --- | --- |
| Wild animals | No wild animals were used in the study |
| Reporting on sex | Danio rerio embryos at the stages considered in this study are considered female as sexual determination become apparent later on in development. |
| Field-collected samples | No field collected samples were used in the study |
| Ethics oversight | All zebrafish husbandry procedures have been approved and accredited by the federal food safety and veterinary office of the canton of Vaud (VD-H23) |

Note that full information on the approval of the study protocol must also be provided in the manuscript.

# Clinical data

Policy information about clinical studies
All manuscripts should comply with the ICMJE guidelines for publication of clinical research and a completed CONSORT checklist must be included with all submissions.

| Clinical trial registration | NA |
| --- | --- |
| Study protocol | NA |
| Data collection | NA |
| Outcomes | NA |

# Dual use research of concern

Policy information about dual use research of concern

### Hazards

Could the accidental, deliberate or reckless misuse of agents or technologies generated in the work, or the application of information presented in the manuscript, pose a threat to:

No | Yes
- ☑ ☐ Public health
- ☑ ☐ National security
- ☑ ☐ Crops and/or livestock
- ☑ ☐ Ecosystems
- ☑ ☐ Any other significant area

## Experiments of concern

Does the work involve any of these experiments of concern:

No | Yes
- ☑ ☐ Demonstrate how to render a vaccine ineffective
- ☑ ☐ Confer resistance to therapeutically useful antibiotics or antiviral agents
- ☑ ☐ Enhance the virulence of a pathogen or render a nonpathogen virulent
- ☑ ☐ Increase transmissibility of a pathogen
- ☑ ☐ Alter the host range of a pathogen
- ☑ ☐ Enable evasion of diagnostic/detection modalities
- ☑ ☐ Enable the weaponization of a biological agent or toxin
- ☑ ☐ Any other potentially harmful combination of experiments and agents

# Plants

| | |
|---|---|
| Seed stocks | NA |
| Novel plant genotypes | NA |
| Authentication | NA |

# ChIP-seq

## Data deposition

☐ Confirm that both raw and final processed data have been deposited in a public database such as GEO.

☐ Confirm that you have deposited or provided access to graph files (e.g. BED files) for the called peaks.

| | |
|---|---|
| Data access links<br>*May remain private before publication.* | NA |
| Files in database submission | NA |
| Genome browser session<br>(e.g. UCSC) | NA |

## Methodology

| | |
|---|---|
| Replicates | NA |
| Sequencing depth | NA |
| Antibodies | NA |
| Peak calling parameters | NA |
| Data quality | NA |
| Software | NA |

# Flow Cytometry

## Plots

Confirm that:

☐ The axis labels state the marker and fluorochrome used (e.g. CD4-FITC).

☐ The axis scales are clearly visible. Include numbers along axes only for bottom left plot of group (a 'group' is an analysis of identical markers).

☐ All plots are contour plots with outliers or pseudocolor plots.

☐ A numerical value for number of cells or percentage (with statistics) is provided.

## Methodology

| | |
|---|---|
| Sample preparation | NA |
| Instrument | NA |
| Software | NA |
| Cell population abundance | NA |
| Gating strategy | NA |

☐ Tick this box to confirm that a figure exemplifying the gating strategy is provided in the Supplementary Information.

# Magnetic resonance imaging

## Experimental design

| | |
|---|---|
| Design type | NA |
| Design specifications | NA |
| Behavioral performance measures | NA |

| | |
|---|---|
| Imaging type(s) | NA |
| Field strength | NA |
| Sequence & imaging parameters | NA |
| Area of acquisition | NA |

Diffusion MRI      ☐ Used      ☐ Not used

## Preprocessing

| | |
|---|---|
| Preprocessing software | NA |
| Normalization | NA |
| Normalization template | NA |
| Noise and artifact removal | NA |
| Volume censoring | NA |

## Statistical modeling & inference

| | |
|---|---|
| Model type and settings | NA |
| Effect(s) tested | NA |

Specify type of analysis:      ☐ Whole brain      ☐ ROI-based      ☐ Both

| Statistic type for inference | NA |
|---|---|
| (See Eklund et al. 2016) | |

| Correction | NA |
|---|---|

## Models & analysis

| n/a | Involved in the study |
|---|---|
| ☐ | ☐ Functional and/or effective connectivity |
| ☐ | ☐ Graph analysis |
| ☐ | ☐ Multivariate modeling or predictive analysis |

| Functional and/or effective connectivity | NA |
|---|---|

| Graph analysis | NA |
|---|---|

| Multivariate modeling and predictive analysis | NA |
|---|---|

