## [Peer Review File · Nature Methods]

Unified Mass Imaging Maps the Lipidome of Vertebrate Development

Corresponding Author: Professor Gioele La Manno

A version of this paper was originally rejected for publication by Nature Methods, however that decision was reconsidered after appeal by the authors.

Version 0:

Decision Letter:

14th Oct 2024

Dear Gioele,

Your Resource entitled "Unified Mass Imaging Maps the Lipidome of Vertebrate Development" has now been seen by 3 reviewers, whose comments are attached. While they find your work of potential interest, they have raised serious concerns which in our view are sufficiently important that they preclude publication of the work in Nature Methods, at least in its present form.

As you will see, the reviewers raise serious concerns especially about the quality of the lipidomics data .

Should further experimental data allow you to fully address these criticisms we would be willing to look at a revised manuscript (unless, of course, something similar has by then been accepted at Nature Methods or appeared elsewhere). This includes submission or publication of a portion of this work somewhere else. We hope you understand that until we have read the revised paper in its entirety we cannot promise that it will be sent back for peer-review.

If you are interested in revising this manuscript for submission to Nature Methods in the future, please contact me to discuss your appeal before making any revisions. Otherwise, we hope that you find the reviewers' comments helpful when preparing your paper for submission elsewhere.

Sincerely,
Madhura

Madhura Mukhopadhyay, PhD
Senior Editor
Nature Methods

Reviewers' Comments:

Reviewer #1:

Remarks to the Author:

chede et al. present a large MSI lipidomics dataset of zebrafish embryos at different time points of development and insights from analyzing this dataset.

The manuscript is interesting to read and has a number of positive things to be said about:

- the authors collected a large novel dataset from a fundamentally-relevant biological system at relevant time points of

development — this is the key contribution of the manuscript

- they have developed a new approach for data processing to reduce batch effect and artifacts — please see below my discussion about it, points 2-3

- they have performed analysis of this dataset and found several patterns

- they have showed the complementary value of MSI compared to spatial transcriptomics, although in a limited way for one example section

However, there is a number of points which should be addressed if the manuscript will be selected for revision:

1. The dataset should not be called an atlas. It's a collection of MSI datasets from sections of different zebrafish. There are no anatomical annotations provided, alignment between sections does not really provide necessary resolution (that's why authors needed to do extra coronal sections in Figure 7e-f), there is no alignment between individual animals. There is only 1 UMAP shown for (potentially) pixels from different sections (or animals), see Figure 4F, mainly to illustrate the batch effect. The dataset is not fully available and is not interoperable — I could not find the 3D volumetric data for all lipids reported with provided metadata. What is referred as “tutorial” provides on GitHub is a pretty technical script illustrating how a part of the data was created, for selected individual sections.

2. The analysis of what they call “peak calling” and “network flow peak matching” is interesting. However, it seems to me that they are digging themselves out of a hole they created. As far as I could understand, MSI data in this paper is of relatively poor quality to the point when it would be strongly questioned in a mass spectrometry journal. Let me explain. I could not access the data to look into it (see the point below about Availability), but from Figure S4a it seems like the authors found that some peaks have mass shifts reach 20 ppm. Having m/z error of 20 ppm is not acceptable for the mass spectrometer authors use (Orbitrap) even for MALDI-imaging and even for a larger number of sections analyzed (when calibration may need to be performed between runs). If this is indeed the case, this on its own is a red flag as it can lead to false positive and false negative lipid identifications. Taking into account that the authors use simple m/z matching for lipid ID (with m/z tolerance of ~10ppm), their strategy to compensate for mass errors of 20 ppm can lead to false positives within m/z range of as high as 30 ppm (which is way too high for lipid ID). So, this raises a question, why authors proceeded with using data of such normally-inacceptably low mass accuracy. This also raises questions whether there is a general need for such sophisticated data preprocessing.

3. The authors over-blow the novelty of analyzing MSI data from many sections in their introduction and discussion; see e.g. “previous MSI analysis tool have focused on the analysis of a single or a small number of acquisitions” and “rendering analysis of >100 acquisitions virtually impossible”. First, their own dataset has less than 100 sections (only 96). This is not that much as already in 2013, Ojetjen et al, in *Journal of Proteomics*, 90, published a 3D MSI dataset of a mouse kidney containing 122 sections with all sections co-registered to create a coherent 3D volume, and the analysis was performed in a commercially-available software and resulted in substantially more pixels (2 million) than the authors present in their manuscript. Many more publications on 3D MSI were published since then. Many studies have analyzed more than 100 sections from different individuals, e.g. Porcari et al. in 2018, *Analytical Chemistry* 90(19), have performed a multi-center DESI-imaging study of samples from 103 patients. Another early example is Balluff et al. 2011, *Am J Pathol* 179(6) where MSI data from 181 tissue sections was analyzed. So, overall, although authors present an interesting dataset, it's by far not the largest.

4. There are several concerns about the lipids ID:

1. The authors m/z-match lipids identified from bulk LC-MS/MS in both positive and negative polarity to peaks from MSI data collected in the positive polarity. This likely led to the false positive detection, likely increased for lipids detected in LC-MS/MS in the negative polarity. How do authors controlled this false positive detection?

2. How do the authors handle isomers?

3. At the considered tolerance in the m/z matching (and particularly after the discussed data preprocessing), the authors should definitely have isobars. And even if those isobars were not identified in LC-MS/MS, it's a well-known fact that they should be considered in lipid ID. However, I found no discussion or description how the authors handled it.

4. Which adducts were considered for LC-MS/MS and for MSI, in particular when doing the m/z-matching

5. Data is not available — that once again stresses the point that this dataset is not really an atlas. In any case, the data should be available as it's the key contribution of the paper.

1. For MSI datasets, I could not download any dataset from METASPACE (I only see a spinning wheel after clicking at the link)

2. For HybISS data, there is no link provided

3. 3D or 4D data is not available

There are a few minor points:

1. There is a discrepancy about m/z bins in Fig 3d-f (0.001, 0.025, 0.05, 0.1) and Fig 3g-h (0.001, 0.0025, 0.005, and 0.01), and in Methods.

2. The data does not have single-cell resolution and the statement “quasi-single-cell” in the abstract is confusing and should not be used.

3. In line 595, I'd recommend to avoid the word “strong” in “strong biochemical organization” as it's a subjective judgement based on the view of the network.

Reviewer #2:

Remarks to the Author:

In the paper "Unified Mass Imaging Maps the Lipidome of the vertebrate development", Schede et al. describes a new computational workflow, uMAIA, for the processing and analysis of large scale MALDI-MSI data. They apply this pipeline to the first large scale un-targeted lipidomic spatial atlas of zebrafish development. The paper is an extremely precious resource for the community, is well written and fully understandable despite the high complexity of the underlying science and methods. The provided python pipeline was easily installed and tested successfully on two workstations (Mac and Ubuntu) despite long running time. Only a limited number of points has to be addressed:

Major points:

The 4D lipidomic atlas generated by the authors is of high interest for the Zebrafish community. However the processed data are not provided by the authors. As the number of people with the required expertise and computational resources to re-process and analyze the raw is extremely limited, the authors should provide a processed file that is easy to access and work with.

The authors have developed a very powerful and useful pipeline for the processing of the large scale MALDI data. However the analysis of the resulting processed data seems quite simple as there is no feature selection performed and that a simple PCA combined with k-means clustering is performed. As the algorithms used for the processed analysis of data can drastically impact the biological interpretation of the data (such as for scRNA-seq analysis). The authors should have tested different strategies of low dimensional embedding (SVD, NMF, autoencoder, laplacian eigenmaps, diffusion map etc...) and clustering (KNN-graph community based detection such as Louvain and Infomap). In addition, as the authors use a PCA, all lipids have the same contribution to the analysis, likely obscuring the signal of a few number but key lipid species. The impact of highly variable lipid selection and adjustment of the variance (notably implemented in the Pagoda2 pipeline) should thus be tested.

The normalization model assumes the multiplication of the sample and feature related batch effects and is elegantly justified by the one-rank approximation of the empirical batch effect. Still, the authors should compare their multiplicative model with three other models:

An additive model, where the batch effect is $\lambda + \gamma$

A model with only the feature effect

A model with only the sample effect

Identification of spatially-informative molecules across timepoints: the identification of those genes is, according to the method section, done by performing an ANOVA on the k-mean clusters. This approach (in addition to be basically double-dipping, see the work of Lucy Gao on the subject) is highly problematic as this approach will not identify genes based on their spatial pattern. Two different approaches could be used :

Basic spatial index scores, such as Geary's C or Moran's I. The measured values could be compared to the one observed in case of spatial shuffling and a corresponding p-value computed.

The (semi) variogram could be computed for each lipid and the ratio of the nugget effect compared to the total variance, allowing to identify genes which variance is primarily driven by spatial variations.

Minor points:

Figure 5i : the different panels are poorly comparable as the order of the lipids is not conserved across. The same order should be used in to better visualize the evolution of the correlation structure. If a hierarchical clustering has been used please indicated it.

A more in-depth joint-analysis of the HybISS and MALDI data could have been performed, typically by using methods such canonical correlation analysis or its sparse variant (Extensions of sparse canonical correlation analysis with applications to genomic data)

The peak calling algorithm is thoroughly described in the method section, however it quickly appears that the authors basically perform a one-dimensional watershed on the histogram spectrum using local maxima as seeding points. The authors should clearly state that they have used this approach and discuss/explain the implicit parameter values of the watershed algorithm they have selected.

Unclear terms used in page 5 to describe the normalisation model: sensitivity and susceptibility. The term sensitivity is already extensively used in the machine learning literature. Instead the terms 'feature-related' and 'sample-related' batch effects could be used.

The authors explain that the metabolic genes are not useful to predict the different 'lipidic regions' and that highly variable genes are more useful. The authors could be maybe show the most highly predictive genes from the HybISS data.

The prior distribution of delta_c (signal difference between foreground and background) is set by the authors to Gaussian(3,1). While the use of such prior is reasonable, is still allows a negative value of delta_c. A prior that is only defined on positive real values could instead be used, typical a gamma or an exponential distribution.

Minor typo : Page 22, legend. "Generative model representation if (d)" and not (c).

Minor typo bis: Page 35. First A is not written in a gothic manner.

Reviewer #3:

Remarks to the Author:

The authors describe a new framework for joint analysis of large mass spec imaging datasets for the construction of multi-dimensional atlases. This approach attempts to combat the challenges faced in the field of mass spec imaging, particularly for lipids and metabolites. Overall the manuscript is of high caliber and comprehensively demonstrates the feasibility of such approach for even temporal image data. However it appears to be severely lacking in the lipidomic aspect, with very little

detail on how annotations are derived for the MSI dataset and how the experiment was run. This needs to be corrected.

Critical

- How were the annotations derived for figure 5 and 6, specifically the lipid annotations? This needs to be explained and outlined in detail.
- How are the authors disentangling PE and PC isomers (similarly for PC O and PE O).
- How was the plasmenyl (PE P) and plasmanyl (PE O) annotations determined? Can the authors provide evidence of this (Figure S12)
- Can the authors provide more detailed mass spectrometry conditions for the data acquisition, notably the mass resolution, acquisition time for each pixel etc. Was the quadrupole used at all? Similarly, was this full-MS scan only?
- Can the authors provide a table of masses, adducts and species reported in the MSI study? Similarly, this needs to be provided for the quantitative LC-MS study as well.
- Brief detail (table format is fine) on how the species were annotated in this set is also required at minimum, particularly since this is a methods journal, and many of the species reported appear to be completely novel (PS O-19:1/18:1 in figure S9, unusual LPI 17:0 and 19:0 representing a large concentration on figure 7B.).

Major

- The authors first describe overcoming technical variation by using an adaptive peak calling approach to better capture unique features. While the described results and figures show a relatively normally distributed mass error, can the authors comment on how this approach handle specific, systematic biases (e.g. temperature changes resulting in gradual shifts for TOFs) which can be instrument dependent, while many of the described results here are orbitrap based. This includes the figure highlighted in S2b, the pink feature that forms a bimodal distribution. Can a bimodal distribution be mistaken as two features? There does not appear to be a description of fine tuning the approaches when applying it. Similarly, simulations appear to be largely based on normal distribution when in real life instances, this might not be the case.
- In the proposed peak calling / matching approaches, it would be good to quantify the processing speed differences between this and the existing approaches described, as that would dictate feasibility of these approaches.
- Can the authors better label the figures on S8? E.g. colors for the A and what they represent, figure s8e y axis etc. These are almost uninterpretable. Similar issues exist in figure 4e.
- Can the authors annotate better figure 5D? Notably some names are cut off and the adducts for each mass needs to be presented
 - o Figure 6 needs to be corrected in a similar manner.
- Figure 5G isn't described correctly (there are no loadings on my figure).
- While the LC-MS/MS and the MSI results are independent samples, it would be worth correlating these results by summing up all pixel per sample in a given MSI analysis. Species identified in the MSI experiment should correlate and change in a similar fashion in this manner to the bulk lipidomics data. This can assist in determining the strength of lipid annotation in the MALDI analysis. E.g. plot PE 38:6 between the LC-MS (concentration) and MSI data (summed intensity) with all the timepoints.
- Can the authors detail how the annotations described in line 489 was done?
- Can the authors outline the results in a supplementary table for Figure S9A-B?
- Can the authors outline the packages used for analysis across the figures in the methods section? Many of these are not commonly used, and many more have different versions available in the public space. Include versions used if available. Examples (not limited to) the UMAP projection, COMBAT normalisation, scArches etc.

Minor

- Can the authors provide details of the tests used for several analysis (e.g. Figure S9A has p-values and fold difference, but the analysis is not described anywhere).
- There is a typo on page 38, line 1241.
- Have the authors investigated what the green island on Figure 4F refers to, it appears to be a significant outlier for a batch.

Although we cannot publish your paper, it may be appropriate for another journal in the Nature Portfolio. If you wish to explore the journals and transfer your manuscript please use our manuscript transfer portal. You will not have to re-supply manuscript metadata and files, unless you wish to make modifications. For more information, please see our [manuscript transfer FAQ](http://www.nature.com/authors/author_resources/transfer_manuscripts.html?WT.mc_id=EMI_NPG_1511_AUTHORTRANSF&WT.ec_id=AUTHOR) page.

** For Nature Portfolio general information and news for authors, see <http://npg.nature.com/authors>.

Version 1:

Decision Letter:

12th Dec 2024

Dear Gioele,

Thank you for your letter asking us to reconsider our decision on your Resource, "Unified Mass Imaging Maps the Lipidome of Vertebrate Development". After careful consideration we have decided that we are willing to consider a revised version of your manuscript as outlined in your plan.

- * include a point-by-point response to our referees and to any editorial suggestions
- * please underline/highlight any additions to the text or areas with other significant changes to facilitate review of the revised manuscript
- * address the points listed described below to conform to our open science requirements
- * ensure it complies with our general format requirements as set out in our guide to authors at www.nature.com/naturemethods
- * resubmit all the necessary files electronically by using the link below to access your home page

Link Redacted

We hope to receive your revised paper within ten weeks. If you cannot send it within this time, please let us know. In this event, we will still be happy to reconsider your paper at a later date so long as nothing similar has been accepted for publication at Nature Methods or published elsewhere.

OPEN SCIENCE REQUIREMENTS

REPORTING SUMMARY AND EDITORIAL POLICY CHECKLISTS

When revising your manuscript, please submit reporting summary and editorial policy checklists.

DATA AVAILABILITY

CODE AVAILABILITY

Please include a "Code Availability" subsection in the Online Methods which details how your custom code is made available. Only in rare cases (where code is not central to the main conclusions of the paper) is the statement "available upon request" allowed (and reasons should be specified).

For more information on our code sharing policy and requirements, please see:
<https://www.nature.com/nature-research/editorial-policies/reporting-standards#availability-of-computer-code>

MATERIALS AVAILABILITY

SUPPLEMENTARY PROTOCOL

To help facilitate reproducibility and uptake of your method, we ask you to prepare a step-by-step Supplementary Protocol for the method described in this paper. We [encourage authors to share their step-by-step experimental protocols](https://www.nature.com/nature-research/editorial-policies/reporting-standards#protocols) on a protocol sharing platform of their choice and report the protocol DOI in the reference list. Nature Portfolio's protocols.io is a free-to-use and open resource for protocols; protocols deposited onto protocols.io are citable and can be linked from the published article. More details can found at [protocols.io](https://www.protocols.io/help/publish-articles).

ORCID

Sincerely,
Madhura

Madhura Mukhopadhyay, PhD
Senior Editor
Nature Methods

Version 2:

Decision Letter:

Our ref: NMETH-RS57455B

10th Mar 2025

Dear Gioele,

Thank you for submitting your revised manuscript "Unified Mass Imaging Maps the Lipidome of Vertebrate Development" (NMETH-RS57455B). It has now been seen by the original referees and their comments are below. The reviewers find that the paper has improved in revision, and therefore we'll be happy in principle to publish it in Nature Methods, pending minor revisions to satisfy the referees' final requests and to comply with our editorial and formatting guidelines.

While not mandatory, we think it would really benefit the paper if you addressed Ref 2's comments on whether the method would work on other MSI data. When resubmitting, please add a point-by-point response to the remaining reviewer comments.

TRANSPARENT PEER REVIEW

Nature Methods offers a transparent peer review option for new original research manuscripts submitted from 17th February 2021. We encourage increased transparency in peer review by publishing the reviewer comments, author rebuttal letters

and editorial decision letters if the authors agree. Such peer review material is made available as a supplementary peer review file. **Please state in the cover letter 'I wish to participate in transparent peer review' if you want to opt in, or 'I do not wish to participate in transparent peer review' if you don't.** Failure to state your preference will result in delays in accepting your manuscript for publication.

ORCID

Sincerely,
Madhura

Madhura Mukhopadhyay, PhD
Senior Editor
Nature Methods

Reviewer #2 (Remarks to the Author):

Review of "Unified Mass Imaging Maps the Lipidome of Vertebrate Development"

Major Points

Point 1: accessibility of the data and of the atlas

I would like to highlight the excellent work done by the authors to make the lipidomic data more accessible. The online atlas is easy to use and more importantly the processed data can easily be downloaded and analyzed using the Python script. In my case I have downloaded several of them and analyzed them using an in-house R script in less than a hour

Point 2: improved analysis of the data

The authors have significantly improved the analysis of the pre-processed data through the use of NMF, diffusion map and graph-based clustering. The results they obtain are highly consistent with their initial results and even provided higher details, thus validating their initial approach and findings. It is worth noting that the authors nicely adapted the approach of Pagoda2 to the lipid data by rescaling the data according to Moran's I score !

Point 3: Comparison of different normalization models

The authors have benchmarked their (multiplicative) model against three other models, showing that only the multiplicative model is able to correct the majority of the observed batch effects. While the benchmark of the authors is quite exhaustive, I think that an additional plot that would summarize figure S9a and S9c by measuring the distance of the distribution (using Jensen Shanon or KL divergence for instance) between the different sections would be more efficient and easier to understand for readers.

Point 4: Spatial analysis

The authors have successfully implemented a meaningful statistical test to detect spatially variable genes by carefully removing the background effect of total lipid amount. The only criticism I have is regarding the number of simulations that seems particularly low (N=20) and that could be increased.

Minor Points

Point 5: Corrected

Point 6 : Could be discussed but agree with the author's point.

Point 7 : Corrected

Point 8 : Corrected

Point 9 : Corrected

Point 10 : Corrected

Additional point

While not being in the scope of this manuscript, knowing whether uMaia would work on other type of mass spectrometry data (typically proteomic or glycomics) and if it could be used to reanalyze previously published datasets is of high interest. Such benchmark of the tool by the authors would be highly useful to the community and would be complementary to this first paper.

Reviewer #2 (Remarks on code availability):

As previously describe in the first revision round, the code is reproducible and runs smoothly. The jupyter notebook are well documented and written. The only downside is that the analysis of the processed hdf5 file is only explained for Python and not for R users, who still represent a very important part of the bioinformatic community. I have written a small draft for the loading and visualization of the data in R that the authors could maybe expand and add to their repo.

Reviewer #3 (Remarks to the Author):

The authors have addressed nearly all of my concerns and queries. A few notable items remain that require double checking, largely on the LC-MS/MS data used for annotating the spatial lipidomics data.

The transitions provided by the authors for plasmenyl and plasmanyl lipids does not appear to discriminate between the species for the LC-MS/MS analysis. For example, the two isomers PE O-16:1/22:6 and PE P-16:0/22:6 have identical retention times (1.75), precursor (746.5) and product ions (327.2), and are thus essentially duplicated in the dataset and reported twice for the same ion. Similar issues are observed for LPC O and LPC P, which some are duplicated. This should be fixed. Unless there is evidence otherwise, PE P species should have the corresponding PE O annotated with it (e.g. PE O 16:1/22:6 & PE P 16:0/22:6).

Many of the isobaric species that are not unit resolution resolved are also in the dataset (e.g. PC O-16:0/16:0 and PC 15:0_16:0 are duplicated in RT, precursor and product).

Similarly for LPC O which are measured in positive mode, the 104.1 fragment would be present in the corresponding isobaric LPC species (e.g. LPC 17:0 and LPC O-18:0).

Can the authors remove all duplicates in the dataset. It would be sufficient to name them as a mixture (e.g. PC O-16:0/16:0 & PC 15:0_16:0) to avoid confusion for readers for ones that were duplicated.

Oddly the concentrations are not identical for the duplicates. Can the authors provide the list of internal standards (aligned to the lipids that used it) that were used to calculate the concentrations for this study?

We would like to thank the reviewers for their thoughtful and constructive feedback on our manuscript. In response to their comments, we have made significant enhancements to both the manuscript and the accessibility of our resource. Specifically, we have extensively revised the main text, figures, and Methods, resulting in **28 new panels**, **24 modified panels**, and **2 new Supplementary Tables**. Additionally, we have developed a **web portal** to facilitate more accessible data download and visualization for the community.

A major focus of our revisions was to clarify the lipid annotation procedures and provide comprehensive molecular information on the identification and validation of the reported molecules. We have also elaborated on the technical aspects of our methodology, particularly regarding mass accuracy and lipid annotation, and expanded our analytical approaches to include additional dimensionality reduction methods and statistical validations.

To enhance the value of our work for the research community, we have developed a data portal (<https://ZEBRA-L.epfl.ch>) that serves as a central hub for accessing raw data, processed results, and interactive visualization tools. The portal is accompanied by detailed tutorials and documentation to facilitate data exploration and reuse. Additionally, we have refined our claims regarding the scope of our contribution, providing better context within the existing literature while emphasizing the specific advances our work offers in metabolomic and lipidomic imaging analysis. We believe that these revisions strengthen the technical and analytical impact of our work and make it more accessible to the research community.

Reviewer #1:

Remarks to the Author:

Schede et al. present a large MSI lipidomics dataset of zebrafish embryos at different time points of development and insights from analyzing this dataset.

The manuscript is interesting to read and has a number of positive things to be said about:

- the authors collected a large novel dataset from a fundamentally-relevant biological system at relevant time points of development — this is the key contribution of the manuscript
- they have developed a new approach for data processing to reduce batch effect and artifacts — please see below my discussion about it, points 2-3
- they have performed analysis of this dataset and found several patterns
- they have showed the complementary value of MSI compared to spatial transcriptomics, although in a limited way for one example section

However, there is a number of points which should be addressed if the manuscript will be selected for revision:

Major points

1.

The dataset should not be called an atlas. It's a collection of MSI datasets from sections of different zebrafish. There are no anatomical annotations provided, alignment between sections does not really provide necessary resolution (that's why authors needed to do extra coronal sections in Figure 7e-f), there is no alignment between individual animals. There is only 1 UMAP shown for (potentially) pixels from different sections (or animals), see Figure 4F, mainly to illustrate the batch effect. The dataset is not fully available and is not interoperable — I could not find the 3D volumetric data for all lipids reported with provided metadata. What is referred as "tutorial" provides on GitHub is a pretty technical script illustrating how a part of the data was created, for selected individual sections.

The point about what constitutes an atlas is well-received.

First, to clarify, we use "atlas" in line with current omics literature, denoting molecular composition surveys of biological systems. One of the contributions of our manuscript is a collection of mass spectrometry images that detail the lipid spatial distribution in a developing organism in 3D.

To raise the resource to the highest standards of the field and the expectations of the reviewer, we have:

1. **Created a website** available at <https://ZEBRA-L.epfl.ch> serving as a hub for raw data, processed data, an online interoperable data viewer, and associated tutorials/notebooks.
2. **Provided raw data** for each section with detailed instructions for loading and processing with uMAIA
 - a. Through the most widely used community-driven platform for spatial metabolomics, METASPACE: <https://metaspace2020.eu/project/uMAIA>
 - b. End-to-end workflow tutorial, from raw data to a basic multivariate analysis : https://github.com/lamanno-epfl/uMAIA/blob/main/uMAIA_tutorial.ipynb

- c. Tutorial for running MSI data with data from different instruments. (<https://github.com/lamanno-epfl/uMAIA/blob/main/tutorials/tutorial-peakCallingExamples.ipynb>).
3. Made available **uMAIA-preprocessed data** as standalone files in hdf5 format (for convenient python and R loading), complete with metadata, pixel coordinates, and annotations (<https://zenodo.org/records/14564876>). Accompanying tutorials demonstrate data loading and analysis procedures (available as vignettes from the website, as well as Jupyter Notebooks found at <https://github.com/lamanno-epfl/uMAIA/tree/main/tutorials>).
4. Deployed **an online lipid viewer** that allows the examination of lipid data in volumetric space across different time points. It is available under the 'Visualise' tab at <https://ZEBRA-L.epfl.ch>.

Panel 1: Overview of the website available at <https://ZEBRA-L.epfl.ch>. Top row: lipid viewer that enables lipid selection for all time points and classes. The viewer allows one to manipulate the objects in 3D. Bottom left: tutorial page that links to corresponding tutorials available on GitHub or vignettes. Bottom right: download page that links to all raw and processed data formats. For more information please visit the website.

Regarding the reviewer's suggestion for point-by-point coordinate registration between embryos, we must note that this remains a challenging task. While we have experience with similar efforts in mouse brain MALDI images using the Allen Institute's Common Coordinate Framework (CCF), the zebrafish community currently lacks an equivalent resource. Existing zebrafish reference atlases are limited to brain structures in larvae (3 and 6 dpf) or adults (>90 dpf) (<https://pmc.ncbi.nlm.nih.gov/articles/PMC10541682/>). Developing a CCF for embryonic stages would constitute a standalone project. For context, creating such a framework for the developing mouse brain required its own NIH-funded project, which recently culminated in a dedicated publication in *Nature Communications* (<https://www.nature.com/articles/s41467-024-53254-w>).

2.

The analysis of what they call “peak calling” and “network flow peak matching” is interesting. However, it seems to me that they are digging themselves out of a hole they created. As far as I could understand, MSI data in this paper is of relatively poor quality to the point when it would be strongly questioned in a mass spectrometry journal. Let me explain. I could not access the data to look into it (see the point below about Availability), but from Figure S4a it seems like the authors found that some peaks have mass shifts reach 20 ppm. Having m/z error of 20 ppm is not acceptable for the mass spectrometer authors use (Orbitrap) even for MALDI-imaging and even for a larger number of sections analyzed (when calibration may need to be performed between runs). If this is indeed the case, this on its own is a red flag as it can lead to false positive and false negative lipid identifications. Taking into account that the authors use simple m/z matching for lipid ID (with m/z tolerance of ~10ppm), their strategy to compensate for mass errors of 20 ppm can lead to false positives within m/z range of as high as 30 ppm (which is way too high for lipid ID). So, this raises a question, why authors proceeded with using data of such normally-inacceptably low mass accuracy. This also raises questions whether there is a general need for such sophisticated data preprocessing.

This comment stems from a misinterpretation we caused through the unclear presentation of **Figure S4a**. Our data does not have mass errors of 20 ppm, but of ~1.5 ppm before mass alignment (see Panel 4 below). The confusion likely arose because the figure displays shifts across multiple different molecules spanning a wide mass range, rather than the actual mass error for any single compound, but this was not clearly stated. We apologize for this and we take the occasion to enrich our reporting of technical metrics of data quality and QCs.

We have made the following changes to address this comment:

- Quantifications of mass error distributions for representative compounds spanning our mass range. **Panel 2 (Figure S4a)** shows mass errors for individual molecules across sections falling within 2 ppm.
- **Panel 3 (Figure S2f)** shows that, prior to mass alignment, the mean bin size across the entire dataset is **3.4 ppm**, as estimated using uMAIA's adaptive binning procedure. This provides a good estimation of the overall mass error in our measurements and corroborates the mass resolution observed in **Panel 2**.
- **Panel 4 (Revised S4b, previously S4a)** now clearly shows that mass errors for individual molecules across sections fall within 2 ppm.

Panel 2 (Figure S4a): Box and whisker plot representing the distribution of mass errors (in ppm) associated with 6 different ions spanning the m/z range between 500 and 830 within 19 sections from our 72 *hpf* zebrafish embryo dataset before (blue) and after (red) alignment with MSIWarp. Box and whisker plot in the bottom row indicate median mass error over sections

Panel 3 (Figure S2f) Left: estimated maximum mass errors after peak calling with uMAIA across all compounds and sections in the 72 *hpf* zebrafish. Number indicates mean value of distribution. Right: same as in the left panel, but for individual sections.

Panel 4 (Figure S4b) Diagram visualizing molecule matching outputs for a representative set of molecules. Markers indicate peaks detected in different sections of the datasets. Detections for each molecule matched by uMAIA are colored and connected with edges of the same color.

To clarify that we are not setting an artificial scenario (i.e., using data of low mass accuracy) for downstream analysis we have **pre-processed the data using standard mass alignment** methods, which compensate for coherent mass shifts among peaks, and demonstrated that our approach provides clear improvements under this setting whereby mass error for individual peaks fall below 1.5 ppm.

First, concerning peak calling, we were able to retrieve significantly more images that surpass image quality metrics with uMAIA peak calling compared to other methods both on non-aligned and aligned data (**Figure S3g; Panel 5**).

We have added to the text (189-192):

“Notably, even when data were pre-processed with standard mass alignment, uMAIA retrieved 33% more high quality images compared to existing methods (Figure S3g, see Methods 3.4).”

Regarding peak matching, while alignment significantly improves the detection efficiency of compounds for all methods tested, we show that our approach is still able to identify 23% more compounds across sections than the highest performing binning setup which does not introduce ambiguous matches ($m/z=0.0025$) (**Figure S4e-g; Panel 6**).

We have added to the text (lines 263-265):

“...a 20% increase in detected signals (uMAIA=1200, binning=1000) than the largest bin size tested (0.01) (Figure 3h-j). Importantly, this improvement was maintained even after the spectra were aligned with reference peaks (Figure S4e-g, see Methods 3.4).”

We have added Methods section 3.4 with details on how evaluation was performed using aligned data.

Panel 5 (Figure S3g): Barplots stratifying the output of the peak extraction from an MALDI-MSI experiment (72 hpf section) using uMAIA and MALDI-Quant after alignment with MSIWarp, and evaluated using the same quality metrics of Fig. 2h. Images are counted as True positives (TP) if they are extracted by a method and they surpass 2 image quality metrics from either method; False Positives (FP) if are extracted by the method but do not pass the quality metrics; False Negatives (FN) if were not called by a method while passing TP criteria for the other.

Panel 6 (Figure S4e-g): (e) Heatmaps visualizing the MALDI matrix ions (rows) identified for each section (column) for different binning sizes and uMAIA after realignment with MSIWarp. Color-coded according to whether 0, 1, 2, or 3 peaks were identified within the bin for a given section. The same matrix compounds as in (f) are displayed. (f) Violin plots reporting the distribution of Jaccard distances between isotopologue M+0 and M+1 presence across the acquisitions after featurization and alignment with MSIWarp. Distributions are computed across all the pairs of sections for the 20 molecules with the highest signal intensity (g) Percentage of compounds with incomplete features (left) and number of ambiguous detections (right) after realignment with MSIWarp for tested binning sizes and uMAIA.

To address concerns about annotation reliability, we now provide **Supplementary Table 1**, which lists all identified lipids along with their theoretical m/z values and the mass error of our measurements relative to these theoretical values. While we used a threshold of 10 ppm to assign an imaged peak to a known lipid, the vast majority (86%) of annotated peaks exhibit a mass error of <2 ppm compared to the reference compound (Figure S11a; Panel 7). This demonstrates that, in most cases, our assignments fall well within a credible range.

Additional checks and quality control measures were introduced to further validate the lipid annotations, as detailed in our response to major point 4 from this referee and to comments from other referees.

Panel 7 (Figure S11) (a) Histogram of mass errors for all annotated compounds with respect to measured m/z in the 72 *hpf* zebrafish embryo dataset.

3.

The authors over-blow the novelty of analyzing MSI data from many sections in their introduction and discussion; see e.g. “previous MSI analysis tool have focused on the analysis of a single or a small number of acquisitions” and “rendering analysis of >100 acquisitions virtually impossible”. First, their own dataset has less than 100 sections (only 96). This is not that much as already in 2013, Ojetjen et al, in Journal of Proteomics, 90, published a 3D MSI dataset of a mouse kidney containing 122 sections with all sections co-registered to create a coherent 3D volume, and the analysis was performed in a commercially-available software and resulted in substantially more pixels (2 million) than the authors present in their manuscript. Many more publications on 3D MSI were published since then. Many studies have analyzed more than 100 sections from different individuals, e.g. Porcari et al. in 2018, Analytical Chemistry 90(19), have performed a multi-center DESI-imaging study of samples from 103 patients. Another early example is Balluff et al. 2011, Am J Pathol 179(6) where MSI data from 181 tissue sections was analyzed. So, overall, although authors present an interesting dataset, it’s by far not the largest.

We thank the reviewer for this important feedback. We have modified the statements in our manuscript to ensure we do not overstate the novelty of analyzing multiple MSI sections, while more accurately qualifying our contribution with respect to the state of the art. We have revised passages in the **Introduction** and **Discussion** sections to specify that our contribution (1) pertains to metabolomics and lipidomics MSI data analysis and (2) advances the analytical capabilities of MSI data processing.

Concerning (1), while our approach can potentially handle proteomics data, we have not directly addressed this application and we now added the appropriate qualification in the Discussion (see below) to avoid suggesting otherwise. We have also referenced the papers mentioned by the reviewer to contextualize our work within the field's established literature.

Lines 766-771

“We envision uMAIA will become a central tool for the analysis of lipidomics and metabolomics datasets, however the method has not been tested on proteomics-focused

MSI experiments and tailoring it to these peculiarities constitutes an important direction for future improvements.”

Lines 741-746

“While analyses of large MSI datasets have been pursued for example in relation to the study of the proteome and organ variability [Balluff2011, Oetjen2015], batch effects and intensity variations across acquisitions have limited the scope of spatial metabolomic and lipidomic investigations. “

Concerning (2), The primary contribution of our work lies in advancing the analytical capabilities of MSI data processing, rather than in the scale of data collection. Specifically, our innovation centers on three key aspects: (a) developing an accessible framework for processing and integrating metabolomic imaging data, (b) enabling researchers to perform a fine-grained unbiased analysis and identify metabolic territories within a whole organism and (c) demonstrating these capabilities through a comprehensive developmental dataset that serves as a valuable resource for the lipidomics community. We acknowledge that we were neither the first to analyze multiple MSI sections nor do we claim to have the largest MALDI-MSI dataset. The seminal spatial proteomics studies cited by the reviewer have made valuable contributions to the field, and we have updated our manuscript to properly reference this work. However, our contribution addresses a distinct challenge in the analysis of metabolomics and lipidomics MSI data, where unbiased, discovery-oriented analyses at fine granularity have remained technically challenging.

We have clarified the scope of our work by revising the manuscript to better reflect uMAIA's current capabilities as well as limitations. The changes reflecting both consideration (1) and (2) now extend both the introduction and discussion section.

Lines 732-738

“We also developed a general and accessible computational framework, uMAIA, which rethinks the processing of MSI metabolomic data towards the integration of multiple acquisitions. This advancement unlocks new possibilities for the construction and analysis of whole-organism metabolomic atlases while facilitating unbiased analyses and ensuring reliable cross-acquisition comparisons.”

Lines 77-80:

“... However, large MSI datasets are challenging to integrate due to suboptimal image extraction, inappropriate molecule matching across samples, and intensity distortions (Figure S1a). For this reason, in previous work, the analysis of MSI datasets was limited to the application of supervised machine learning approaches or the study of major tissue compartments [Balluff2011, Oetjen2015, Porcari2018]. Here, we devised a computational framework which we named the unified Mass Imaging Analyzer (uMAIA) to tackle these problems. uMAIA allows the generation of 4D lipid atlases of whole organisms starting from large collections of raw MSI acquisitions (Figure 1d).”

Lines 739-756:

“MSI analysis of metabolites and lipids presents technical challenges that become particularly relevant when performing unbiased analyses of large datasets. While

analyses of large MSI datasets have been pursued for example in relation to the study of the proteome and organ variability [Balluff2011, Oetjen2015], batch effects and intensity variations across acquisitions have limited the scope of spatial metabolomic and lipidomic investigations. Previous studies adapted to these constraints through the use of supervised machine learning approaches [Balluff2011, Porcari2018] or by restricting analyses to major tissue compartments [Oetjen2015], as finer-scale analysis might have been confounded by technical variations. uMAIA addresses technical challenges by providing an integrated pipeline that enables both reliable cross-acquisition comparisons and unbiased detection of metabolic territories, expanding the analytical capabilities of large-scale metabolomic and lipidomic MSI studies."

To give a representative illustration of these technical challenges to the reviewer, we analyzed one of the datasets mentioned that performed unbiased analyses (of peptides composition), Oetjen et al. 2015. The paper claimed that major tissue subdivisions were discovered using clustering. Our examination on these already-processed data, revealed underlying batch effects that had survived regularisation attempts and that become apparent already in the PCA of the data (**Panel 8**). These effects did not impact the identification of the three major regions of the organ but became critical when attempting to reveal finer-grained compositional territories with a finer cluster analysis (pushing clusters ≥ 4). This observation illustrates why previous studies often limited their analyses to broader tissue subdivisions - a reasonable approach given the technical limitations at the time. However, the field increasingly needs tools that can reliably detect and account for these systematic variations to enable more granular analysis of tissue heterogeneity.

Panel 8: Outputs of the analyses on the proteomic MSI data of 10 consecutive sections of the mouse kidney from Oetjen et al. 2015. Left: PCA coordinates for each pixel and component visualized as images. Arrows indicate clear batch effects. Right: K-Means ($k=10$) clustering on the first 10 principal components of the data with UMAP embedding displayed beside each section.

4.

There are several concerns about the lipids ID:

1. The authors m/z-match lipids identified from bulk LC-MS/MS in both positive and negative polarity to peaks from MSI data collected in the positive polarity. This likely led to the false positive detection, likely increased for lipids detected in LC-MS/MS in the negative polarity. How do authors controlled this false positive detection?
2. How do the authors handle isomers?
3. At the considered tolerance in the m/z matching (and particularly after the discussed data preprocessing), the authors should definitely have isobars. And even if those isobars were not identified in LC-MS/MS, it's a well-known fact that they should be considered in lipid ID. However, I found no discussion or description how the authors handled it.
4. Which adducts were considered for LC-MS/MS and for MSI, in particular when doing the m/z-matching

We have clarified the annotation procedures in the Methods regarding all processing steps and added **Supplementary Tables 1 and 2** with all the information regarding annotated lipids and bulk lipidomics analysis. To provide information that responds to all the concerns raised by the reviewer we have:

1. **Revised the Methods section** to detail how lipid matching was performed. The quantitative bulk LC-MS/MS pipeline employed both negative and positive ion modes, providing a comprehensive dataset of lipids present in the sample (approximately 800 out of 2500 tested, specified in **Methods section 1.4**). For these lipids, we calculated the m/z values for their [M+H]⁺, [M+Na]⁺, [M+K]⁺, [M+NH₄]⁺, and [M+H-H₂O]⁺ adducts and matched their exact masses with MALDI ions, considering only those with an m/z difference below 10 ppm (specified in **Methods section 4.1**). The vast majority (86%) of such matches were with ions with less than 2 ppm difference (see **Panel 7**, revised **Figure S11a**). To further increase our confidence in the annotation of these compounds, we utilized the METASPACE platform. Over 2/3 of annotated m/z (67%) reported in the paper were confirmed by METASPACE (SwissLipids). We report **all possible annotations in Supplementary Table 1**.
2. A significant challenge for most MALDI-MSI pipelines is the inability to distinguish isomers; therefore, in reporting the imaged lipids, we refer to their sum composition, i.e., the total number of carbon atoms and double bonds. From quantitative bulk LC-MS/MS experiments, we can indicate the specific isomers present and their relative abundance. We **added this information to the Supplementary Table 1 and specified this in the Methods section 4.1** (lines 1490-1510).
3. All isobars that were detected were disentangled with the help of LC-MS/MS measurements. While for simplicity we consider only the most abundant isobar in visual representations, we **incorporated information about all isobars and their relative abundance in the Supplementary Table 1** and in the **Methods section 4.1**.
4. In this study, we considered only the positive adducts [M+H]⁺, [M+H-H₂O]⁺, [M+NH₄]⁺, [M+Na]⁺, and [M+K]⁺ when assigning compounds detected by MALDI MSI to lipids measured by bulk LC-MS/MS. For the LC-MS/MS analysis, the adducts considered were [M+H]⁺ and [M+NH₄]⁺ in positive ion mode, and [M+AcO]⁻, [M-2H]⁻, and [M-H]⁻ in negative ion mode. We added this information to **Supplementary Table 1** and specified

it in the **Methods** (sections 1.4 and 4.1 for bulk lipidomics experiment and annotation of imaged lipids, respectively). Additionally, we provide **Supplementary Table 2** with details of all lipid species detected by bulk LC-MS/MS, including compounds that were not imaged.

5.

Data is not available — that once again stresses the point that this dataset is not really an atlas. In any case, the data should be available as it's the key contribution of the paper.

1. For MSI datasets, I could not download any dataset from METASPACE (I only see a spinning wheel after clicking at the link)
2. For HybISS data, there is no link provided
3. 3D or 4D data is not available

We apologize for the inconvenience. This was due to a clerical error in the data link. **All MSI data are available and downloadable at this link:**

<https://metaspace2020.eu/project/uMAIA>

To further address the data availability concern:

1. As stated in point #1, we **created a website** that serves as a hub for raw data, processed data, an online interoperable **data viewer**, and associated tutorials/notebooks. The website is accessible at <https://zebra-L.epfl.ch/>
2. We have **provided the raw HybISS data as a Supplementary File** with x and y coordinates for each detected spot and the corresponding gene, and is available at <https://zenodo.org/records/14170238>
In addition, the **processed HybISS data** and the corresponding **MALDI MSI** section is now available at <https://zenodo.org/records/14514399>.
3. We have added tutorials on how to load and visualise the data, available at <https://github.com/lamanno-epfl/uMAIA/tree/main/tutorials>
4. We have made available **uMAIA-preprocessed data** as standalone files in hdf5 format (<https://zenodo.org/records/14564876>). Accompanying Jupyter Notebooks tutorials demonstrate data loading and analysis procedures (available as vignettes from the website, as well as Jupyter Notebooks) found at <https://github.com/lamanno-epfl/uMAIA/tree/main/tutorials>

Minor points

There are a few minor points:

1. There is a discrepancy about m/z bins in Fig 3d-f (0.001, 0.025, 0.05, 0.1) and Fig 3g-h (0.001, 0.0025, 0.005, and 0.01), and in Methods.

We have corrected this typo in Figure 3d,f. The correct values are 0.001, 0.0025, 0.005, 0.01.

2. The data does not have single-cell resolution and the statement “quasi-single-cell” in the abstract is confusing and should not be used.

We have removed the statement to avoid confusion.

3. In line 595, I'd recommend to avoid the word "strong" in "strong biochemical organization" as it's a subjective judgement based on the view of the network.

We have rewritten the sentence, avoiding subjective descriptors. Now it reads (line 636-639):

"Along similar lines, visualizing lipid-lipid relationships with a graph whose connectivity and layout is based on their spatial covariation revealed biochemical organization by lipid class".

Reviewer #2:

Remarks to the Author:

In the paper "Unified Mass Imaging Maps the Lipidome of the vertebrate development", Schede et al. describes a new computational workflow, uMAIA, for the processing and analysis of large scale MALDI-MSI data. They apply this pipeline to the first large scale un-targeted lipidomic spatial atlas of zebrafish development. The paper is an extremely precious resource for the community, is well written and fully understandable despite the high complexity of the underlying science and methods. The provided python pipeline was easily installed and tested successfully on two workstations (Mac and Ubuntu) despite long running time. Only a limited number of points has to be addressed:

Major points

1.

The 4D lipidomic atlas generated by the authors is of high interest for the Zebrafish community. However the processed data are not provided by the authors. As the number of people with the required expertise and computational resources to re-process and analyze the raw is extremely limited, the authors should provide a processed file that is easy to access and work with.

This is an important point about data accessibility, and we apologize for not having met these reasonable expectations. Indeed, a key goal of our work is to make this resource valuable and accessible to the broader zebrafish research community.

To address the data availability concern, we have:

1. **Created a website** available at <https://ZEBRA-L.epfl.ch> serving as a central hub for raw data, processed data, an online interoperable data viewer, and associated tutorials/notebooks.
2. **Provided raw data** for each section with detailed instructions for loading and processing with uMAIA (https://metaspace2020.eu/api_auth/review?prj=fd035620-e6de-11ee-86c1-7fce1db28462&token=DtylbPVV_FyN).
3. Made available **uMAIA-preprocessed data** as standalone files in hdf5 format (for convenient python and R loading), complete with metadata, pixel coordinates, and annotations (<https://zenodo.org/records/14524806>). Accompanying tutorials will demonstrate data loading and analysis procedures (available as vignettes from the

website, as well as Jupyter Notebooks found at <https://github.com/lamanno-epfl/uMAIA/tree/main/tutorials>

4. Produced and included **new anatomical annotations** of pixels according to embryo anatomy within the preprocessed data to enable further stratification.
5. Deployed **an online lipid viewer** that allows the examination of lipid data in volumetric space across different time points. It is available under the 'Visualise' tab at <https://ZEBRA-L.epfl.ch>.

2.

The authors have developed a very powerful and useful pipeline for the processing of the large scale MALDI data. However the analysis of the resulting processed data seems quite simple as there is no feature selection performed and that a simple PCA combined with k-means clustering is performed. As the algorithms used for the processed analysis of data can drastically impact the biological interpretation of the data (such as for scRNA-seq analysis). The authors should have tested different strategies of low dimensional embedding (SVD, NMF, autoencoder, laplacian eigenmaps, diffusion map etc...) and clustering (KNN-graph community based detection such as Louvain and Infomap). In addition, as the authors use a PCA, all lipids have the same contribution to the analysis, likely obscuring the signal of a few number but key lipid species. The impact of highly variable lipid selection and adjustment of the variance (notably implemented in the Pagoda2 pipeline) should thus be tested.

We thank the reviewer for this thoughtful comment. We note that we **deliberately kept** the analysis downstream of uMAIA processing **simple** to highlight how the method enables extracting biological insight **without requiring computational prowess**. However, we agree that additional analytical approaches could reveal a more nuanced biological structure within our data. To address this point, we have performed the analyses suggested and incorporated the following new figures and text into the manuscript.

We incorporated in **Figure S14** the approaches that yielded additional insights or enhanced interpretation of our results.

The application of NMF provided a complementary perspective to our initial analysis, as shown in **Figure S14b (Panel 9)**. By decomposing the lipid data into strictly positive factors, NMF generated more interpretable directions/lipid programs compared to traditional clustering approaches. This factorization enabled us to evaluate the spatial overlap of different lipid programs while maintaining biological interpretability. Through examination of the component weights, which exhibited greater sparsity than PCA-derived components, we identified key lipid species corresponding to these programs.

Diffusion maps, presented in **Figure S14d-h (Panel 10, 11)**, proved particularly valuable for representing a low dimensional embedding of lipid variations directly in lipid space. By leveraging their ability to effectively decompose the main directions of a neighborhood graph, we gained insights into subtle lipid covariations that are typically challenging to discern in standard spatial visualizations. When combined with lipid abundance overlays, this approach revealed fine-grained patterns in lipid distribution and covariation across the embryo (**Panel 11**).

The adoption of Leiden clustering, a KNN-graph community-based detection method, yielded results that largely corroborated our initial K-means clustering **Figure S14d,e (Panel 10)**. All these analyses were performed following a variance adjustment procedure adapted from the Pagoda2 pipeline for which we have added a description in **Methods section 4.3**. While Pagoda2 traditionally rescales features based on the relationship between expression level and variance in RNA sequencing data, this noise model and related scaling does not apply to MSI data and was therefore modified. Specifically, we implemented feature rescaling based on each feature's Moran's I score. This adaptation led to prioritize features that are more spatially variable than average.

Panel 9: (Figure S14b) Non-negative matrix factorization (NMF) of lipid distributions after feature selection and rescaling of features with corresponding Moran's I scores. Spatial distributions of the factors' coordinates shown (left). Factors were named by inspecting the top non-negative loadings. Lipids indicated are selected among the 5 with highest loadings per factor (right).

Panel 10 (Figure S14d-f) (d) Scatter plot matrix of pairs of diffusion map axes, with corresponding 2D representation achieved using UMAP. (e) Visualization of Leiden clusters in image for different sections. (f) Confusion matrix of the Leiden and K-Means cluster assignments .

Panel 11 (Figure S14g,h): (g) Low-dimensional embedding of the zebrafish MSI pixels (UMAP of 5-component diffusion map representations) colored by lipid intensities. (h) The same scatter plots of two selected lipids LPC intensities and their differences (upper row) with corresponding images (lower row).

Consistently with the additional value of these analyses the text, was modified as follows (lines 538-553):

“This correspondence between lipid distributions and anatomy generalized across the lipidome, as revealed by complementary low-dimensional representations of the data (Principal component analysis; PCA and Non negative matrix factorization; NMF). The distribution throughout the organism of pixels PCA coordinates indicated a strong anatomical covariance of lipid abundance and mapped its spatial structure (Figure 6c). Furthermore, NMF factors resolved this covariance by decomposing the data into positive, directly readable lipid contributions that correspond to axes of lipid program modulation (Figure S14b). To detect finer levels of covariation not apparent through image inspection, we used diffusion map embedding of pixels, revealing subtle distribution differences between related lipids such as LPC 18:1 and LPC 22:6 (Figure S14g,h).”

...

(lines 562-564)

“The correspondence between lipid territories and clusters was robust with respect to the clustering method applied (Figure S14d-f).”

Overall, complementing our original approach with new analyses has served both the purpose of (a) featuring the preprocessing capabilities (i.e., good preprocessing saves downstream efforts) and (b) maximizing the biological insights we can extract. Therefore, we demonstrate the robustness of our findings, and we highlight further details.

3.

The normalization model assumes the multiplication of the sample and feature related batch effects and is elegantly justified by the one-rank approximation of the empirical batch effect. Still, the authors should compare their multiplicative model with three other models:

An additive model, where the batch effect is $\lambda + \gamma$

A model with only the feature effect

A model with only the sample effect

We implemented and tested the suggested alternative models on real data. These models performed worse than our original formulation, an outcome we anticipated given that these models do not align with the noise structure we characterized in Figure 4b, and two of the

three models have fewer degrees of freedom.

Our analysis relied on both univariate and multivariate analyses of the outputs, which was necessary due to the absence of ground truth data. The single-intensity distribution of different lipids across different sections (**Panel 12, Figure S9a**) showed that the $+\lambda$ model underperformed, while $\lambda + \gamma$ emerged as the strongest of the three new alternatives. However, the default uMAIA still showed by far better performance, which was observed from the distribution of PC coordinates of pixels (**Figure S9b,c**). Additional dimensionality reduction using UMAP, with coloring by section, highlighted the residual batch effects in the alternative approaches (**Figure S9d**).

The effect of normalization on clustering outputs showed no improvement compared to what we observed with default uMAIA (**Figure S9e, Figure 4g**). We also examined PCA coordinates distribution across sections, similarly to what done with the spatial heatmaps in **Figure S8f**. Inspection of these coordinates revealed that the alternative models did not remove—and in some cases intensified—differences that appear technically rather than biologically driven, as evidenced by inconsistencies among adjacent sections (**Figure S9b**).

We have modified the main text to summarize these analyses (lines 391-395) :

"In addition, simpler noise models with different approaches to consider molecule- and acquisition-specific factors were not as effective at correcting intensity distortions across acquisitions (Figure S9, see Methods section 3.3)."

Panel 12 (Figure S9): (a) Intensity distributions of 10 different compounds in the 72 hpf zebrafish after normalization using 3 alternative models for batch-effect factors. Colors correspond to different sections. (b) Images of principal component coordinates of Raw, uMAIA and the 3 alternative models (see Methods 3.3). Black boxes indicate residual batch-effects present in the images. (c) PC coordinates of dataset after correction by various models across sections after normalization using the alternative models. (d) Low dimensional representation (UMAP embedding) of pixels for dataset after normalization by alternative models. Pixels are colored by the section from which they originate. (e) Heatmaps representing matrices counting the number of pixels belonging to a given pixel cluster across sections ('cluster count'), distribution of cluster proportion across sections ('row-normalized') and presence of cluster within section ('>10 pixels in cluster') within each section for raw and corrected data. Clustering was performed for each dataset independently.

4.

Identification of spatially-informative molecules across timepoints: the identification of those genes is, according to the method section, done by performing an ANOVA on the k-mean clusters. This approach (in addition to be basically double-dipping, see the work of Lucy Gao on the subject) is highly problematic as this approach will not identify genes based on their spatial pattern. Two different approaches could be used :

Basic spatial index scores, such as Geary's C or Moran's I. The measured values could be compared to the one observed in case of spatial shuffling and a corresponding p-value computed.

The (semi) variogram could be computed for each lipid and the ratio of the nugget effect compared to the total variance, allowing to identify genes which variance is primarily driven by spatial variations.

We thank the reviewer for this insightful suggestion. We have implemented the recommended analyses and incorporated them in **Figure 5 (Panel 13, 14)**, adopting the suggested statistical approach for identifying spatially variable molecules by comparing computed Moran's I values with an empirically derived null distribution through randomization.

We appreciate the need to be particularly stringent in these tests, and we scrutinized this aspect carefully. Indeed, our initial assessment revealed that simple pixel reshuffling is too trivial of a null model, as it fails to account for baseline expectations of lipid levels in different regions (e.g. expected lipid counts in a region could be confounded by varying degrees of cellularization). Using it would consistently result in very small p-values. Thus, we considered a more stringent null model that accounts for total cumulative lipids content to stratify the shuffling, resulting in a higher threshold for Moran's I. This methodology is now described in detail in the Methods (**section 4.2**, lines 1543-1551) :

"To identify spatially informative lipids, we implemented a statistical test based on Moran's I spatial autocorrelation index. For each lipid image, we calculated the observed Moran's I and compared it to a null distribution established through pixel value reshuffling. To account for local baselines, we stratified pixels based on their associated total-ion-count measurements into 8 bins. Then, independently for each lipid, the randomization procedure shuffled pixel values within these strata, maintaining a global notion of underlying tissue structure while disrupting specific lipid patterns. For each lipid, we computed the mean and standard deviation of the null distribution of Moran's I from the realizations (N=20) and calculated the Z-score for the observed Moran's I, converted to a p-value. P-values were adjusted using the Bonferroni method to control for multiple comparisons. Lipid images with adjusted p-values below 0.01 were classified as showing significant spatial structure."

We have updated **Figure 5** to report the results of the suggested approach. We further report the Moran's I of each of the lipids sorted by time point. While our fundamental observations regarding the increase in spatial variability remain consistent, they are now supported by a larger set of lipids and rest on more robust statistical foundations. The corresponding text in the manuscript has been revised to reflect these improvements (lines 457-461):

"We systematically assessed these changes and found that the number of lipids showing spatial patterns more than doubled from 29 at 8 hpf to 60 by 24 hpf and kept increasing as development proceeded, with 79 and 101 lipids at the 48 and 72 hpf stages, respectively (Figure 5e, f and S12a, see Methods 4.2)"

Panel 13 (Figure 5e,f): (e) Intensity distributions quantification of spatially-localized lipids across the sampled time points. Upset plot depicting total molecule counts for each timepoint (left). The time points that are considered within each set are indicated by solid black circles. Stacked bar plot depicting lipid class break up for different sets shown in upset plot (right). (f) Stacked barplots representing the total number (left) and proportions (right) of spatially-informative molecules detected in specified time points stratified by lipid class.

Panel 14 (Figure S12a): Barplots indicating the Moran's I score per lipid after subtraction of the expected score per time point. Colors are according to the sorting of lipids by the observed - expected score at 72 hpf.

Minor points

5.

Figure 5i : the different panels are poorly comparable as the order of the lipids is not conserved across. The same order should be used in to better visualize the evolution of the correlation structure. If a hierarchical clustering has been used please indicated it.

We agree with the referee that maintaining a consistent order of lipids across the heatmaps makes the figure easier to read. In the revised manuscript, we modified the order (**Figure 5i; Panel 15**). For completeness, we note that the originally displayed order was not random; the lipids were sorted by the SPIN algorithm at each timepoint (<https://academic.oup.com/bioinformatics/article/21/10/2301/206463>).

This approach highlighted the covariance structure present at each timepoint and confirmed that the covariance structure is increasing.

Panel 15 (Figure 5i): Correlation matrices between PC species for each developmental time plot sorted according to the optimal sorting at 72 hpf with SPIN. Colorbar indicates number of unsaturations for each lipid species. **(j)** Lipid intensity trends for three PC species (36:1, 36:2, 36:3) along the AP axis for each developmental stage (left). Varying degrees of unsaturations are indicated by colors. Heatmap of mean-centered average lipid intensity for the 3 species over developmental time (right).

6.

A more in-depth joint-analysis of the HybISS and MALDI data could have been performed, typically by using methods such canonical correlation analysis or its sparse variant (Extensions of sparse canonical correlation analysis with applications to genomic data)

We appreciate the reviewer's suggestion to use unsupervised analytical approaches such as CCA to deepen our joint analyses. However, our experimental design for the HybISS data was tailored to address focused questions about the relationship between lipidomic and transcriptomic spatial organization. Rather than pursuing exploratory analyses on our limited gene expression data that might yield fragile results, we have deepened the supervised analyses, which we reported below in our response to point #9.

7.

The peak calling algorithm is thoroughly described in the method section, however it quickly appears that the authors basically perform a one-dimensional watershed on the histogram

spectrum using local maxima as seeding points. The authors should clearly state that they have used this approach and discuss/explain the implicit parameter values of the watershed algorithm they have selected.

The peak calling algorithm we developed shares principles with watershed segmentation from computer vision. However, our method innovates by operating on the frequency histogram of measured m/z across an image rather than on intensity values as in traditional watershed approaches. This distinction is fundamental to handling mass spectrometry data effectively.

We have modified the main text (lines 155-156):

“Based on this principle and inspired by the watershed algorithm from computer vision [Najman and Schmitt 1994], we devised an approach where m/z intervals are initialized at the histogram maxima and then expanded until they either encounter other intervals or reach a background level of event counts (Figure 2e, see Methods 2.1)”

Methods section 2.1 to make it clear that our implementation is a modification of a 1D watershed algorithm on the histogram representation:

“Then, the following 2-steps (Algorithms 1 and 2) adaptive peak-calling procedure, inspired by the watershed algorithm from computer vision, is applied to the histogram representation of the data to retrieve the set of peaks M and their boundaries B . In brief, Algorithm 1 identifies seeding points based on peak maxima on the histogram representation, whereas Algorithm 2 retrieves the bin sizes for imaging.”

The parameters we use are described in the methods section. Default parameters are typically stable. To help users understand how to optimize their selection to their datasets, we now provide a tutorial (that can be found at <https://github.com/lamanno-epfl/uMAIA/blob/main/tutorials/tutorial-peakCallingExamples.ipynb>).

8.

Unclear terms used in page 5 to describe the normalization model: sensitivity and susceptibility. The term sensitivity is already extensively used in the machine learning literature. Instead, the terms ‘feature-related’ and ‘sample-related’ batch effects could be used.

We have removed these terms as they are not referenced later in the text to avoid confusion with machine learning literature.

9.

The authors explain that the metabolic genes are not useful to predict the different ‘lipidic regions’ and that highly variable genes are more useful. The authors could be maybe show the most highly predictive genes from the HybISS data.

To clarify, our claim is not that metabolic genes are not useful at all but that marker genes are more useful. To investigate which genes are most associated with lipid territories, to identify which genes were most predictive of lipidic regions, we calculated importance scores using a random forest classifier trained on our HybISS data. These results are now

displayed in **Figure S15f (Panel 16)**. We modified **Figure 6** to show this score beside the genes and reported this part of the analysis in the main text:

(lines 611-613) *“Only three of the ten most region-predicting genes were metabolic: apobb1, apoeb and fabp1b (Figure 6g and S15f, see Methods section 4.4).”*

We have modified the Methods (**section 4.4**, lines 1616-1619) to incorporate the details of this analysis:

“To identify the most informative genes for each lipid cluster, a decision tree was fit per cluster using gene expression as independent variables and binarized cluster labels as response variables (1 when belonging to the cluster in question, 0 otherwise). Feature importance was computed from the impurity decrease within each tree of the classifier for each lipid cluster”

Panel 16 (Figure S15f): (a) Feature importance of genes to predict lipid territories globally using decision tree classifiers. Red and gray bars indicate genes from the marker and metabolic subsets, respectively.

10.

The prior distribution of delta_c (signal difference between foreground and background) is set by the authors to Gaussian(3,1). While the use of such prior is reasonable, it still allows a negative value of delta_c. A prior that is only defined on positive real values could instead be used, typically a gamma or an exponential distribution.

We agree that delta_c should have a strictly positive domain and prior, yet the prior we use has 99.9% of the pdf in the positive domain. The reviewer can be assured that the current model never fits negative MAP estimates in practice - our preliminary tests showed no noticeable difference when using a strictly positive prior. Nonetheless, we appreciate theoretical rigor and have added to the uMAIA the version with a Gamma distribution of equivalent variance and mean.

Minor typo : Page 22, legend. “Generative model representation if (d)” and not (c).

We have made the required change in the legend.

Minor typo bis: Page 35. First A is not written in a gothic manner.

We have fixed the typo.

Reviewer #3:

Remarks to the Author:

The authors describe a new framework for joint analysis of large mass spec imaging datasets for the construction of multi-dimensional atlases. This approach attempts to combat the challenges faced in the field of mass spec imaging, particularly for lipids and metabolites. Overall the manuscript is of high caliber and comprehensively demonstrates the feasibility of such approach for even temporal image data. However it appears to be severely lacking in the lipidomic aspect, with very little detail on how annotations are derived for the MSI dataset and how the experiment was run. This needs to be corrected.

Critical

1.

How were the annotations derived for figure 5 and 6, specifically the lipid annotations? This needs to be explained and outlined in detail.

We clarified the annotation procedures in the **Methods (section 4.1)** regarding all processing steps and added **Supplementary Table 1** with all the information regarding annotated lipids (see also our reply to Reviewer 1 point 4). Specifically:

1. We modified **the Methods section 4.1** to include all the details regarding this aspect, which now reads:

“Annotation of the detected peaks was achieved by considering the m/z values retrieved from bulk quantitative lipidomics experiments. As most MALDI-MSI pipelines are unable to distinguish isomers, only the sum composition of lipids was taken into account. All lipids identified by bulk LC-MS/MS in positive and negative ion-mode were considered and H⁺, Na⁺, K⁺, H-H₂O⁺ and NH₄⁺ adducts were selected for matching to MALDI MSI data. MALDI compounds were matched to the nearest neighbor in m/z from possible annotations and only those with a distance below 0.01Da were included.

*In cases of multiple lipids matching a single m/z, including isobars and compounds with extremely similar m/z, the signal was attributed to the most abundant lipid, as measured by bulk lipidomics. To disentangle isobaric PC and PE species, as well as PC-P/O and PE-P/O, their relative abundance was specifically assessed (**Figure S11b,c**) and the m/z was assigned to the most abundant lipid. When comparable amounts of isobaric species were found, multiple annotations for the same m/z were reported. Lipid identity was further assessed by querying the SwissLipids database via the METASPACE online platform (Palmer et al 2016). Lipid annotations unconfirmed by SwissLipids (denoted by 'no' or blank entries in the 'Confirm_lipid_identity' column of Supplementary Table 1) should be interpreted with caution. “*

- “
- We added **Supplementary Table 1**, it indicates all annotated lipid species in the MALDI-MSI dataset that includes measured and theoretical m/z, potential isobars, isomers, annotation retrieved by METASPACE and relative abundances as assessed by LC-MS/MS. The Table is available for download at <https://zenodo.org/records/14514099>.

2. How are the authors disentangling PE and PC isomers (similarly for PC O and PE O). We assessed the signal obtained from PE and PC species for which we have no evidence of isobars in our dataset and found no significant difference in the ionization and detection efficiency by our MALDI system (**Figure S11b; Panel 17b**). We then evaluated the relative abundances of PC and related PE isobars (as obtained by LC-MS/MS) and plotted them (**Figure S11c; Panel 17**). As shown in **Panel 17c** the majority of peaks detected in the MALDI dataset can be ascribed to only one of the two classes with good confidence.

We have made the following adjustments:

- We **modified the Methods 4.1** to describe how we disentangled PE (PE-O) and PC (PC-O) isobars and their contributions to MALDI-MSI signals:

“In cases of multiple lipids matching a single m/z, including isobars and compounds with extremely similar m/z, the signal was attributed to the most abundant lipid, as measured by bulk lipidomics. To disentangle isobaric PC and PE species, as well as PC-P/O and PE-P/O, their relative abundance was specifically assessed (Figure S11b,c) and the m/z was assigned to the most abundant lipid. When comparable amounts of isobaric species were found, multiple annotations for the same m/z were reported. Lipid identity was further assessed by querying the SwissLipids database via the METASPACE online platform (Palmer et al 2016). Lipid annotations unconfirmed by SwissLipids (denoted by 'no' or blank entries in the 'Confirm_lipid_identity' column of Supplementary Table 1) should be interpreted with caution.”

- We added panels to **Figure S11 (a-c)** to visualize these results. Where a significant contribution comes from PE/PC isobars, this is clearly shown in the figures and stated in the text.
- All this information is also provided in **Supplementary Table 1**.

Panel 17 (Figure S11) (a) Histogram of mass errors for all annotated compounds with respect to measured m/z in the 72 *hpf* zebrafish embryo dataset. (b) Scatter plot depicting measured concentration from bulk LC/MS against MALDI-MSI quantification after TIC normalization. (c) Stacked barplot of the percent contribution of PC and PE isobars (based on LC/MS quantification) for specific m/z peaks identified in MALDI-MSI.

3.

How was the plasmenyl (PE P) and plasmanyl (PE O) annotations determined? Can the authors provide evidence of this (Figure S12)

Plasmenyl and plasmanyl lipids were annotated univocally when they could be discriminated by ion fragmentation in LC-MS/MS (see **Supplementary Table 2**, Sheet “Lipids Transition Table”). In all other cases both plasmenyl and plasmanyl annotations are reported.

4.

Can the authors provide more detailed mass spectrometry conditions for the data acquisition, notably the mass resolution, acquisition time for each pixel etc. Was the quadrupole used at all? Similarly, was this full-MS scan only?

We updated the **Methods section 1.2** (lines 1170-1179) to include this information:

“All MALDI-MSI acquisitions were performed using an AP-SMALDI5 AF system coupled to a Q Exactive orbitrap mass spectrometer, in full-MS scan and positive ion mode, within the 400–1200 m/z range. The MALDI laser focus was optimized manually using the source cameras aiming at a diameter of the focused beam of <5 μm. For each pixel, the spectrum was accumulated from 50 laser shots at 100 Hz. MS parameters in the Tune software (Thermo Fisher Scientific) were set to the spray voltage of 4 kV, S-Lens 100 eV, capillary temperature to 250°C. Calibration was performed regularly with internal standards (i.e., known matrix ions) and mass error kept within +/-2 ppm. A pixelated scan mode was used at a speed of 1.6 pixels per second. The spatial resolution for acquisitions of the 24, 48, and 72 hpf embryos was set to 7 μm. For the 8 hpf embryo, a resolution of 5 μm was used to account for its smaller size. In addition to these sections used for 3D image reconstruction, we acquired sections from a biological replicate for each time point. A total of 96 MS images were acquired.”

5.

Can the authors provide a table of masses, adducts and species reported in the MSI study? Similarly, this needs to be provided for the quantitative LC-MS study as well.

With the revised manuscript, we provide the requested information in two Supplementary Tables. More specifically:

1. We **created Supplementary Table 1**. This Table contains annotations identified in our MSI data with information on the table of masses, adducts, species and quantities (as measured by bulk LC/MS) reported in the MSI study.
2. We **created Supplementary Table 2**. This Table contains all information retrieved on the bulk LC/MS experiment.
3. Both Supplementary Tables are **available for download** from our website and Zenodo at the following link: <https://zenodo.org/records/14514099>.

6.

Brief detail (table format is fine) on how the species were annotated in this set is also required at minimum, particularly since this is a methods journal, and many of the species reported appear to be completely novel (PS O-19:1/18:1 in figure S9, unusual LPI 17:0 and 19:0 representing a large concentration on figure 7B.).

We thank the referee for this useful note; we agree that it is important to access the annotation rationale for each molecule. In response:

1. We have **modified the Methods 4.1** to include details regarding this aspect (see point #2)).
2. We have added **Supplementary Table 1** indicating all annotated lipid species in the MALDI-MSI dataset that includes measured and theoretical m/z, potential isobars, isomers, and confidence scores retrieved from METASPACE. We have also added **Supplementary Table 2** with all the information about the lipids retrieved by quantitative LC-MS/MS.
3. We re-inspected our raw data to evaluate specifically the confidence in the annotation of novel species. We confirm the identity of LPI 17:0 and 19:0, which we also found to be surprisingly abundant in zebrafish. This was not the case for PS O-19:1/18:1, which turned out to be misannotated. We corrected the annotation as this species is most likely PS 18:1/18:1.

Major

7.

The authors first describe overcoming technical variation by using an adaptive peak calling approach to better capture unique features. While the described results and figures show a relatively normally distributed mass error, can the authors comment on how this approach handle specific, systematic biases (e.g. temperature changes resulting in gradual shifts for TOFs) which can be instrument dependent, while many of the described results here are orbitrap based. This includes the figure highlighted in S2b, the pink feature that forms a bimodal distribution. Can a bimodal distribution be mistaken as two features? There does not appear to be a description of fine tuning the approaches when applying it. Similarly, simulations appear to be largely based on normal distribution when in real life instances, this might not the case.

The adaptive peak calling approach does not require mass errors to be Normally distributed. During the iterative procedure, the bin range will expand in one direction as long as the frequency at a given m/z surpasses the specified threshold.

To demonstrate how the method performs with distributions that are not normal or symmetric, we performed a simulation assuming mass errors are distributed differently, i.e., with skewed distributions. We have included these results in **Figure S3h (Panel 19) and Panel 18**. We have **modified the Methods 3.1** to describe this simulation with asymmetric peaks.

Panel 18: Frequency histogram of simulated dataset where mass errors are drawn from a Gamma distribution. Vertical red lines indicate uMAIA-detected compounds and colored bins depict the extent of the adaptive bin for each compound.

h Sensitivity Analysis - SNR and Molecular Crowding for Gamma-distributed Mass Errors

Panel 19 (Figure S3h): Line plot tracking the performance (mutual information score) of uMAIA on simulated spectra with mass errors drawn from a Gamma distribution for various extents of molecular crowding indicated by colors. Average of 50 simulations; standard deviation over realizations represented as shaded area.

In recognition of different needs associated with different instrumentation we provide additional analyses and material related to **MSI data from TOF instruments**. We provide a **tutorial** as part of our **GitHub** repo that details how to adapt input parameters for different use cases

(<https://github.com/lamanno-epfl/uMAIA/blob/main/tutorials/tutorial-peakCallingExamples.ipynb>).

Concerning the figure in **S2b**, we believe there is a misunderstanding of the plot: the pink circle indicates a specific pixel, not a feature (we acknowledge it was easy to miss this for how we plotted), and each dot is the estimated mass error of m/z features. We have improved the visualization and clarified that each panel represents a specific pixel in the image on the top right.

8.

In the proposed peak calling / matching approaches, it would be good to quantify the processing speed differences between this and the existing approaches described, as that would dictate feasibility of these approaches.

Considering data processing is not the bottleneck of MSI atlas creation (i.e. acquiring a single MSI image can take even more than 24h, more than our pipeline takes to process over 20 sections) we had originally neglected time profiling aspect. Therefore, we are really grateful to the reviewer for making us profile the components of uMAIA, as some discrepancies while time profiling lead us to discover parts of our implementation (e.g. related to how we accessed and copied tensors) was adding unjustified significant overhead to the actual core solvers. We have modified it now and improved the speed of the method by 1 order of magnitude in relevant use cases.

In the revised manuscript we give users precise information of the expected time required to run our method. We have profiled our method against existing ones that align the data, followed by binning. **Figure S3i (Panel 21)** and **Figure S4h (Panel 22)** report processing times between methods as dataset size increases.

We have added to the main text :

(lines 192-194) *“In addition, uMAIA processing times remained generally within the same order of magnitude as alignment methods (Figure S3i)”*.

(lines 272-275) *“In terms of processing time, features are also identified efficiently: datasets can be featurized for over 50 sections and 20,000 molecules in less than 15 minutes (Figure S4h)”*

Panel 21 (Figure S3i): Processing time for dataset as a function of number of spectra for uMAIA's peakcalling module and alignment with MSIWarp.

Panel 22 (Figure S4h): Processing time for uMAIA's matching module as a function of the number of sections provided to the algorithm.

9.

Can the authors better label the figures on S8? E.g. colors for the A and what they represent, figure s8e y axis etc. These are almost uninterpretable. Similar issues exist in figure 4e.

We modified Figure S8 and 4E to improve their clarity. Specifically, we added a color legend to S8A. Furthermore, we added Y-axis labels in S8E and 4E.

Panel 23 (Figure S8a): Signal intensity distributions across three acquisitions, color corresponding to the acquisition. On the left is the raw MALDI MSI data. To the right, the same data corrected with ComBat and uMAIA with related quantile-quantile plots representing the transformation between raw and normalized intensities

Panel 24 (Left: Figure S8e; and Right: Figure 4e): Distribution plots showing principal component loadings of pixels across different acquisitions for raw data and normalized data.

10.

Can the authors annotate better figure 5D? Notably some names are cut off and the adducts for each mass needs to be presented. Figure 6 needs to be corrected in a similar manner.

We apologize for this unnoticed error of figure formatting. We updated Figures 5D and 6 so that names are not cut off, and the adducts for each mass are displayed. We have also updated Figure 7 in the same way.

Panel 25 (Figure 4d): Distributions for selected lipids (max projection) for selected time points (sagittal and dorsal projections, schematic of orientation shown in top row with yolk outlined in yellow; D: Dorsal; A: Anterior; L: Lateral.). Note: precise orientation of 8 hpf embryo is unknown.

Panel 26 (Figure 5a): Sagittal and dorsal view of 3D reconstructions for 4 lipids with m/z value and lipid indicated. Schematic indicating yolk, head and tail displayed above. D: Dorsal; A: Anterior; L: Lateral.

Panel 27 (Figure 5e): Top row: medial section (sagittal view) of H&E staining overlaid with contours of metabolic regions shown in (d). Lipids that spatially localize to specific regions are indicated along with an inset of the H&E image. Both images are overlaid with contours representing delineated clusters.

11.

Figure 5G isn't described correctly (there are no loadings on my figure).

We corrected the mislabeling of Figure 5G.

12.

While the LC-MS/MS and the MSI results are independent samples, it would be worth correlating these results by summing up all pixel per sample in a given MSI analysis. Species identified in the MSI experiment should correlate and change in a similar fashion in this manner to the bulk lipidomics data. This can assist in determining the strength of lipid annotation in the MALDI analysis. E.g. plot PE 38:6 between the LC-MS (concentration) and MSI data (summed intensity) with all the timepoints.

We originally avoided reporting these correlations because their estimation is complicated not only by differences in sample preparation and ionization (requiring a global

renormalization) but also by technical challenges in summing across tissue volume signals with non-linear response. Nonetheless, a **relative assessment of correlations** to discover less reliably annotated lipids, in line with what was proposed by the referee, is feasible and valuable. Therefore, we performed this analysis, which led to further refinement of our final pool of annotated molecules; the results are summarized in **Figure S11g-i**.

Specifically this evaluation of correlations allowed us to:

- (i) Sort lipids by these correlations as a proxy of reliability of annotation, reporting many lipids with high reliability (**Figure S11g**)
- (ii) Discover and remove from our analyses 4 out-of-distribution lipids that did not correlate between the independent methods and are most likely false-positives (**Figure S11g and h**).
- (iii) Report the typical correlation per lipid species (**Figure S11i**).

We comment on these results in the main text:

(lines 446-451) *“To increase our confidence in the annotated lipids, we performed a global re-normalization of MALDI and LC-MS/MS signals and calculated the correlation of lipids detected in both (see Methods section 4.1) (Figure S11g-i). Lipids that did not correlate between the two independent methods were discarded as potential false positives”*

We also corrected the total number of lipids for which we can accurately represent 3D distributions:

(lines 442-445) *“From this dataset we annotated by exact m/z matching 176 lipids that were also detected by bulk lipidomics and for which we could reconstruct 3D distributions”*

Furthermore, the **Methods section 4.1** was expanded to include the details of the renormalization procedure and correlation computation.

Panel 28 (Figure S11g-i): (g) Boxplot and swarmplot of Pearson's R correlation scores between MALDI-MSI intensity sums and quantitative LC/MS measurements per lipid. Horizontal dashed line at 0. (h) Number of lipids with positive correlation coefficients in (g). (i) Median correlations in (g) after grouping by lipid class.

13.

Can the authors detail how the annotations described in line 489 was done?

Similarly to what we described in comment #1 of this reviewer, the revised manuscript details how the annotation was done in the **Methods sections 4.1**. Specifically to the aspects of the cited line (previously line 489, now 516), the methods now reads:

“Annotation of the detected peaks was achieved by considering the m/z values retrieved from bulk quantitative lipidomics experiments. As most MALDI-MSI pipelines are unable to distinguish isomers, only the sum composition of lipids was taken into account. All lipids identified by bulk LC-MS/MS in positive and negative ion-mode were considered and H⁺, Na⁺,K⁺, H-H₂O⁺ and NH₄⁺ adducts were selected for matching to MALDI MSI data. MALDI compounds were matched to the nearest neighbor in m/z from possible annotations and only those with a distance below 0.01 Da were included.

In cases of multiple lipids matching a single mz, including isobars and compounds with extremely similar m/z, the signal was attributed to the most abundant lipid, as measured by bulk lipidomics. To disentangle isobaric PC and PE species, as well as PC-P/O and PE-P/O, their relative abundance was specifically assessed (Figure S11b,c) and the mz was assigned to the most abundant lipid. When comparable amounts of isobaric species were found, multiple annotations for the same m/z were reported. Lipid identity was further assessed by querying the SwissLipids database via the METASPACE online platform (Palmer et al 2016). Lipid annotations unconfirmed by SwissLipids (denoted by 'no' or blank entries in the 'Confirm_lipid_identity' column of Supplementary Table 1) should be interpreted with caution. False-positive annotations were further removed by considering bulk LC-MS/MS data and MALDI-MSI data jointly. A global ad-hoc renormalization was implemented by identifying 4 coefficients to rescale all lipid concentrations from MALDI-MSI data that would maximize the average similarity between datasets. Lipids whose quantities did not result in a positive correlation coefficient were discarded from the analysis.”

14.

Can the authors outline the results in a supplementary table for Figure S9A-B?

We **added all information** and results about the bulk LC-MS/MS experiment in **Supplementary Table 2**.

15.

Can the authors outline the packages used for analysis across the figures in the methods section? Many of these are not commonly used, and many more have different versions available in the public space. Include versions used if available. Examples (not limited to) the UMAP projection, COMBAT normalisation, scArches etc.

We **updated the Methods section with a Table** containing the packages, many of which are more commonly used in the genomics community, along with their versions used across the figures. We also **uploaded a .yaml environment file** on GitHub so that the correct version of the packages can be easily installed.

Minor

16.

Can the authors provide details of the tests used for several analysis (e.g. Figure S9A has p-values and fold difference, but the analysis is not described anywhere).

The test was a Student's t-test between lipids concentrations from 8 and 72 hpf embryos (n=4). In the revised manuscript, the type of test is specified in the **figure legend** as well as the **Methods section 4.2**.

17.

There is a typo on page 38, line 1241.

We fixed the typo.

18.

Have the authors investigate what the green island on Figure 4F refers to, it appears to be a significant outlier for a batch.

The distinct green cluster in **Figure 4F** indeed represents a residual batch effect - the most pronounced one in our raw dataset. Here, we present **Panel 29** examining this case in detail, showing the feature intensity distributions that characterize this batch-specific variation. While our method successfully mitigates most technical variations, this example illustrates

that the complete removal of the strongest batch effects remains challenging.

Panel 29: Depiction of ‘green island’ in UMAP to corresponding spatial location (upper row). Spatial distributions of intensities for 4 molecules from raw data (bottom row). Red arrow indicates the section where the instrument picks up a very strong signal.

We believe it is important to be transparent about such limitations, as they help define the boundaries of what can be achieved with current normalization approaches. This case also demonstrates that our method **does not artificially force complete alignment** between batches, which could potentially obscure genuine biological variation in other contexts.

We modified the Limitation section to clarify this point:

(lines 846-852) *“Unlike methods that enforce batch overlap, uMAIA employs a balanced approach to normalization, guided by dataset-wide patterns while preserving individual image characteristics and biological variation. However, this implies that particularly extreme intensity distortions may still persist after normalization.”*

Editor response

We would like to thank the reviewers for their comments, which we have addressed below.

We have also performed a brief evaluation of our method on proteomic and metabolite MSI data. For this analysis we used publicly available datasets that represent consecutive, or highly similar, sets of acquisitions. The results have resulted in an addition of 6 panels that we describe in the Results and Discussion section of our manuscript.

Reviewer #2 (Remarks to the Author):

Review of “Unified Mass Imaging Maps the Lipidome of Vertebrate Development”

Major Points

1. Accessibility of the data and of the atlas

I would like to highlight the excellent work done by the authors to make the lipidomic data more accessible. The online atlas is easy to use and more importantly the processed data can easily be downloaded and analyzed using the Python script. In my case I have downloaded several of them and analyzed them using an in-house R script in less than a hour

We are very happy to hear that the atlas is easily accessible.

2. Improved analysis of the data

The authors have significantly improved the analysis of the pre-processed data through the use of NMF, diffusion map and graph-based clustering. The results they obtain are highly consistent with their initial results and even provided higher details, thus validating their initial approach and findings. It is worth noting that the authors nicely adapted the approach of Pagoda2 to the lipid data by rescaling the data according to Moran's I score !

We thank the reviewer for their previous suggestions and are pleased that they approve of the adaption of the Pagoda2 pipeline.

3. Comparison of different normalization models

The authors have benchmarked their (multiplicative) model against three other models, showing that only the multiplicative model is able to correct the majority of the observed batch effects. While the benchmark of the authors is quite exhaustive, I think that an additional plot that would summarize figure S9a and S9c by measuring the distance of the distribution (using Jensen Shanon or KL divergence for instance) between the different sections would be more efficient and easier to understand for readers.

We have added a summary panel (Extended Data 6f) for Extended Data 6a, c.

Panel Extended Data 6f: Average Wasserstein distance between neighboring sections for raw data and data after application of uMAIA and model variations across compounds (left) and top principal components (right).

4. Spatial analysis

The authors have successfully implemented a meaningful statistical test to detect spatially variable genes by carefully removing the background effect of total lipid amount. The only criticism I have is regarding the number of simulations that seems particularly low (N=20) and that could be increased.

We have increased the number of simulations to N=100. Results have not changed significantly. We have updated panels (Figures 5e,f, and Extended Data 8a) to reflect the modification in the simulation approach.

Figure 5 (e) Quantification of spatially-localized lipids across the sampled time points. Upset plot depicting total molecule counts for each timepoint (left). The time points that are considered within each set are indicated by solid black circles. Stacked bar plot depicting lipid class break up for different sets shown in upset plot (right). (f) Stacked barplots representing the total number (left) and proportions (right) of spatially-informative molecules detected in specified time points stratified by lipid class.

Additional points

While not being in the scope of this manuscript, knowing whether uMaia would work on other type of mass spectrometry data (typically proteomic or glycomics) and if it could be used to reanalyze previously published datasets is of high interest. Such benchmark of the tool by

the authors would be highly useful to the community and would be complementary to this first paper.

We have performed a brief evaluation of our method on proteomic and metabolite MSI data. For this we used publicly available datasets that represent consecutive, or highly similar, sets of acquisitions. The results suggest uMAIA applicability to a broader range of MSI acquisitions.

We have added the following sentences in the main text:

Analysis of a proteomic and metabolite MSI dataset suggested that our method was also effective at reducing intensity distortions captured by different modalities (Figure S5)

And in the discussion:

Furthermore, our initial tests also suggested applicability of uMAIA to other MSI datasets, including proteomics.

Figure S5 **Performance evaluation of uMAIA on proteomic and metabolite MSI data.** (a) Top 4 PC coordinates (columns) of 10 consecutive sections (rows) from kidney data (Oetjen et al., 2014) for raw data (left partition) and uMAIA-corrected data (right partition). (b) Same as in (a), but with metabolite MSI data from the NIH kidney dataset. (c) Intensity distributions of compounds for all sections before (upper row) and after uMAIA correction (lower row) for proteomic MSI data. Bold number above panels indicate m/z of compound. (d) Same as in (c), but with metabolite MSI data from the NIH kidney dataset. (e) Average Wasserstein distance between neighboring sections across all compounds from the proteomic MSI dataset for raw (orange) and uMAIA-corrected (blue) data. (f) Same as in (e), but with metabolite MSI data from the NIH kidney dataset.

5.

As previously describe in the first revision round, the code is reproducible and runs smoothly. The jupyter notebook are well documented and written. The only downside is that the analysis of the processed hdf5 file is only explained for Python and not for R users, who still represent a very important part of the bioinformatic community. I have written a small draft for the loading and visualization of the data in R that the authors could maybe expand and add to their repo.

We have uploaded a tutorial that instructs users how to load and analyse the hdf5 formatted object in R. The tutorial can be found at this link: <https://github.com/lamanno-epfl/uMAIA/blob/main/tutorials/R-LoadProcessedObjects.ipynb>

Reviewer #3 (Remarks to the Author):

The authors have addressed nearly all of my concerns and queries. A few notable items remain that require double checking, largely on the LC-MS/MS data used for annotating the spatial lipidomics data.

1.

The transitions provided by the authors for plasmenyl and plasmanylyl lipids does not appear to discriminate between the species for the LC-MS/MS analysis. For example, the two isomers PE O-16:1/22:6 and PE P-16:0/22:6 have identical retention times (1.75), precursor (746.5) and product ions (327.2), and are thus essentially duplicated in the dataset and reported twice for the same ion. Similar issues are observed for LPC O and LPC P, which some are duplicated.

This should be fixed. Unless there is evidence otherwise, PE P species should have the corresponding PE O annotated with it (e.g. PE O 16:1/22:6 & PE P 16:0/22:6).

Many of the isobaric species that are not unit resolution resolved are also in the dataset (e.g. PC O-16:0/16:0 and PC 15:0_16:0 are duplicated in RT, precursor and product).

Similarly for LPC O which are measured in positive mode, the 104.1 fragment would be present in the corresponding isobaric LPC species (e.g. LPC 17:0 and LPC O-18:0).

Can the authors remove all duplicates in the dataset. It would be sufficient to name them as a mixture (e.g. PC O-16:0/16:0 & PC 15:0_16:0) to avoid confusion for readers for ones that were duplicated.

We thank the reviewer for bringing our attention to this point. Indeed, in our first assessment there were redundant entries. We have revised all these cases, removed duplicates and renamed the lipids accordingly in the dataset and in the Supplementary Tables. We chose not to report all the possible plasmenyl and plasmanylyl species systematically: only those species which have been previously reported in literature were reported. In Supplementary Table 2 redundant lipids are highlighted in red and we added a legend which explains such cases.

2.

Oddly the concentrations are not identical for the duplicates. Can the authors provide the list of internal standards (aligned to the lipids that used it) that were used to calculate the concentrations for this study?

We now provide the list of internal standards used next to the quantified lipids in Supplementary Table 2.

```
library(rhdf5)
```

```
color_conversion=function(x,max_scale=NULL) {  
  f <- colorRamp(c("grey","yellow","orange","red"))  
  x=as.numeric(x)  
  if (is.null(max_scale)) {  
    max_scale=quantile(x,0.99,na.rm = TRUE)  
  }  
  x_prime=ifelse(x>max_scale,max_scale,x)  
  x_prime=x_prime/max_scale  
  x_color=f(x_prime)/255  
  x_color[!complete.cases(x_color),]=c(0,0,0)  
  x_color=rgb(x_color)  
  return(x_color)  
}
```

```
Load_umaia_data = function(h5_file_path) {  
  Data = h5read(h5_file_path,name = "/X")  
  Data = t(Data)  
  
  X = h5read(h5_file_path,name = "/uns/x_coord")  
  Y = h5read(h5_file_path,name = "/uns/y_coord")  
  Z = h5read(h5_file_path,name = "/uns/z_coord")  
  Space_location = data.frame(X = X, Y=Y,Z=Z)  
  
  ID_col = h5read(h5_file_path,name = "/var")  
  ID_col = ID_col$`0`  
  colnames(Data) = paste("mz_",ID_col,sep="")  
  return(list(Lipid_data=Data,Meta_data = Space_location))  
}
```

```
Spatial_plot = function(Feature = 1,Selected_cut=1) {  
  delta_X = max(Meta_data$Z[Meta_data$X==Selected_cut])-min(Meta_data$Z[Meta_data$X==Selected_cut])  
  delta_Y = max(Meta_data$Y[Meta_data$X==Selected_cut])-min(Meta_data$Y[Meta_data$X==Selected_cut])  
  delta_max = max(c(delta_X,delta_Y))  
  plot(Meta_data$Z[Meta_data$X==Selected_cut],Meta_data$Y[Meta_data$X==Selected_cut],pch=21,
```

```
xlim=c(min(Meta_data$Z[Meta_data$X==Selected_cut])*0.9,min(Meta_data$Z[Meta_data$X==Selected_cut])+delta_max)*1.05,
```

```
ylim=c(min(Meta_data$Y[Meta_data$X==Selected_cut])*0.9,min(Meta_data$Y[Meta_data$X==Selected_cut])+delta_max)*1.05,
```

```
  bg=color_conversion(Lipid_data[Meta_data$X==Selected_cut,Feature]),cex=0.55,lwd=0.1,xlab="X",ylab="Y",  
  main= paste(colnames(Lipid_data)[Feature]))  
}
```

```
X = Load_umaia_data("Downloads/processed24.h5ad")  
Lipid_data = X$Lipid_data  
Meta_data = X$Meta_data  
Spatial_plot(1,1)
```